# Fairness in Representation for Multilingual NLP: Insights from Controlled Experiments on Conditional Language Modeling

**Ada Wan**
University of Zurich
`ada.wan@uzh.ch`

## Abstract

We perform systematically and fairly controlled experiments with the 6-layer Transformer to investigate the hardness in conditional-language-modeling languages which have been traditionally considered morphologically rich (AR and RU) and poor (ZH). We evaluate through statistical comparisons across 30 possible language directions from the 6 languages of the United Nations Parallel Corpus across 5 data sizes on 3 representation levels — character, byte, and word. Results show that performance is relative to the representational granularity of each of the languages, not to the language as a whole. On the character and byte levels, we are able to eliminate statistically significant performance disparity, hence demonstrating that a language cannot be intrinsically hard. The disparity that mirrors the morphological complexity hierarchy is shown to be a byproduct of word segmentation. Evidence from data statistics, along with the fact that word segmentation is qualitatively indeterminate, renders a decades-long debate on morphological complexity (unless it is being intentionally modeled in a word-based, meaning-driven context) irrelevant in the context of computing. The intent of our work is to help effect more objectivity and adequacy in evaluation as well as fairness and inclusivity in experimental setup in the area of language and computing so to uphold diversity in Machine Learning and Artificial Intelligence research. Multilinguality is real and relevant in computing not due to canonical, structural linguistic concepts such as morphology or "words" in our minds, but rather standards related to internationalization and localization, such as character encoding — something which has thus far been sorely overlooked in our discourse and curricula.

## 1 Introduction

### 1.1 Background and motivation

Most current work on fairness in Machine Learning (ML) and Natural Language Processing (NLP) focuses on the societal biases encoded in natural language data that are propagated and amplified when they are used at scale for/as Artificial Intelligence (AI) solutions[1]. But little has been said or questioned about the bias, as in, the favoring of certain outcomes, implicit in our theoretical/scientific assumptions that results in the varying performance of different languages in computing.

**Disparity in machine translation results** For instance, results reported in Junczys-Dowmunt et al. (2016) for Phrase-Based Statistical Machine Translation (PBSMT) (Koehn et al., 2003) and Neural MT (Bahdanau et al., 2014) on the 6 official languages[2] of the United Nations (UN) Parallel Corpus (Ziemski et al., 2016) indicate a disparity between EN/ES/FR and AR/RU/ZH in BLEU (Papineni et al., 2002) — translation performance in the latter group is generally worse, regardless of the MT algorithm used. AR and RU are traditionally considered morphologically complex (see e.g. Minkov et al. (2007), Seddah et al. (2010) and proceedings of related workshops in subsequent

---

[1] see e.g. work from conference (`https://facctconference.org`) and workshops in previous years on "Fairness, Accountability, and Transparency" (FAccT)

[2] Arabic (AR), English (EN), Spanish (ES), French (FR), Russian (RU), and Chinese (ZH)

years for *Statistical Parsing of Morphologically Rich Languages*), and ZH morphologically frugal (for its lacking determiners and plural or tense markers) (Koehn, 2005). While Koehn (2005) found translating into EN to be easier than into morphologically rich languages based on word-level BLEU scores from PBSMT systems of 110 language directions from the 11 Europarl languages then, Bugliarello et al. (2020) found it is easier to translate out of EN than into it based on 21 Europarl languages in BPEs (Byte Pair Encodings) (Sennrich et al., 2016) with the Transformer (Vaswani et al., 2017) in a new metric, cross-mutual information.

**Disparity in language modeling results**    Disparate performances across different languages seem to have been implicitly accepted in that it is often believed that some languages are harder to model than others. Bender (2009) advocated the relevance of linguistic typology for the design of language-universal NLP systems due to differences based on crosslinguistic structural notions, such as parts of speech and morphological complexity. Cotterell et al. (2018) studied (monolingual) language models (LMs) on the 21 Europarl languages using a word-level 7-gram standard Kneser & Ney (1995) model and LSTM-LMs (Sundermeyer et al., 2012) with characters and lemmatized forms in information-theoretic terms, and found morphological complexity to be the primary culprit for the differences in performance. Mielke et al. (2019) extended the coverage to 69 languages with the multilingual Bible corpus (Mayer & Cysouw, 2014), tested on RNN-LMs (an implementation of LSTM (Hochreiter & Schmidhuber, 1997)) with characters and BPEs, but concluded that basic data statistics in vocabulary size ($|V|$) and sequence length were the most predictive performance features.

We noticed, however, a discrepancy in the results from Mielke et al. (2019) for ZH — it came out as the least difficult for the character model, but it is the 6th most difficult language for the BPE model. As different input representations have been tested with different architectures with divergent results in different metrics in previous studies, each of them only testing with one data size, we decided to investigate the matter more systematically once again with statistical comparisons of score distributions between languages.

## 1.2    RESEARCH QUESTIONS AND CONTRIBUTIONS

**Research questions**    Are there any statistically significant differences in hardness when it comes to Conditional-Language-Modeling (CLMing) languages which have been traditionally considered morphologically rich (AR and RU) and poor (ZH) with the 6-layer Transformer? Is morphological complexity inherent in language? When is the notion of morphological complexity relevant in computing?

**Summary of findings and insights**    Based on our bilingual CLMing setup with the UN Parallel Corpus in the data size range from $10^2$ to $10^6$ lines on the character, byte, and word levels, we find:

1. Language has many finer-grained dimensions with different representations and learning patterns. Hardness in modeling is relative to its representational granularity (*representation relativity*).
2. There is neutralization of source language instances, i.e. there are no statistically significant differences between source language pairs. Only pairs of target languages differ significantly.
3. On the character and byte levels, hardness is correlated with statistical properties concerning sequence length and $|V|$ of a language, regardless of its morphological profile. As it is possible to eliminate performance disparity by decomposing sequences into finer-grained units in characters and bytes, we show that morphological complexity is not an intrinsic property of language. Unless word-based methods are used, or unless we implement/model it explicitly, the notion of morphological complexity is irrelevant in computing.
4. On the word level, hardness is correlated with $|V|$, and a complexity hierarchy arises through the manual preprocessing step of word tokenization. This complexity/disparity effected by word segmentation can be improved by subword tokenization but cannot be eliminated due to the fundamental qualitative differences in the definition of a "word" being one that neither holds universally nor is suitable/consistent for fair crosslinguistic comparisons.
5. Representational units of finer granularity can help close the gap in performance disparity.

Orthogonal to our main research questions, we also observed 2 types of sample-wise non-monotonicity — Double Descent (Belkin et al., 2019; Nakkiran et al., 2020) and *erraticity*. For reasons due to length and scope for this paper, we will defer discussions and analyses of these beyond what is addressed in § 5 to future work.

**Outline of the paper**    In § 2, we define our method and experimental setup. We present our results and analysis on the primary representations in § 3 and those from the secondary set of controls in § 4 in a progressive manner to ease understanding. Meta analysis on performance disparity and other discussions are in § 5.

## 2    METHOD AND DEFINITIONS

**Conditional language modeling**    CLMing is the modeling of the probability of the next token, given the history of the preceding tokens and conditioning context. In our case, such conditioning context is a line from the source language. To explicitly focus on modeling the complexities that may or may not be *intrinsic* to the languages, we study the more fundamental process of CLMing without performing any translation. This allows us to eliminate confounds associated with generation and other evaluation metrics. One could think of our setup as estimating conditional probabilities with the Transformer, with a bilingual (one-to-one) setup where the perplexity (PP) of one target language ($l_{trg}$) is estimated given the parallel data in one source language ($l_{src}$), where $l_{src} \neq l_{trg}$. We focus on the very basics and examine the first step in our pipeline — input representations, holding everything else constant. Instead of measuring absolute cross-entropy scores at one data size, we evaluate the relative differences between development (dev) set score distributions between languages.

**Controlled experiments as basic research for scientific understanding of language data**    Using the UN Parallel Corpus, the data from which the MT results in Junczys-Dowmunt et al. (2016) stem, we perform a series of controlled experiments with the Transformer, holding the hyperparameter settings for all 30 one-to-one language directions from the 6 languages constant. We control for size (from $10^2$ to $10^6$ lines) and language with respect to representational granularity. We examine 3 primary representation types/levels — character, byte (UTF-8), and word, and upon encountering some unusual phenomena, we perform a secondary set of controls with 5 alternate representations — on the character level: Pinyin and Wubi (ASCII representations for ZH phones and character strokes, respectively), on the byte level: code page 1256 (for AR) and code page 1251 (for RU), and on the word level: BPE. These symbolic variants allow us to manipulate the statistical properties of the representations, while staying as "faithful" to the language as possible. We adopt this symbolic data-centric[3] approach because we would like to more directly interpret the confounds, if any, that make language data different from other data types. We operate on a smaller data size range as this is more common in traditional language sciences and one of our higher goals is to bridge an understanding between language sciences and engineering (the latter being the dominant focus in NLP), and between traditional symbolic sciences and ML. We run statistical tests to identify the strongest correlates of performance and to assess whether the differences between the mean performance of different groups are indeed significant. We are concerned *not* with the absolute scores, but with the *differences* between score distributions from different languages.

**Fair evaluation with multitexts**    Multitexts are multiway parallel corpora. The UN Parallel Corpus is a 6-way parallel corpus consisting of manually translated UN documents from the 25-year period between 1990 and 2014. We use the UN Parallel Corpus because it contains languages conventionally regarded as morphologically rich and poor, has quality and size sufficient for evaluation, and more importantly, it comes as raw texts (untokenized), unlike both of the corpora that Mielke et al. (2019) used. Detokenization (esp. the evaluation thereof) is not a trivial task.

**Fair information-theoretic evaluation metric**    Most sequence-to-sequence models are optimized using a cross-entropy loss, defined as:

$$H(\boldsymbol{t}, \boldsymbol{s}) = -\sum_{i=1}^{N} \log_2 p(t_i \mid \boldsymbol{t}_{<i}, \boldsymbol{s}) \tag{1}$$

where $\boldsymbol{t}$ is the sequence of tokens to be predicted, $t_i$ refers to the $i^{th}$ token in that sequence, $\boldsymbol{s}$ is the sequence of tokens conditioned on, and $N = |\boldsymbol{t}|$. It is customary to report scores as PP, which is $2^{\frac{1}{N}H(\boldsymbol{t}, \boldsymbol{s})}$, i.e. 2 to the power of the cross-entropy averaged by the number of tokens in the dev

---

[3]Two testing/evaluation approaches — data-centric: hold the algorithm constant and tweak data, vs. algorithm-centric: hold data constant and tweak the algorithm.

data. Cotterell et al. (2018) proposed to use "renormalized" PP to evaluate LMs tokenwise fairly by dividing the overall bits per utterance/sequence by one constant token count in any one arbitrary language (e.g. so to arrive at "bits per character" in one language to evaluate all languages). But we find that it is not necessary to assign a perspective that is centered on any one particular language, when we can evaluate simply by the total number of bits for a larger portion of texts/sequences. This can be a fairer, more general and flexible way of evaluating data that has not been or cannot be perfectly segmented or aligned line by line. We hence used instead *unnormalized* PP, i.e. the total number of bits needed to encode the dev set (3,077 lines per language, after length filtering, in our case). As the implementation we used only reports PP, we transformed it back to entropy as defined above via $H(\boldsymbol{t}, \boldsymbol{s}) = \log_2 PP(\boldsymbol{t}|\boldsymbol{s}) \times N$.

**Disparity/Inequality** In the context of our CLMing experiments, we consider there to be "disparity" or "inequality" between languages $l_1$ and $l_2$ if there are significant differences between the performance distributions of these two languages with respect to each representation. Here, by performance we mean the number of bits required to encode the held-out data using a trained CLM. With 30 directions, there are 15 pairs of source languages $(l_{\text{src1}}, l_{\text{src2}})$ and 15 pairs of target languages $(l_{\text{trg1}}, l_{\text{trg2}})$ possible. We compare the source languages among each other, and the target languages among each other. Each $l_{\text{src}}$ or each $l_{\text{trg}}$ consists of scores from all models trained across various sizes and directions. To assess whether the differences are significant, we perform unpaired two-sided significance tests with the null hypothesis that the score distributions for the two languages are not different. Upon testing for normality with the Shapiro-Wilk test (Shapiro & Wilk, 1965; Royston, 1995), we use the parametric unpaired two-sample Welch's t-test (Welch, 1947) (when normal) or the non-parametric unpaired Wilcoxon test (Wilcoxon, 1945) (when not normal) for the comparisons. We use the implementation in R (R Core Team, 2014) for these 3 tests. To account for the multiple comparisons we are performing, we correct all p-values using Bonferroni correction (Benjamini & Heller, 2008; Dror et al., 2017) and follow Holm's procedure[4] (Holm, 1979; Dror et al., 2017) to identify the pairs of $l_1$ and $l_2$ with significant differences after correction. We report all 3 levels of significance ($\alpha \leq 0.05, 0.01, 0.001$) for a more comprehensive overview. In contrast to Dror et al. (2017), which aimed to compare the performance of different algorithms, we compare languages (in the context of computing).

**Experimental setup** The systematic, identical treatment we give to our data is described as follows with further preprocessing and hyperparameter details in Appendices A and B, respectively.

After filtering length to 300 characters maximum per line in parallel for the 6 languages, we made 3 subsets of the data with 1 million lines each — one having lines in the order of the original corpus (dataset A) and two other randomly sampled (without replacement) from the full corpus (datasets B & C). Lines in all datasets are extracted in parallel and remain fully aligned for the 6 languages. For each run and each representation, there are 30 pairwise directions (i.e. one $l_{\text{src}}$ to one $l_{\text{trg}}$) that result from the 6 languages. We trained all 150 (for 5 sizes) 6-layer Transformer models for each run using the SOCKEYE Toolkit (Hieber et al., 2018). We optimize using PP and use early stopping if no PP improvement occurs after 3 checkpoints up to 50 epochs maximum, taking the best checkpoint. Characters and bytes are supposed to mitigate the out-of-vocabulary (OOV) problem on the word level. In order to assess the effect of modeling with finer granularity more precisely, all vocabulary items appearing once in the train set are accounted for (i.e. full vocabulary on train, as in Gerz et al. (2018a;b)). But we allow our system to categorize all unknown items in the dev set to be unknown (UNK) so to measure OOVs (open vocabulary on dev (Jurafsky & Martin, 2009)). To identify correlates of performance, we compute Spearman's correlation (Spearman, 1904) with some basic statistical properties of the data (e.g. length, $|V|$, type-token-ratio, OOV rate) as metrics — a complete list is provided in App. C. See App. D for sample construction for statistical comparisons.

## 3 EXPERIMENTAL RESULTS OF PRIMARY REPRESENTATIONS

Subfigures 1a, 1b, and 1c show the mean results across 12 runs of the 3 primary representations — character, byte, and word, respectively. The x-axis represents data size in number of lines and the y-axis the total conditional cross-entropy, measured in bits (Eq. 1). Each line connects 5 data points corresponding to the number of bits the CLMs (trained with training data of $10^2$, $10^3$, $10^4$,

---

[4]using implementation from `https://github.com/rtmdrr/replicability-analysis-NLP`

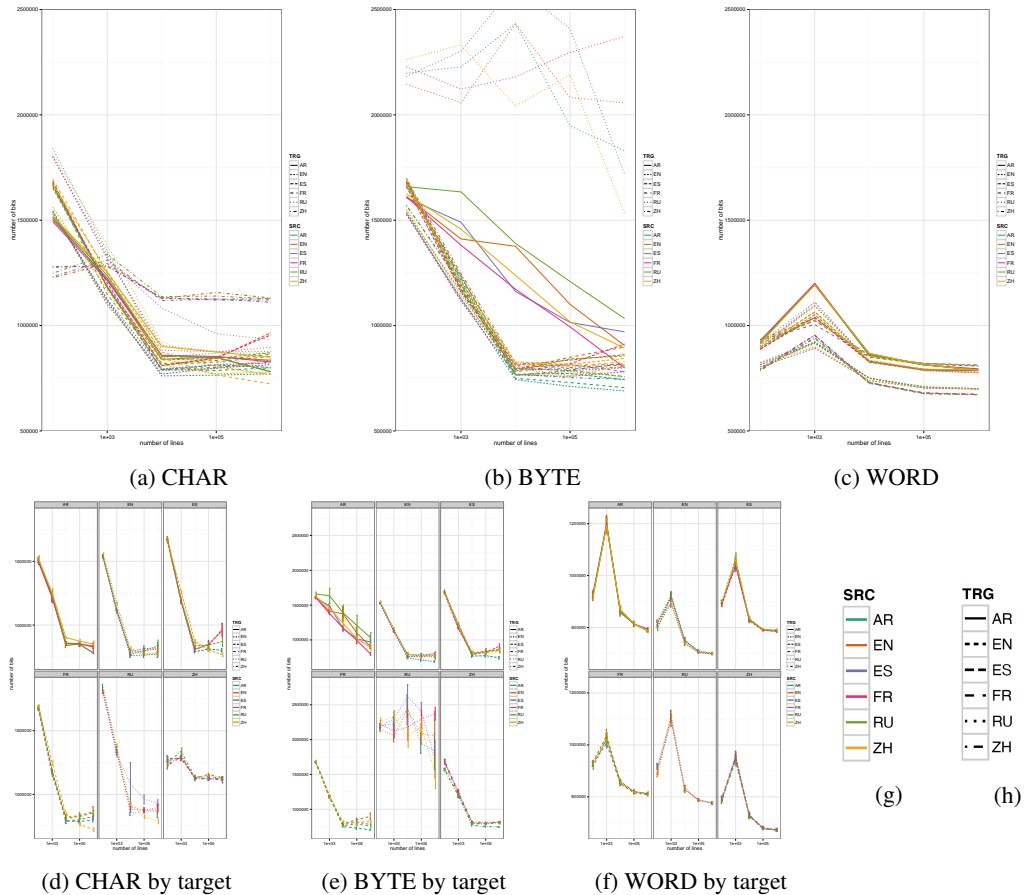

(a) CHAR  (b) BYTE  (c) WORD

(d) CHAR by target  (e) BYTE by target  (f) WORD by target

(g)  (h)

Figure 1: Number of bits (the lower the better) across data size from $10^2$ to $10^6$ lines plotted for all 30 directions. Subfigures 1a, 1b, and 1c show mean scores across 12 runs. Subfigures 1d, 1e, and 1f depict the corresponding information respectively sorted in 6 facets by target language and with error bars. Legend in Subfigure 1g shows the correspondence between colors and source languages, in Subfigure 1h between line types and target languages. (These figures are also shown enlarged in Appendix F. Please note that results pertinent to our first research question of this paper concerning statistically significant differences are summarized in Table 1, figures are a visual aid only. We are not concerned with the absolute scores but the distances between scores, i.e. spaces between the sets of lines by $l_{\text{trg}}$. The point here is to show the differences in Transformer's overall learning patterns relative to the representational granularity.)

$10^5$, and $10^6$ lines) needed to encode the target language dev set given the corresponding text in the source language. These are the same data in the same 30 language directions and 5 sizes with the same training regime, just preprocessed/segmented differently. This confirms **representation relativity** — hardness in modeling is relative to its representational granularity. Languages (or any objects being modeled) need to be evaluated relative to their representation. "One size does not fit all" (Durrani et al., 2019). Our conventional way of referring to "language" (as a socio-cultural product or with traditional word-based approaches, or even for most multilingual tasks and competitions) is too coarse-grained for computing (see also Fisch et al. (2019) and Ponti et al. (2020)).

Subfigures 1d, 1e, and 1f display the corresponding information sorted into facets by target language, source languages represented as line types. Through these we see more clearly that results can be grouped rather neatly by target language — as implicit in the Transformer's architecture, the decoder is unaware of the source language in the encoder. As shown in Table 1 in § 5 summarizing the number of source and target language pairs with significant differences, there are **no significant differences across any source language pairs**. The Transformer neutralizes source language instances. This could explain why transfer learning or multilingual/zero-shot translation (Johnson et al., 2017) is possible at all on a conceptual level.

In general, for character and byte models, most language directions do seem to converge at $10^4$ lines to similar values across all target languages, with few notable exceptions. There are some fluctuations

past $10^4$, indicating further tuning of hyperparameters would be beneficial due to our present setting possibly working most favorably at $10^4$. On the character level, target language ZH ($\text{ZH}_{\text{trg}}$) shows a different learning pattern throughout. And on the byte level, $\text{AR}_{\text{trg}}$ and $\text{RU}_{\text{trg}}$ display highly unstable behavior, which we refer to as *erratic*. Word models exhibit Double Descent across the board (note the spike at $10^3$), but overall, difficult/easy languages stay consistent, with AR and RU being the hardest, followed by ES and FR, then EN and ZH. A practical takeaway from this set of experiments: in order to obtain more robust training results, use bytes for ZH (as suggested in Li et al. (2019a)) and characters for AR and RU (e.g. Lee et al. (2017)) — also if one wanted to avoid any "class" problems in performance disparity with words. Performance disparity for these representations is reported in Table 1 under "CHAR", "BYTE", and "WORD". Do note, however, that the intrinsic performance of ZH with word segmentation is not particularly subpar. But this often does not correlate with its poorer downstream tasks results (recall results from Junczys-Dowmunt et al. (2016)). Since the notion of word in ZH is highly contested and ambiguous — i) it is often aimed to align with that in other languages so to accommodate academic theories and manual feature engineering[5], ii) there is great variation among different conventions, and iii) native ZH speakers identify characters as words — there are reasons to rethink this procedure now that fairer and language-independent processing in finer granularity is possible. Li et al. (2019b) questioned the necessity of CWS in Deep Learning (DL)-based ZH NLP and presented evidence in favor of character-based processing, including results from downstream NLP tasks. In Linguistics, Duanmu (2017) presented a summary on the contested nature of wordhood in (Mandarin) ZH in relation to EN. A more native account of ZH, however, despite a couple of dialects/varieties of it being considered a high-resource language, has not yet been fully recognized and accepted in NLP.

## 4 UNDERSTANDING THE PHENOMENA WITH ALTERNATE REPRESENTATIONS

To understand why some languages show different results than others, we carried out a secondary set of controlled experiments with representations targeting the problematic statistical properties of the corresponding target languages.

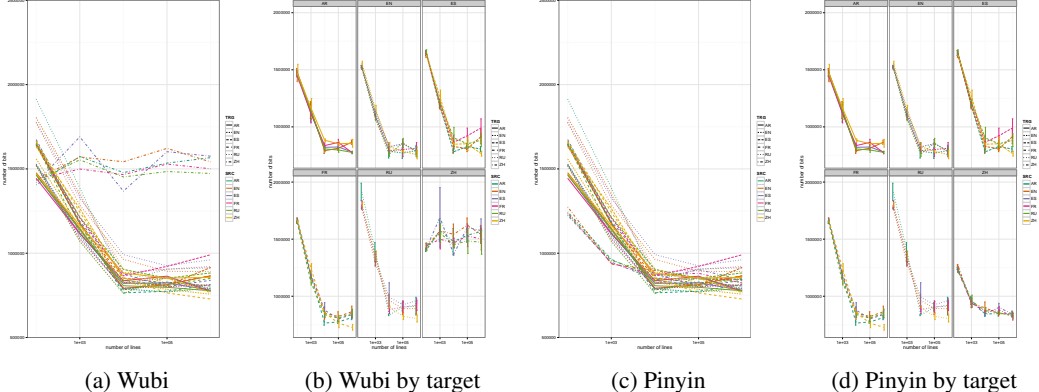

| (a) Wubi | (b) Wubi by target | (c) Pinyin | (d) Pinyin by target |

Figure 2: Character-level remedies for ZH: Wubi vs. Pinyin.

**Character level** We reduced the high $|V|$ in ZH with representations in ASCII characters — Pinyin and Wubi. We replaced the ZH data in these formats *only on the target side* and reran the experiments involving $\text{ZH}_{\text{trg}}$ on the character level. Results in Figure 2 and Table 1 show that the elimination of

---

[5]It is a "legacy interpretation" which stemmed from a practical compromise from the early days in ZH NLP when the goal was to align with EN words for MT. Chinese word segmentation (CWS) has been a decades-long issue in text processing. But even in EN, for computing, the variability in "word" counts (from the trivial convention of whitespace tokenization) results in different bit counts, affecting file sizes. In NLP, such method of "word" counting brings about a high $|V|$, hence different tokenization schemes have been designed to mitigate this problem. For humans, there is no consensus about the definition of "words". Even for a purely academic account, it is held to be indeterminate (see Haspelmath (2011) and references therein from the past century). Kilgarriff (1997; 2014) pointed out that "words" and "word senses" and the number thereof, in terms of lexical entries for dictionaries, are contextual and arbitrary.

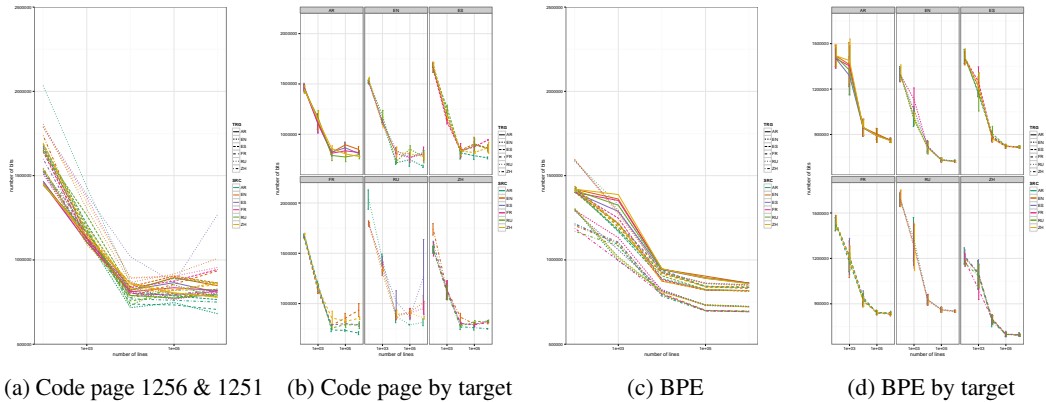

(a) Code page 1256 & 1251      (b) Code page by target            (c) BPE                  (d) BPE by target

Figure 3: Byte-level (Subfigures 3a & 3b) remedies with code page 1256 for target AR and 1251 for target RU, and word-level (Subfigures 3c & 3d) remedy with BPE for all languages.

disparity on the character level is possible if ZH is represented through Pinyin (transliteration), as in Subfigure 2c, though at the cost of the native script information. Models represented through Wubi, an input algorithm that decomposes character-internal information into stroke shape and ordering and matches these to 5 classes of radicals (Lunde, 2008), display a behavioral tendency unlike those with other (phonetic) alphabetic scripts[6] (Subfigure 2a), suggesting that this script/stroke pattern decomposes differently. But ZH the language is not an outlier all around.

**Byte level**    Length is the most salient statistical attribute that makes AR and RU outliers. To shorten their sequence lengths, we tested with alternate encodings on $AR_{trg}$ and $RU_{trg}$ — code page 1256 and 1251, which provide 1-byte encodings specific to AR and RU, respectively. Results are shown in Subfigures 3a and 3b. Not only is erraticity resolved, the number of 15 possible target language pairs with significant differences reduces from 8 with the UTF-8 byte representation to **0** (Table 1 under "ARRU$_t$"), indicating that we eliminated disparity with this optimization heuristic. Since our heuristic is a lossless and reversible transform, it shows that **a complexity that is intrinsic and necessary in language**[7] **does not exist** in computing, however diverse they may be, as our 6 are, from the conventional linguistic typological, phylogenetic, historical, or geographical perspectives.

**Word level**    The main difference between word and character/byte models is length not being a top contributing factor correlating with performance, but instead $|V|$ is. This is understandable as word segmentation neutralizes sequence lengths. To remedy the OOV problem, we use BPE, which learns a fixed vocabulary of variable-length character sequences (on word level, as it presupposes word segmentation) from the training data. It is more fine-grained than word segmentation and is known to better model subword units for morphologically complex languages. We use the same vocabulary of 30,000 as specified in Junczys-Dowmunt et al. (2016). This reduced our averaged OOV token rate by 89-100% across the 5 sizes. The number of language pairs with significant differences reduced to 7 from 8 for word models. While BPEs are still not as effective as our character/byte variants, their results show how **finer-grained modeling contributes positively to closing the disparity gap**.

## 5    META RESULTS, ANALYSES, AND DISCUSSION

**Performance disparity**    Table 1 lists the number of language pairs with significant differences under the representations studied. Since it is **possible** for our character and byte models to effect no performance disparity for the same languages on the same data, a complexity intrinsic to language does not exist. In fact, the customary expectation that languages ought to perform differently is created through our word segmentation practice. Furthermore, the order of AR/RU > ES/FR > EN/ZH (Figure 1c) resembles the idea of morphological complexity. Considering there are character-internal

---

[6]which are sequences with an implicit/explicit pattern made up of consonants and vowels

[7]aside from its statistical properties related to length and vocabulary. To show something is not necessarily true, only 1 counter observation is needed.

Table 1: **Disparity Table** Number of language pairs out of 15 with significant differences, with respective p-values. $ARRU_t$ refers to AR & RU being optimized only on the target side; whereas $ARRU_{s,t}$ denotes optimization on both source and target sides (relevant for directions AR-RU and RU-AR).

| p-value | CHAR src | CHAR trg | Pinyin src | Pinyin trg | Wubi src | Wubi trg | BYTE src | BYTE trg | $ARRU_t$ src | $ARRU_t$ trg | $ARRU_{s,t}$ src | $ARRU_{s,t}$ trg | WORD src | WORD trg | BPE src | BPE trg |
|---|---|---|---|---|---|---|---|---|---|---|---|---|---|---|---|---|
| 0.05 | 0 | 7 | 0 | 4 | 0 | 8 | 0 | 9 | 0 | 4 | 0 | 4 | 0 | 11 | 0 | 10 |
| 0.01 | 0 | 5 | 0 | 2 | 0 | 6 | 0 | 8 | 0 | 3 | 0 | 4 | 0 | 8 | 0 | 8 |
| ☞ 0.001 | 0 | 3 | **0** | **0** | 0 | 5 | 0 | 8 | **0** | **0** | 0 | 2 | 0 | 8 | 0 | 7 |

Table 2: Target language pairs with significant differences indicate that the 2 languages are *not* equally/similarly good or equally/similarly bad. 15 (non-directional) language pairs total possible from 30 language directions, p=0.001.

| $LANG_{trg}$ PAIR | CHAR | Pinyin | Wubi | BYTE | $ARRU_t$ | $ARRU_{s,t}$ | WORD | BPE |
|---|---|---|---|---|---|---|---|---|
| AR-EN |  |  |  | X |  |  | X | X |
| AR-ES |  |  |  |  |  |  |  |  |
| EN-ES |  |  |  |  |  |  | X |  |
| AR-FR |  |  |  | X |  |  |  |  |
| EN-FR |  |  |  |  |  |  | X | X |
| ES-FR |  |  |  |  |  |  |  |  |
| AR-RU |  |  |  | X |  |  |  |  |
| EN-RU |  |  |  | X |  | X | X | X |
| ES-RU |  |  |  | X |  |  |  |  |
| FR-RU |  |  |  | X |  |  |  |  |
| AR-ZH | X |  | X | X |  |  | X | X |
| EN-ZH | X |  | X |  |  |  |  |  |
| ES-ZH |  |  | X |  |  |  | X | X |
| FR-ZH | X |  | X |  |  |  | X | X |
| RU-ZH |  |  | X | X |  | X | X | X |

meaningful units in languages with logographic script such as ZH (cf. Zhang & Komachi (2018)) that are rarely captured, studied, or referred to as "morphemes", this goes to show that linguistic morphology, along with its complexity, as it is practiced today[8] and that which has occurred in the NLP discourse thus far, has only been relevant on the "word" level, conceptually constrained by unstandardizable units such as "words" (and "sentences"). The definition of word, however, has been recognized as problematic for a very long time in the language sciences (cf. Footnote 5).

While the lack of significant differences between pairs of source languages would signify neutralization of source language instances, it does not mean that source languages have no effect on the target. For our byte solutions with code pages, we experimented also with source side optimization in the directions that involve AR/RU as source. This affected the distribution of the disparity results for that representation — with 2 pairs being significantly different (see Table 1 under "$ARRU_{s,t}$"). We defer further investigation on the nature of source language neutralization to future work.

Target language pairs with significant differences are summarized in Table 2. We show that morphological complexity can be empirically eliminated in this one-setting-for-all configuration with a 6-layer network, no hyperparameter tuning, and a maximum line length of 300 characters (and its corresponding equivalence in other representations) as constrained by our hardware and compute time listed in App. A and current data availability. A more analytical solution can be obtained through data statistics (see App. E). A conceptual solution lies in the definition of "words" and morphology.

**Sample-wise Double Descent (DD)** Sample-wise non-monotonicity/DD (Nakkiran et al., 2020) denotes a degradation followed by an improvement in performance with increasing data size. We notice word models and character models with $ZH_{trg}$, i.e. models with high target $|V|$, are prone

---

[8]But there are no reasons why we cannot adopt a statistical science of language in finer granularities beyond/without "words", with standardized units (characters/bytes) and/or continuous representations. Resources, e.g. quality parallel data or contrast sets, can serve both data science and ML interpretation and evaluation well.

to exhibit a spike at $10^3$. A common pattern for these is the ratio of target training token count to number of parameters falls into $O(10^{-4})$ for $10^2$ lines, $O(10^{-3})$ at $10^3$, $O(10^{-2})$ at $10^4$, and $O(10^{-1})$ for $10^5$ lines and so on. But for more atomic units such as alphabetic (not logographic) characters (may it be Latin, Cyrillic, or Abjad) and for bytes, this progression instead begins at $O(10^{-3})$ at $10^2$ lines. Instead of considering this spike of $10^3$ as irregular, we may instead want to think of this learning curve as shifted by 1 order of magnitude to the right for characters and bytes and/or the performance at $10^2$ lines for words and ZH-characters due to being overparameterized and hence abnormal. This would also fit in with the findings by Belkin et al. (2019) and Nakkiran et al. (2020) attributing DD to overparameterization. While almost all work attribute DD to algorithmic reasons, findings from Chen et al. (2020) corroborate our observation and confirms that DD arises due to "the interaction between the properties of the data and the inductive biases of learning algorithms". Other related work on the DD phenomenon and its development can also be found in their work.

**Erraticity** We observe another type of sample-wise non-monotonicity, one that signals irregular and unstable performance across data sizes and runs. Within one run, erraticity can be observed directly as changes in direction on the y-axis. Across runs, large variance can be observed, even with the same dataset. Erraticity can also be observed indirectly through a negative correlation between data size and performance. Much work on length bias in NMT have focused on solutions related to search, e.g. Murray & Chiang (2018). Our experiments show that a kind of length bias can surface already with CLMing, without generation taking place.

**Additional related work** That basic data statistics are the driver of success in performance in multilingual modeling has so far only been explicitly argued for in Mielke et al. (2019). We go beyond their work in monolingual LMs to study CLMs and evaluate also in relation to data size, representational granularity, and quantitative and qualitative fairness. To the best of our knowledge, there has been no prior work on demonstrating the neutralization of source language instances through statistical comparisons, a numerical analysis on DD for sequence-to-sequence models, the meta phenomenon of a sample-wise non-monotonicity (erraticity) being related to length.

## 6 CONCLUSION

**Summary** We investigate whether the performance disparity between languages which have been traditionally considered morphologically rich (AR and RU) and poor (ZH) in the 6-layer Transformer CLM due to morphological complexity is justified and find that it is not. Performance disparity can be explained by data statistics and in the context of computing, it can be eliminated by optimization on length and $|V|$ through character/byte representations. In fact, morphological complexity is not a necessary concept in computing because "word" is not a necessary concept in computing, unless we make it so through word segmentation. A morphological complexity hierarchy can result simply through word segmentation. Furthermore, there are many possible interpretations to "words" for humans and since morphology is defined with the concept of "word", there is no stable ground for assessing this complexity. Representational units of finer granularity were shown to help eliminate performance disparity though at the cost of longer sequence length, which can have a negative impact on robustness. In addition, we found all word models and character models with $ZH_{trg}$ to behave similarly in their being prone to exhibit a peak (as sample-wise DD) around $10^3$ lines in our setting.

**Outlook** ML has enabled greater diversity in NLP (Joshi et al., 2020). Fairness, in the elimination of disparity, does not require big data. This paper made a pioneering attempt to bridge research in NNs/DL, language sciences, and language engineering through a data-centric perspective. Multilinguality is real and relevant in computing not due to canonical, structural linguistic concepts such as morphology or "words" in our minds, but rather standards related to internationalization and localization, such as character encoding — something which has thus far been sorely overlooked in our discourse and curricula. We also believe that a more fine-grained statistical data science can well complement algorithmic analyses with a view that is more empirically robust (i.e. experimentally verifiable) and more relevant to machine processing, contributing to a more generalizable and interpretable pool of knowledge for ML/NNs/DL. A more comprehensive study can lead us not only to new scientific frontiers, but also to better designs and evaluation, benefitting the development of a more general, diverse and inclusive AI.

ETHICS STATEMENT: FAIRNESS CONCERNS FOR MULTILINGUALITY

**Clearer nomenclature**   If/When the intent is *not* to explicitly model linguistic morphology in computing, one can simply describe languages and their statistical profiles with respect to their representational granularity in characters or bytes (which are and/or can be exhaustively standardized in computing), or refer to sequences as longer/shorter or having a higher/lower vocabulary size when comparing them with each other, rather than "richer"/"poorer" based on concepts (e.g. "words", "sentences") that can be ambiguous, contested, and inaccessible to many.

**Accessibility**   Language communities who are unfamiliar with languages similar to dominant languages or those who are reluctant to conform to one structurally similar interpretation should not have to feel inadequate in processing their own language "from scratch", if they so choose. As technologists, we can help take equitable measures to make fairer data representations and infrastructure available. A "word"-free view of language preempts linguistic/cultural hegemony and such an interpretation would also help make the analyses of language data more objective and clearer.

**Scarcity of quality and/or multiway data for science, evaluation, and documentation**   With the rise of multilingual models comes an alleged decrease in reliance on parallel corpora for MT. But data, esp. high-fidelity/quality[9] textual and multimodal *multiway* parallel data, play not only an important role in scientific research, but also one for historical/cultural documentation. And as this paper shows, they can also serve as evaluation data for ML models for better understanding and interpretation. As challenge sets, data for machine processing would need to be statistically diverse and challenging. Parallel data from previous years have often come in the form of bitexts (2-way parallel text data), usually "word"-tokenized where real length information has been compromised. (The Bible data from Mayer & Cysouw (2014) came with another confound — the ZH numeral '一' ("one") is recognized as a dash (punctuation) and hence tokenized with surrounding whitespaces.) At the time of our present study (2019-present), the UN Parallel Corpus was the only unperturbed, fully multiway parallel data sufficient for reliable evaluation for our size range. Data for science, evaluation, and documentation require long-term, stable platform(s) and support. There are many forms of data science. But in terms of having a sustainable practice to collect and curate good data, and exercising enough of a science with that data so to improve collective intelligence and mutual understanding, instead of looking at data from an utilitarian point of view only for consumer application purposes, there seems to be room for improvement still. We hope our work could help effect a positive change in this direction.[10]

ACKNOWLEDGMENTS

The author thanks all anonymous reviewers and PCs as well as the following individuals for their informative comments on previous version(s) of this paper (names in alphabetic order by last name): Benjamin Börschinger, Jordan Boyd-Graber, Christian Buck, Kyunghyun Cho, Kenneth Church, Miryam de Lhoneux, Rotem Dror, San Duanmu, Chris Dyer, Roman Flury, John Goldsmith, Yannic Kilcher, Sandra Kübler, Thomas McColgan, Jason Naradowsky, Ryokan Ri, Anna Rogers, Mark Rowan, Nikunj Saunshi, Rico Sennrich, Ekaterina Vylomova, Jason Weston, and Kie Zuraw.

We also thank Timothy Baldwin, Kevin Duh, Daniela Gerz, Kenneth Heafield, Felix Hieber, Marcin Junczys-Dowmunt, Gal Kaplun, Dan Klein, Sabrina Mielke, Mathias Müller, Nakkiran Preetum, and Annette Rios for their kind responses in correspondence concerning related work and software. We thank Ismail Moukadiri for his help with the Arabic language.

Some initial experiments were supported by the computing resources at the Department of Informatics and Computational Linguistics at UZH and the Institute of Neuroinformatics at UZH and ETH Zurich (for which we especially thank Richard H.R. Hahnloser, Nikola I. Nikolov, Yuhuang Hu, and Pawel Pyk).

---

[9]Note that "high-quality" does not necessarily mean "clean(ed)" data — it would depend on the situation/task.
[10]Please see further discussion in Wan (2022).

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

# APPENDICES

## A  DATA SELECTION AND PREPROCESSING DETAILS

The UN Parallel Corpus v1.0 (Ziemski et al., 2016) consists of manually translated UN documents from 1990 to 2014 in the 6 official UN languages. Therein is a subcorpus that is fully aligned by line, comprising the 6-way parallel corpus we use. We tried to have as little preprocessing or filtering as necessary to eliminate possible confounds. But as the initial runs of our experiment failed due to insufficient memory on a single GPU with 12 GB VRAM[11], we filtered out lines with more than 300 characters in any language in lockstep with one another for all the 6 languages such that the subcorpora would remain parallel, thereby keeping the material of each language semantically equivalent to one another. 8,944,859 lines for each language were retained as our training data which cover up to the 75$^{th}$ percentile in line length for all 6 languages. In order to monitor the effect of data size, we made subcorpora of each language in 5 sizes by `heading` the first $10^2, 10^3, 10^4, 10^5, 10^6$ lines[12]. We refer to this as dataset A. In addition, to better understand and verify the consistency of the phenomena observed, we made 2 supplemental datasets by shuffling the 8,944,859 lines two different times randomly and `heading` the number of lines in our 5 sizes for each language, again in lockstep with one another (datasets B and C).

The systematic training regime that we gave to our language directions is identical for all and we controlled also for seeds. For each of the 3 primary representations — character, byte, and word, we performed:

- 5 runs in 5 sizes ($10^2 - 10^6$ lines): A0 (seed=13), B0 (13), C0 (9948), A1 (9948), A2 (265), and
- 7 more runs in 4 sizes ($10^2 - 10^5$ lines): A3 (777), A4 (42), A5 (340589), A6 (1000), A7 (83146), B1 (9948), & C1 (13).

Figure 1 shows results from all 12 runs in all sizes for the primary representations.

For the alternate/secondary representations, we performed 3 runs each in 5 sizes ($10^2$-$10^6$ lines) (A0, B0, & C0). Due to limitations in computing resources, we were not able to perform as many runs as the primary representations. But important for our statistical comparisons is that we evaluate based on an equal number of runs and on the same data for all candidates. Tables 1 & 2 are the results.

For each run and each size, there are 30 pairwise directions (i.e. 1 source language to 1 target language, e.g. AR-EN for Arabic to English) that result from the 6 languages. We trained all 150 jobs (30 directions x 5 sizes) for each run and representation using the Transformer model (Vaswani et al., 2017) as supported by the SOCKEYE Toolkit (Hieber et al., 2018) (version 1.18.85), based on MXNet (Chen et al., 2015). A detailed description of the architecture of the Transformer can be found in (Vaswani et al., 2017). The same set of hyperparameters applies to all and its values are listed in Appendix B.

For character modeling, we used a dummy symbol to denote each whitespace. For byte, we turned each UTF-8-encoded character into a byte string in decimal value, such that each token is a number between 0 and 255, inclusive. For word, we followed (Junczys-Dowmunt et al., 2016) and used the Moses tokenizer (Koehn et al., 2007) as is standard in NMT practice when word tokenization is applied and Jieba[13] for segmentation in ZH.

Pinyin is a romanization of ZH characters based on their pronunciations and Wubi an input algorithm that decomposes character-internal information into stroke shape and ordering and matches these to 5 classes of radicals (Lunde, 2008). For Pinyin, we used the implementation from `https://github.com/lxyu/pinyin` in the numerical format such that each character/syllable is

---

[11]GPUs used for experiments in this paper range from a NVIDIA TITAN RTX (24 GB), NVIDIA GeForce RTX 2080 Ti (11 GB), a GTX Titan X (12 GB), to a GTX 1080 (8 GB). All jobs were run on a single GPU setting. Some word-level experiments involving AR$_{trg}$ or RU$_{trg}$ at $10^6$ had to be run on a CPU as 24 GB VRAM were not sufficient. Models with higher maximum sequence lengths (e.g. byte models) were trained with 24 GB VRAM. Difference in equipment does not necessarily lead to degradation/improvement in scores.

[12]The terms "line" and "sentence" have been used interchangeably in the NLP literature. We use "line" to denote a sequence that ends with a newline character and "sentence" as one with an ending punctuation. Most parallel corpora, such as ours, are aligned by line, as a line may be part of a sentence or without an ending punctuation (e.g. a header/title). Using a standardized unit such as "line" would also be a fairer measure to linguae/scriptiones continuae (languages/scripts with no explicit punctuation).

[13]`https://github.com/fxsjy/jieba`

followed by a single digit indicating its lexical tone in Mandarin. For Wubi, we used the dictionary from the implementation from `https://github.com/arcsecw/wubi`.

We have implemented all representations such that they would be reversible even when the sequence contains code-mixing. Additional code will be made available at `https://github.com/dadasci`.

We used the official dev set as provided in (Ziemski et al., 2016), 3,077 lines per language remained from 4,000 after filtering line length to 300 characters. Data statistics is provided in Appendix E for reference.

**Notes on training time**    Each run of 30 directions in 5 sizes took approximately 8-12 days for character and byte models. Byte models generally took longer — hence training time is positively correlated with length (concurring with observations by Cherry et al. (2018) as they compared character with BPE models). A maximum length of 300 characters entails a maximum length of *at least* 300 bytes in UTF-8. Each run of word models (30 directions, 5 sizes) took about 6 days (excluding the training of some 7-9 directions out of 30 per run involving $AR_{trg}$ or $RU_{trg}$ at $10^6$ on word level which took about 12-18 hours *each direction* to train on a CPU as these required more space and would run out of memory (OOM) on our GPUs otherwise). These figures do not include the additional probing experiments described in § 4.

## B    HYPERPARAMETER SETTING

- encoder transformer;
- decoder transformer;
- num-layers 6:6;
- num-embed 512:512;
- transformer-model-size 512;
- transformer-attention-heads 8;
- transformer-feed-forward-num-hidden 2048;
- transformer-activation-type relu;
- transformer-positional-embedding-type fixed;
- transformer-preprocess d; transformer-postprocess drn;
- transformer-dropout-attention 0.1;
- transformer-dropout-act 0.1;
- transformer-dropout-prepost 0.1;
- batch-size 15;
- batch-type sentence;
- max-num-checkpoint-not-improved 3;
- max-num-epochs 50;
- optimizer adam;
- optimized-metric perplexity;
- optimizer-params epsilon: 0.000000001, beta1: 0.9, beta2: 0.98;
- label-smoothing 0.0;
- learning-rate-reduce-num-not-improved 4;
- learning-rate-reduce-factor 0.001;
- loss-normalization-type valid;
- max-seq-len 300 for character, word, and BPE, 672 for all bytes, 688 for Wubi, 680 for Pinyin;
- checkpoint-frequency/interval 4000.
  (For smaller datasets, the end of 50 epochs is often reached before the first checkpoint. Since SOCKEYE only outputs scores at checkpoints, we adjusted the checkpoint frequency as follows to get a score outputted by the end of 50 epochs: 1000 for 100 lines for all character & byte instances, 400 for 100 lines for word and 500 for 100 lines BPE, 3450 for 1000 lines for word & BPE. For the very few cases that this default does not suffice due to bucketing of similar length sequences, we manually set the checkpoint frequency to the last batch.)

## C  CORRELATION STATISTICS

Best correlating metrics, i.e. the union of top 3 metrics for all representations.
For each representation, the **top 3 metrics** are boldfaced.
All correlations are **highly significant** ($p < 10^{-30}$), except for min source length for WORD ($p \approx 0.0001$) and min target length for WORD ($p \approx 0.3861$).

| Metric | CHAR | Pinyin | Wubi | BYTE | ARRU$_t$ | ARRU$_{s,t}$ | WORD | BPE |
|---|---|---|---|---|---|---|---|---|
| minimum length (target) | **0.84** | **0.85** | **0.86** | **0.60** | **0.84** | **0.84** | −0.02 | 0.65 |
| minimum length (source) | **0.82** | **0.84** | **0.85** | 0.57 | **0.84** | **0.84** | 0.10 | 0.64 |
| number of tokens (source) | −0.78 | −0.81 | −0.82 | **−0.60** | **−0.81** | **−0.81** | −0.59 | **−0.83** |
| TTR (target) | **0.83** | **0.83** | **0.84** | 0.48 | 0.81 | 0.81 | 0.61 | 0.83 |
| $|V|$ (source) | −0.54 | −0.51 | −0.51 | −0.50 | −0.67 | −0.68 | **−0.63** | **−0.86** |
| data size in lines | −0.80 | −0.83 | −0.83 | −0.59 | −0.81 | −0.81 | −0.62 | **−0.86** |
| OOV token rate (target) | 0.69 | 0.66 | 0.66 | 0.47 | 0.67 | 0.68 | **0.66** | 0.62 |
| OOV type rate (target) | 0.70 | 0.71 | 0.72 | 0.47 | 0.69 | 0.70 | **0.65** | 0.62 |
| TTR (source) | 0.67 | 0.71 | 0.71 | **0.60** | 0.81 | 0.81 | 0.56 | 0.82 |

The full list of metrics used for the correlation analysis is:

1. minimum length (source),
2. minimum length (target),
3. maximum length (source),
4. maximum length (target),
5. median length (source),
6. median length (target),
7. mean length (source),
8. mean length (target),
9. length std (source),
10. length std (target),
11. data size in lines,
12. number of parameters,
13. number of types ($|V|$) (source),
14. number of types ($|V|$) (target),
15. number of tokens (source),
16. number of tokens (target),
17. type-token-ratio (TTR) (source),
18. type-token-ratio (TTR) (target),
19. OOV type rate (source),
20. OOV type rate (target),
21. OOV token rate (source),
22. OOV token rate (target),
23. token ratio,
24. target type-to-parameter ratio,
25. target token-to-parameter ratio,
26. distance between the TTRs of source and target = $(1 - \text{TTR}_{src}/\text{TTR}_{trg})^2$,
27. token-to-parameter ratio (i) = (median length source * median length target * num_lines) / num_parameters,
28. token-to-parameter ratio (ii) = (num_source_tokens * num_target_tokens) / num_parameters.

## D  STATISTICAL COMPARISONS

Recall the definition and method for our **Disparity/Inequality** assessment from § 2:

> In the context of our CLMing experiments, we consider there to be "disparity" or "inequality" between languages $l_1$ and $l_2$ if there are significant differences between the performance distributions of these two languages with respect to each representation. Here, by performance we mean the number of bits required to encode the held-out data using a trained CLM. With 30 directions, there are 15 pairs of source languages $(l_{src1}, l_{src2})$ and 15 pairs of target languages $(l_{trg1}, l_{trg2})$ possible. We compare the source languages among each other, and the target languages among each other. Each $l_{src}$ or each $l_{trg}$ consists of scores from all models trained across various sizes and directions. To assess whether the differences are significant, we perform unpaired two-sided significance tests with the null hypothesis that the score distributions for the two languages are not different. Upon testing for normality with the Shapiro-Wilk test (Shapiro & Wilk, 1965; Royston, 1995), we use the parametric unpaired two-sample Welch's t-test (Welch, 1947) (when normal) or the non-parametric unpaired Wilcoxon test (Wilcoxon, 1945) (when not normal) for the comparisons. We use the implementation in R (R Core Team, 2014) for these 3 tests. To account for the multiple comparisons we are performing, we correct all p-values using Bonferroni's correction (Benjamini & Heller, 2008; Dror et al., 2017) and follow Holm's procedure[14] (Holm, 1979; Dror et al., 2017) to identify the pairs of $l_1$ and $l_2$ with significant differences after correction. We report all 3 levels of significance ($\alpha \leq 0.05, 0.01, 0.001$) for a more comprehensive overview. In contrast to Dror et al. (2017), which aimed to compare the performance of different algorithms, we compare languages (in the context of computing).

**To get samples for the statistical comparison results for the Disparity Table (Table 1):**

For each representation, we used 3 runs (A0, B0, C0) in 5 sizes ($10^2$-$10^6$ lines) for each $l_{src}$ and each $l_{trg}$. There are:

6 $l_{src}$ (AR$_{src}$, EN$_{src}$, ES$_{src}$, FR$_{src}$, RU$_{src}$, ZH$_{src}$) and
6 $l_{trg}$ (AR$_{trg}$, EN$_{trg}$, ES$_{trg}$, FR$_{trg}$, RU$_{trg}$, ZH$_{trg}$).

We compare pairwise among the $l_{src}$. The 15 pairs are:

AR$_{src}$-EN$_{src}$, AR$_{src}$-ES$_{src}$, AR$_{src}$-FR$_{src}$, AR$_{src}$-RU$_{src}$, AR$_{src}$-ZH$_{src}$,
EN$_{src}$-ES$_{src}$, EN$_{src}$-FR$_{src}$, EN$_{src}$-RU$_{src}$, EN$_{src}$-ZH$_{src}$,
ES$_{src}$-FR$_{src}$, ES$_{src}$-RU$_{src}$, ES$_{src}$-ZH$_{src}$,
FR$_{src}$-RU$_{src}$, FR$_{src}$-ZH$_{src}$,
RU$_{src}$-ZH$_{src}$.

Likewise 15 pairs among the $l_{trg}$.

**For example**, for the character (primary) representation to compare between AR$_{src}$ and EN$_{src}$, we construct the sample for AR$_{src}$ (`sample_ARsrc`) and the sample for EN$_{src}$ (`sample_ENsrc`) as follows:

Out of the 30 CHAR directions, there are 5 directions involving AR$_{src}$ trained for each run and data size (i.e. the directions: AR-EN, AR-ES, AR-FR, AR-RU, AR-ZH).

For each direction, there are 15 models trained (3 runs x 5 sizes). We take all 75 CHAR models (15 models x 5 directions) involving AR$_{src}$ as `sample_ARsrc`. That's a sample of size 75.

Likewise, for EN$_{src}$ (5 directions: EN-AR, EN-ES, EN-FR, EN-RU, EN-ZH), we also have 75 data points for `sample_ENsrc`. (Likewise also for all 6 $l_{src}$ and all 6 $l_{trg}$.)

For the comparisons, we compare pairwise, i.e. with two samples each time, but with *unpaired* two-sample Welch's t-test (when normal) or the non-parametric *unpaired* Wilcoxon test (when not normal) because `sample_ARsrc` and `sample_ENsrc` have one direction that is not paired: AR-EN and EN-AR. Other directions can be seen as paired, e.g. AR-ES and EN-ES as both having the same $l_{trg}$.

---

[14]using implementation from `https://github.com/rtmdrr/replicability-analysis-NLP`

# E DATA STATISTICS

- Number of types, i.e. vocabulary size ($|V|$). Note that Sockeye adds for its calculation 4 additional types: <pad>, , , <unk>.
- Number of tokens. This excludes the 1 EOS/BOS (end-/beginning-of-sentence) marker added by Sockeye to each line.
- Out-of-vocabulary (OOV) type rate (in %), i.e. the fraction of the types in the dev data that is not covered by the types in the training data.
- OOV token rate (in %), i.e. the fraction of tokens in the dev data that is treated as UNKnowns.
- Type-token-ratio (in %), i.e. the ratio between the number of types and tokens in the data. This is a rough proxy for lexical diversity in that a value of 1 would indicate that no type is ever seen twice, and a value very close to 0 would indicate that very few distinct types account for almost all of the data.
- Line length (excl. EOS/BOS marker): mean±standard deviation, and the 0/25/50/75/100-th percentile.

## Statistics for dataset A

*The detailed numerical table (spanning representations CHAR, BYTE, WORD, BPE across vocabulary sizes, with rows for Number of TYPES, Number of TOKENS, OOV type rate (%), OOV token rate (%), TTR (%), and Mean line length±std 0/25/50/75/100-th for languages AR, EN, ES, FR, RU, ZH and the variants ZH_pinyin, ZH_wubi, AR_cp1256, RU_cp1251).*

## Statistics for dataset B

| Representation | CHAR | | | | | BYTE | | | | | BPE | | | | | WORD | | | | |
| --- | --- | --- | --- | --- | --- | --- | --- | --- | --- | --- | --- | --- | --- | --- | --- | --- | --- | --- | --- | --- |
| Number of lines | 100 | 1,000 | 10,000 | 100,000 | 1,000,000 | 100 | 1,000 | 10,000 | 100,000 | 1,000,000 | 100 | 1,000 | 10,000 | 100,000 | 1,000,000 | 100 | 1,000 | 10,000 | 100,000 | 1,000,000 |

*Number of TYPES, Number of TOKENS, OOV type rate (%), OOV token rate (%), TTR (%), and Mean line length±std sections for representations AR, EN, ES, FR, RU, ZH, ZH_wubi, AR_cp1256, RU_cp1251.*

## Statistics for dataset C

## Statistics for development (dev) set

As a different set of vocabulary is learned from each training dataset and data size, BPE has a distinct dev set for each.

| Representation / Number of lines in trainset | CHAR | BYTE | WORD | BPE_A 100 | 1,000 | 100,000 | 1,000,000 | BPE_B 100 | 1,000 | 100,000 | 1,000,000 | BPE_C 100 | 1,000 | 100,000 | 1,000,000 |
|---|---|---|---|---|---|---|---|---|---|---|---|---|---|---|---|
| **Number of TYPES** | | | | | | | | | | | | | | | |
| AR | 130 | 123 | 13,836 | 708 | 3,232 | 8,994 | 12,430 | 865 | 3,991 | 10,735 | 13,000 | 908 | 4,119 | 12,934 | 13,003 |
| EN | 97 | 102 | 7,199 | 656 | 2,567 | 5,598 | 7,556 | 801 | 3,149 | 6,518 | 7,528 | 823 | 3,245 | 7,524 | 7,573 |
| ES | 111 | 110 | 8,551 | 680 | 2,832 | 6,919 | 8,871 | 833 | 3,436 | 7,528 | 8,566 | 834 | 3,424 | 8,871 | 8,509 |
| FR | 113 | 118 | 8,312 | 744 | 2,821 | 6,905 | 8,466 | 827 | 3,407 | 7,489 | 8,705 | 883 | 3,463 | 7,631 | 8,708 |
| RU | 146 | 144 | 12,819 | 878 | 3,769 | 10,085 | 12,788 | 1,024 | 4,438 | 11,402 | 12,954 | 1,038 | 4,509 | 12,882 | 12,958 |
| ZH | 1,976 | 153 | 7,413 | 3,224 | 4,215 | 6,386 | 7,654 | 3,261 | 4,481 | 6,735 | 7,702 | 3,260 | 4,524 | 7,654 | 7,721 |
| ZH_pinyin | 98 | 130 | | | | | | | | | | | | | |
| ZH_wubi | 120 | 146 | | | | | | | | | | | | | |
| AR_cp1256 | 376,679 | 334,358 | | | | | | | | | | | | | |
| RU_cp1251 | 330,734 | 431,538 | | | | | | | | | | | | | |
| **Number of TOKENS** | | | | | | | | | | | | | | | |
| AR | 334,358 | 605,516 | 61,371 | 167,574 | 115,693 | 83,001 | 70,527 | 149,689 | 97,843 | 73,314 | 68,270 | 149,623 | 97,231 | 68,579 | 68,278 |
| EN | 391,222 | 391,260 | 67,629 | 150,826 | 101,782 | 77,089 | 70,339 | 140,256 | 90,871 | 73,531 | 69,348 | 140,377 | 90,360 | 69,633 | 69,341 |
| ES | 443,358 | 452,810 | 78,087 | 170,133 | 113,083 | 88,634 | 81,341 | 155,687 | 103,788 | 84,914 | 80,387 | 154,746 | 103,172 | 80,579 | 80,371 |
| FR | 438,083 | 452,556 | 78,745 | 166,289 | 114,694 | 88,726 | 81,559 | 156,256 | 104,067 | 84,914 | 80,844 | 153,745 | 104,001 | 80,912 | 80,604 |
| RU | 431,538 | 703,214 | 64,180 | 177,818 | 113,628 | 79,763 | 71,081 | 163,319 | 100,294 | 75,051 | 80,991 | 163,806 | 70,196 | 85,125 | 69,982 |
| ZH | 107,990 | 301,085 | 60,013 | 90,745 | 80,231 | 68,129 | 62,830 | 98,775 | 75,636 | 65,212 | 61,882 | 94,127 | 75,718 | 75,086 | 61,823 |
| | | | | | | | | | | | | | | 61,916 | |
| ZH_pinyin | | | | | | | | | | | | | | | | |
| ZH_wubi | | | | | | | | | | | | | | | | |
| AR_cp1256 | 334,358 | 334,358 | | | | | | | | | | | | | |
| RU_cp1251 | | 431,538 | | | | | | | | | | | | | |
| **TTR (%)** | | | | | | | | | | | | | | | |
| AR | 0.04 | 0.02 | 22.54 | 0.42 | 2.79 | 10.84 | 17.62 | 0.58 | 4.08 | 14.64 | 19.17 | 0.61 | 4.24 | 18.86 | 19.18 |
| EN | 0.03 | 0.03 | 10.64 | 0.42 | 2.52 | 7.78 | 10.74 | 0.57 | 3.47 | 8.86 | 10.93 | 0.59 | 3.59 | 10.81 | 10.92 |
| ES | 0.03 | 0.03 | 10.95 | 0.40 | 2.48 | 7.81 | 10.91 | 0.54 | 3.30 | 8.87 | 11.15 | 0.54 | 3.32 | 11.01 | 11.12 |
| FR | 0.03 | 0.03 | 10.56 | 0.45 | 2.46 | 7.80 | 10.63 | 0.53 | 3.27 | 8.80 | 10.79 | 0.57 | 3.33 | 10.73 | 10.80 |
| RU | 0.03 | 0.02 | 19.97 | 0.49 | 3.32 | 12.64 | 17.99 | 0.63 | 4.42 | 14.73 | 18.51 | 0.63 | 4.51 | 18.35 | 18.52 |
| ZH | 1.83 | 0.05 | 12.35 | 3.33 | 5.25 | 9.37 | 12.19 | 3.48 | 5.92 | 10.33 | 12.46 | 3.46 | 5.97 | 12.36 | 12.49 |
| ZH_pinyin | 0.03 | 0.04 | | | | | | | | | | | | | |
| ZH_wubi | 0.04 | 0.03 | | | | | | | | | | | | | |
| **Mean line length ±std @25/50/75/100-th** | | | | | | | | | | | | | | | |
| AR | 108.66 ± 58.01 3/60/110/153/277 | 196.79 ± 105.85 6/107/199/277/303 | 39.95 ± 10.58 1/12/20/27/58 | 54.46 ± 29.51 1/30/54/76/152 | 37.60 ± 20.51 1/22/37/52/125 | 26.07 ± 14.80 1/16/27/37/95 | 22.92 ± 12.56 1/14/22/32/75 | 48.65 ± 26.35 1/27/49/68/156 | 31.80 ± 17.76 1/18/32/44/110 | 23.83 ± 13.15 1/14/24/33/80 | 22.19 ± 12.07 1/13/22/31/71 | 48.63 ± 26.21 1/28/49/67/145 | 31.60 ± 17.53 1/18/31/44/110 | 22.29 ± 12.14 1/13/22/31/73 | 22.19 ± 12.06 1/13/22/31/72 |
| EN | 127.14 ± 68.64 6/86/130/181/289 | 127.16 ± 68.65 6/86/130/181/299 | 21.98 ± 11.81 1/10/22/31/61 | 50.64 ± 27.44 1/27/51/72/136 | 33.08 ± 18.07 1/19/33/46/110 | 25.05 ± 13.56 1/17/25/35/78 | 22.86 ± 12.27 1/14/23/32/71 | 45.58 ± 24.86 1/26/46/65/129 | 29.53 ± 16.30 1/17/29/41/94 | 23.90 ± 12.95 1/14/24/33/74 | 22.54 ± 12.05 1/13/23/31/65 | 45.62 ± 24.67 1/26/46/66/122 | 29.37 ± 16.13 1/17/29/41/92 | 22.63 ± 12.12 1/13/23/31/65 | 22.54 ± 12.05 1/13/23/31/66 |
| ES | 144.28 ± 77.78 5/77/145/207/300 | 148.06 ± 79.21 5/78/148/211/307 | 25.28 ± 13.56 1/15/25/36/63 | 55.39 ± 30.13 1/30/56/79/141 | 37.04 ± 20.13 1/21/37/52/111 | 28.31 ± 15.53 1/17/28/41/85 | 26.44 ± 14.14 1/15/26/37/73 | 50.49 ± 27.66 1/27/51/72/133 | 33.72 ± 18.48 1/19/34/47/103 | 27.00 ± 14.88 1/15/28/39/75 | 26.11 ± 13.94 1/15/26/37/71 | 50.29 ± 27.27 1/28/51/71/131 | 33.53 ± 18.38 1/19/34/47/95 | 26.19 ± 13.99 1/15/26/37/71 | 26.12 ± 13.94 1/15/26/37/71 |
| FR | 142.37 ± 77.81 4/74/145/205/300 | 147.08 ± 80.30 4/77/150/212/310 | 25.59 ± 13.86 1/14/26/36/86 | 54.04 ± 29.55 1/29/55/78/139 | 37.27 ± 20.65 1/20/38/52/119 | 28.84 ± 15.76 1/16/29/40/89 | 26.51 ± 14.35 1/15/27/37/71 | 50.78 ± 27.79 1/27/52/72/135 | 33.82 ± 18.74 1/18/34/47/108 | 27.66 ± 15.11 1/15/28/39/80 | 26.22 ± 14.16 1/14/27/37/69 | 49.97 ± 27.25 1/27/51/70/131 | 33.83 ± 18.80 1/18/34/47/107 | 26.39 ± 14.21 1/14/27/37/69 | 26.23 ± 14.16 1/14/27/37/69 |
| RU | 140.25 ± 76.21 5/75/141/200/300 | 207.79 ± 141.53 7/136/259/370/369 | 20.86 ± 11.25 1/12/21/29/89 | 57.79 ± 32.12 1/31/58/82/185 | 36.93 ± 20.73 1/21/36/52/185 | 25.02 ± 14.48 1/15/26/36/128 | 23.10 ± 12.71 1/13/23/32/95 | 53.08 ± 29.00 1/29/53/75/185 | 32.59 ± 18.42 1/18/32/46/106 | 24.39 ± 13.59 1/14/24/34/112 | 22.75 ± 12.46 1/13/23/32/93 | 53.24 ± 29.18 2/30/53/76/185 | 32.49 ± 18.49 1/18/32/45/161 | 22.81 ± 12.50 1/13/23/32/90 | 22.74 ± 12.45 1/13/23/32/92 |
| ZH | 35.10 ± 18.48 2/21/35/49/125 | 97.85 ± 52.10 4/57/99/138/288 | 19.50 ± 10.42 1/11/20/27/64 | 31.44 ± 16.96 1/18/31/44/100 | 26.07 ± 14.19 1/15/26/36/93 | 22.14 ± 12.02 1/13/22/31/80 | 20.41 ± 11.05 1/12/20/28/74 | 30.68 ± 16.47 1/17/30/43/104 | 24.58 ± 13.42 1/14/24/34/92 | 21.19 ± 11.49 1/12/21/29/71 | 20.09 ± 10.79 1/12/20/28/65 | 30.79 ± 16.45 1/18/30/43/101 | 24.01 ± 13.49 1/14/24/34/93 | 20.12 ± 10.81 1/12/20/28/64 | 20.09 ± 10.79 1/12/20/28/65 |
| ZH_pinyin | 122.32 ± 65.59 6/67/125/173/353 | 109.66 ± 58.01 3/60/110/153/277 | | | | | | | | | | | | | |
| ZH_wubi | 107.49 ± 56.73 4/60/108/151/294 | | | | | | | | | | | | | | |
| AR_cp1256 | | 140.25 ± 76.21 5/75/141/200/300 | | | | | | | | | | | | | |
| RU_cp1251 | | | | | | | | | | | | | | | | |

# F ENLARGED FIGURES FOR ALL 30 LANGUAGE DIRECTIONS (AGGREGATE RESULTS FROM ALL RUNS)

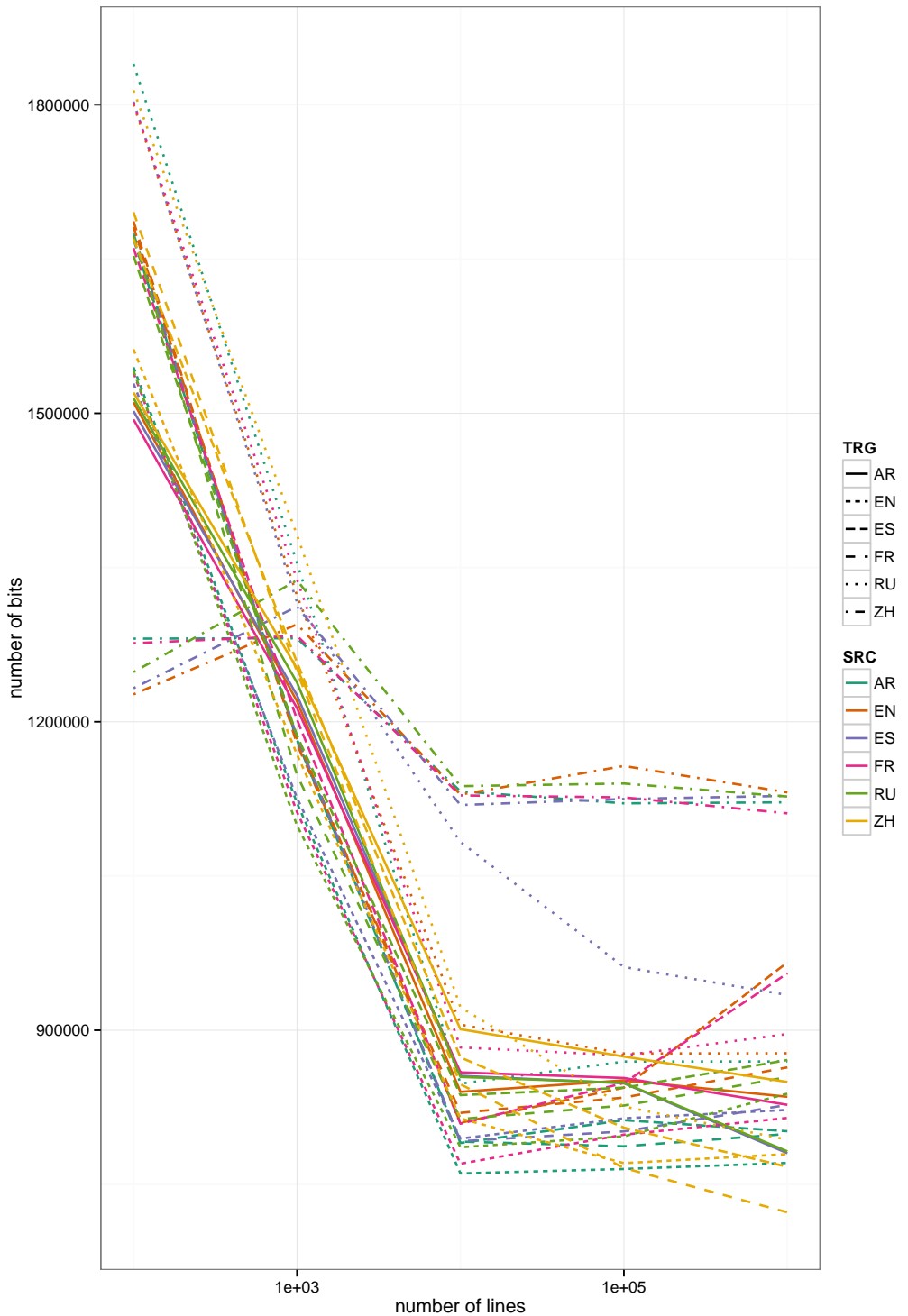

Figure 4: CHAR: character models

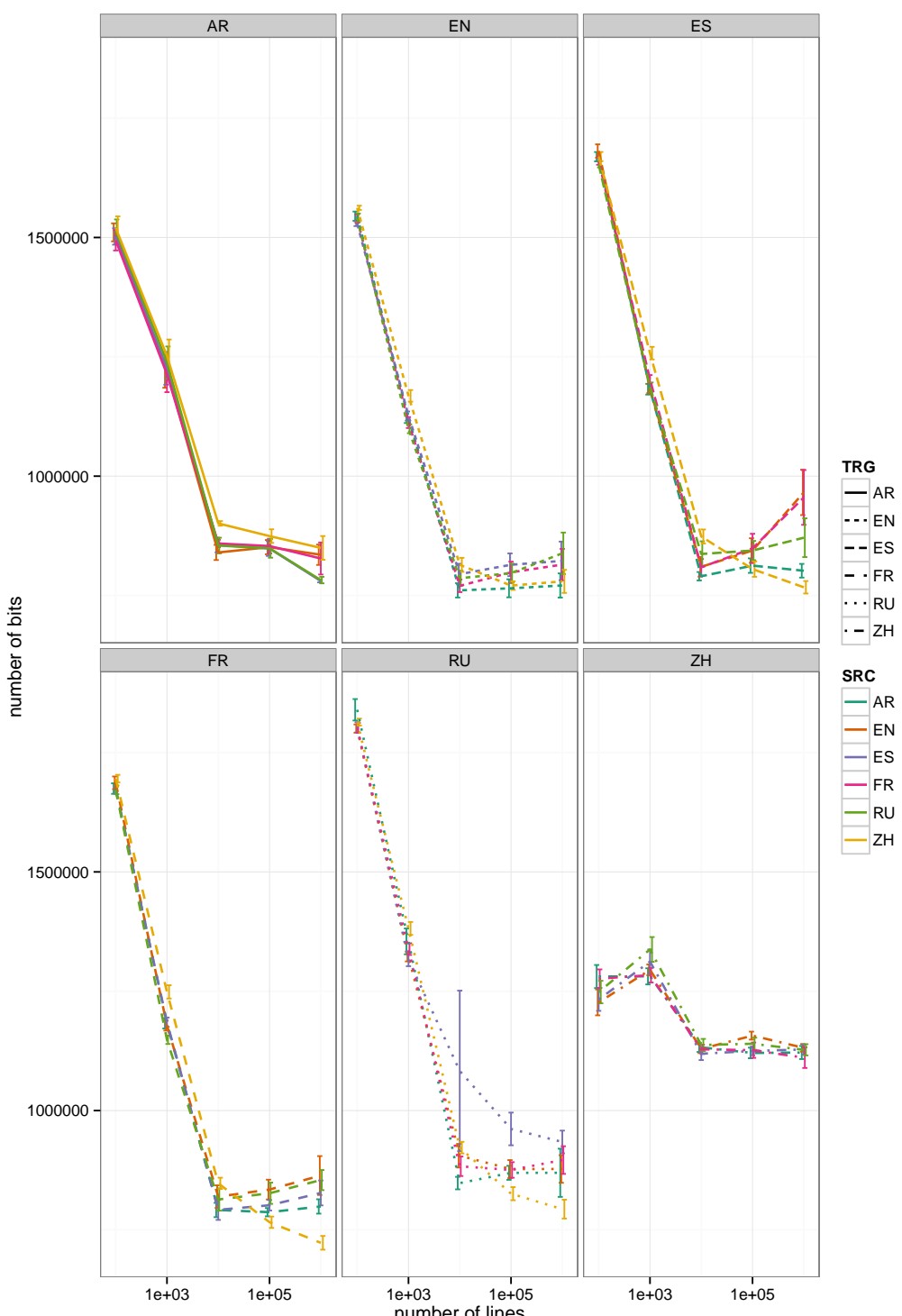

Figure 4: CHAR: character models (target language as facet)

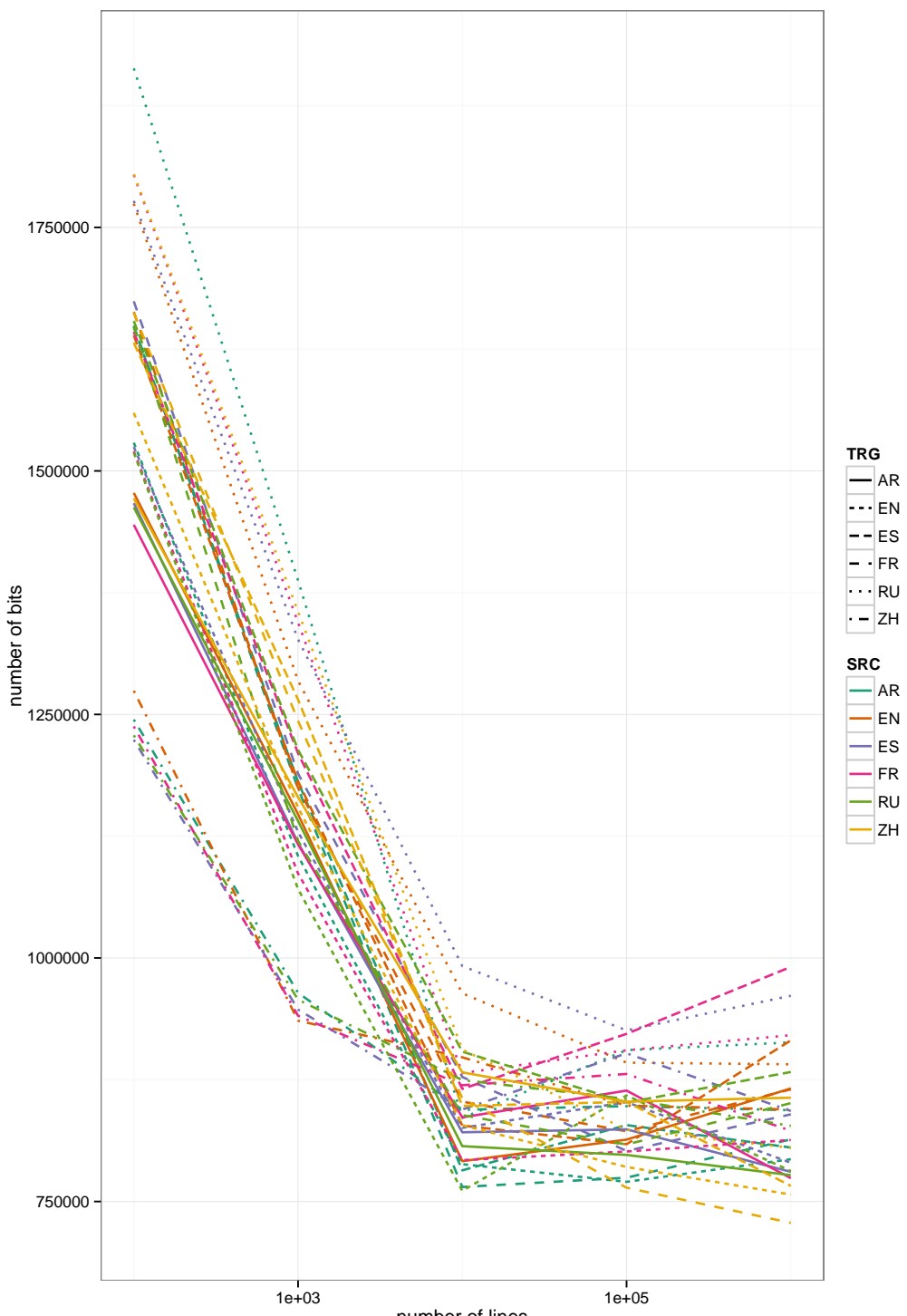

Figure 5: CHAR with Pinyin for $ZH_{trg}$

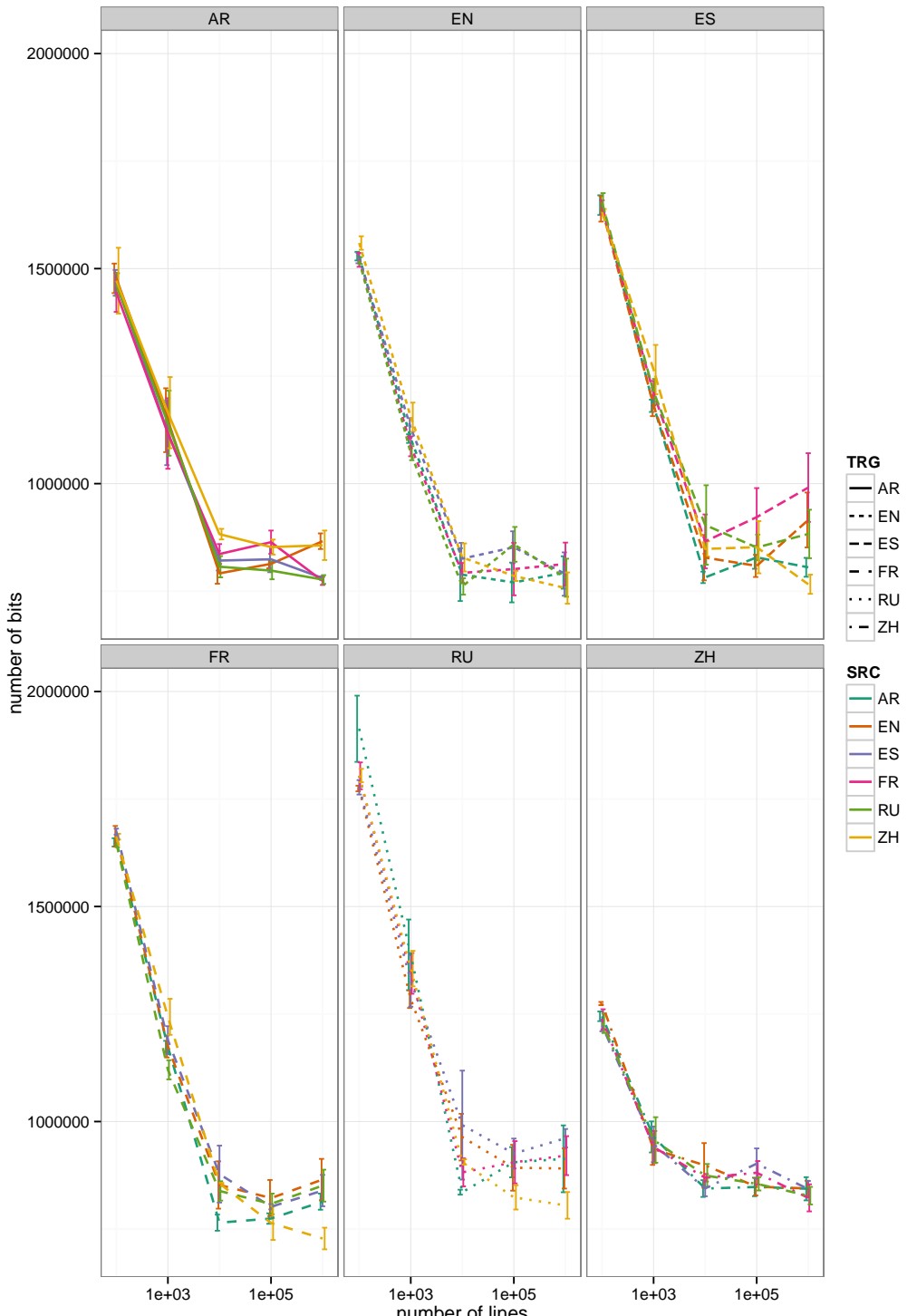

Figure 5: CHAR with Pinyin for $ZH_{trg}$ (target language as facet)

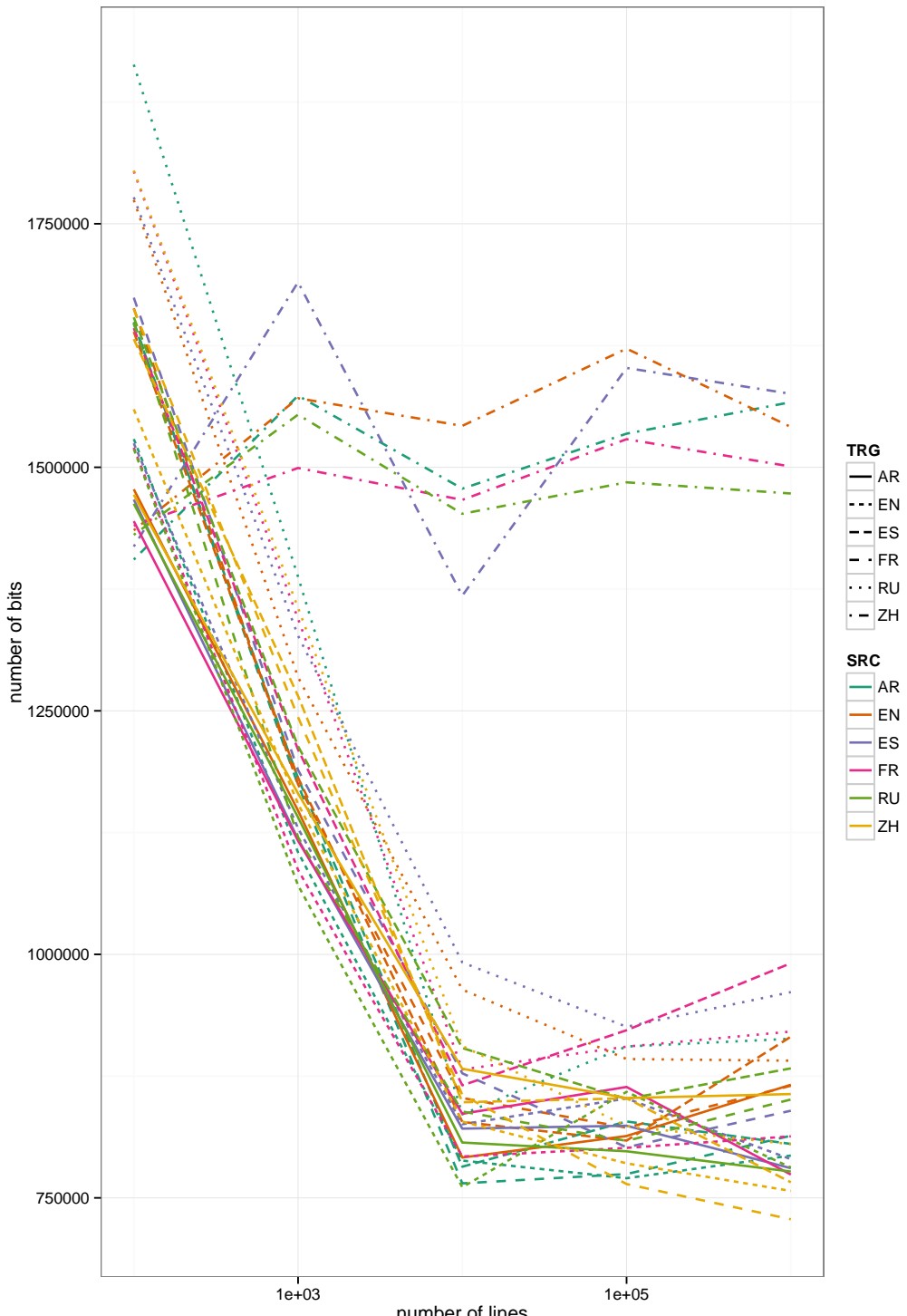

Figure 6: CHAR with Wubi for ZH$_{trg}$

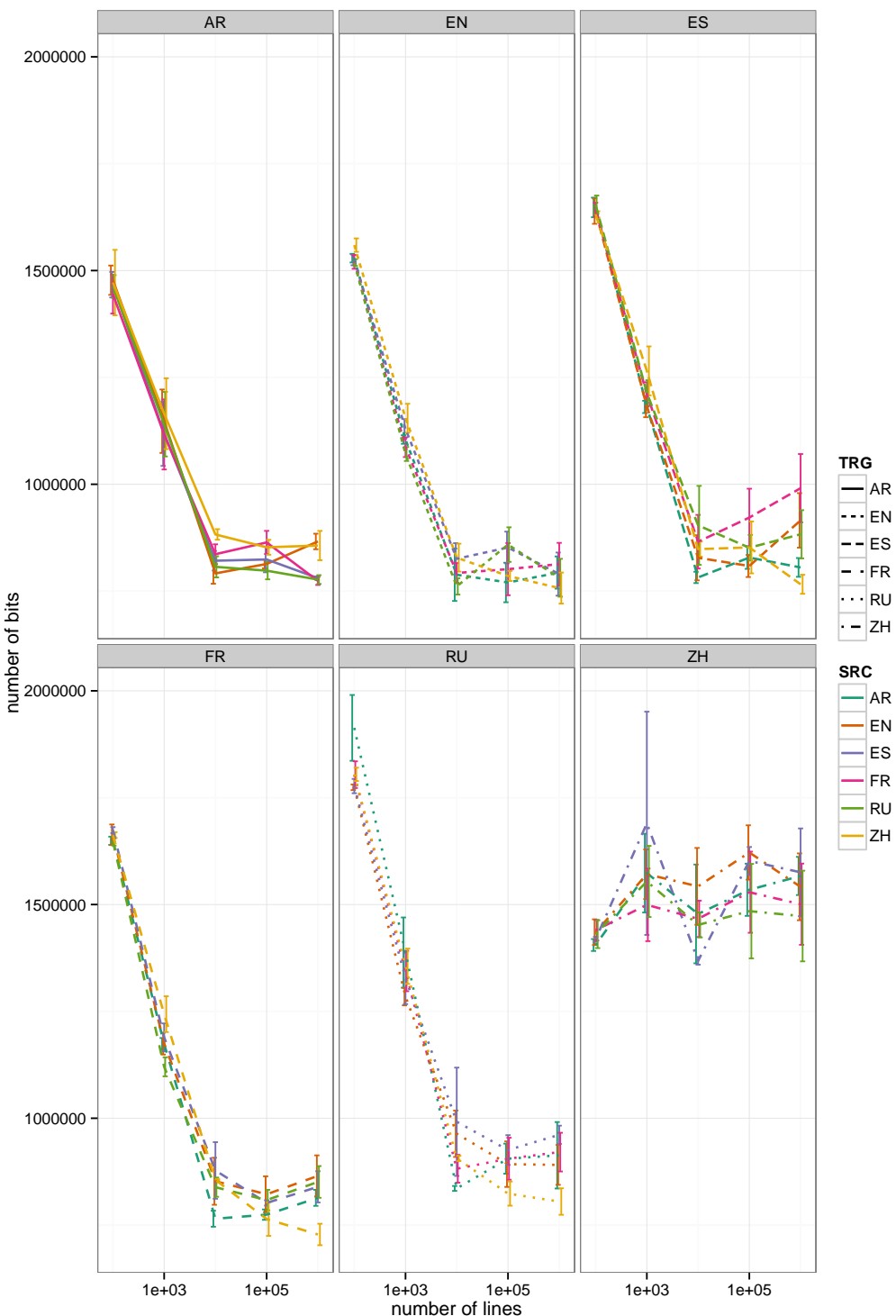

Figure 6: CHAR with Wubi for $ZH_{trg}$ (target language as facet)

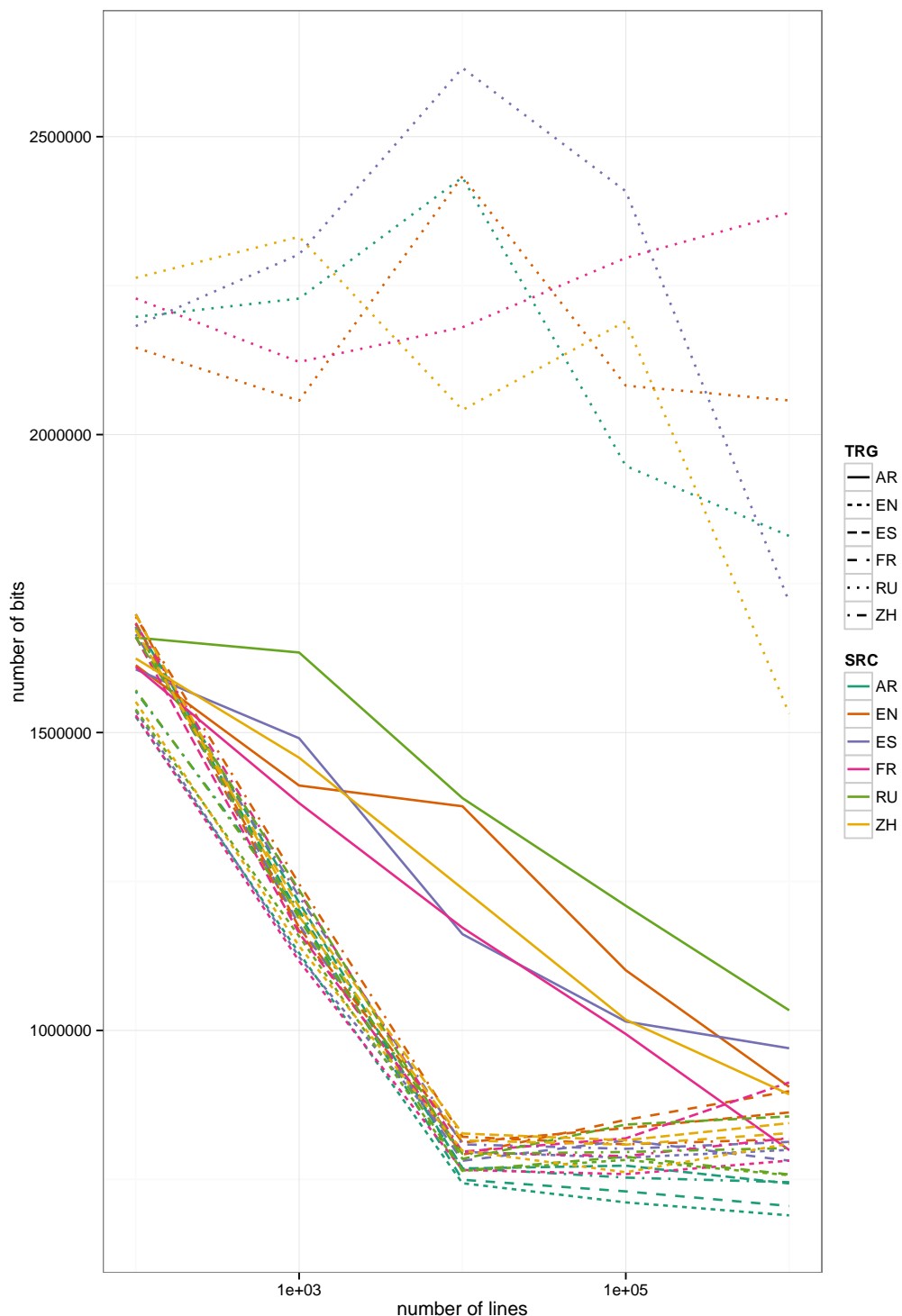

Figure 7: BYTE models with UTF-8 encoding

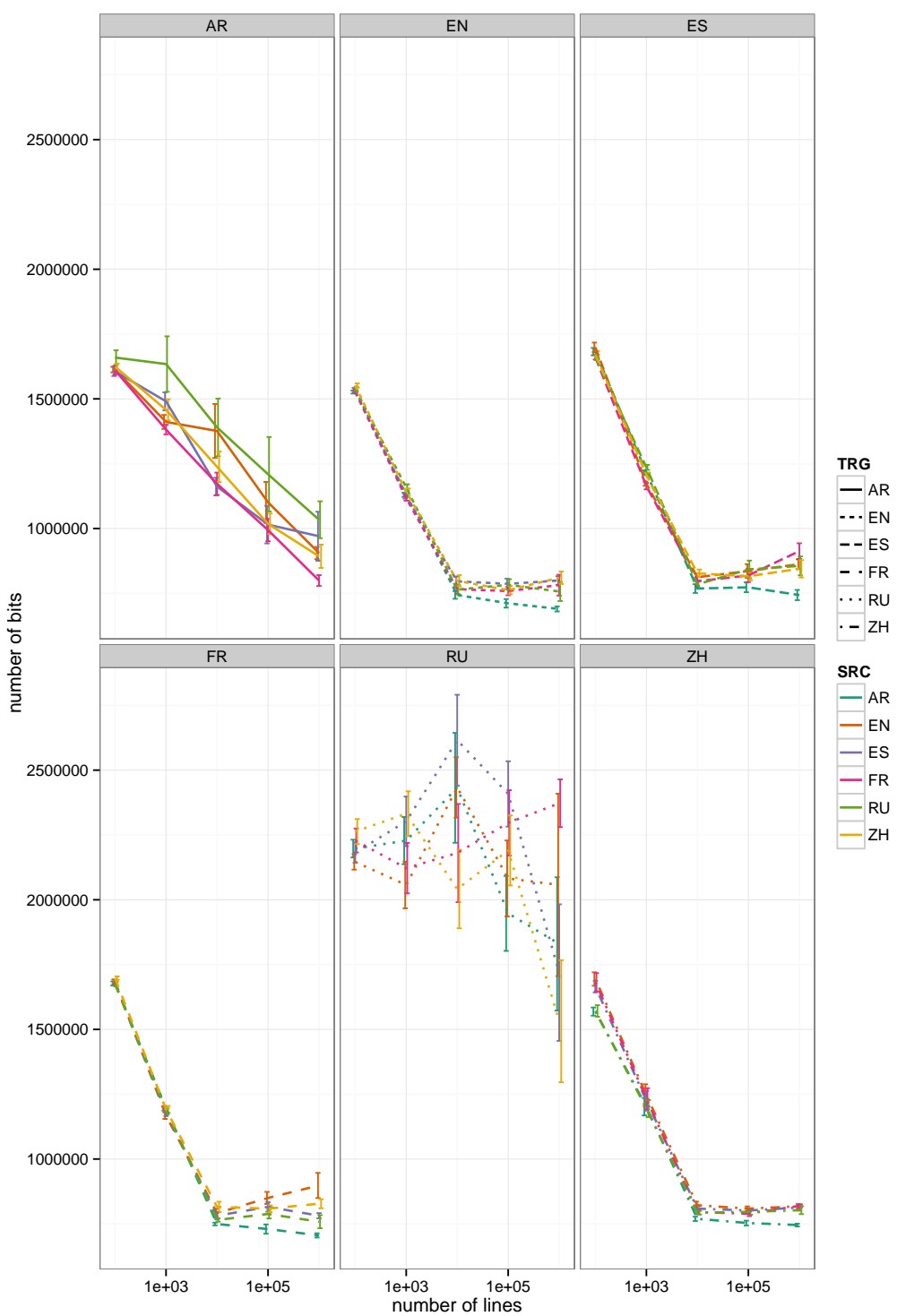

Figure 7: BYTE models with UTF-8 encoding (target language as facet)

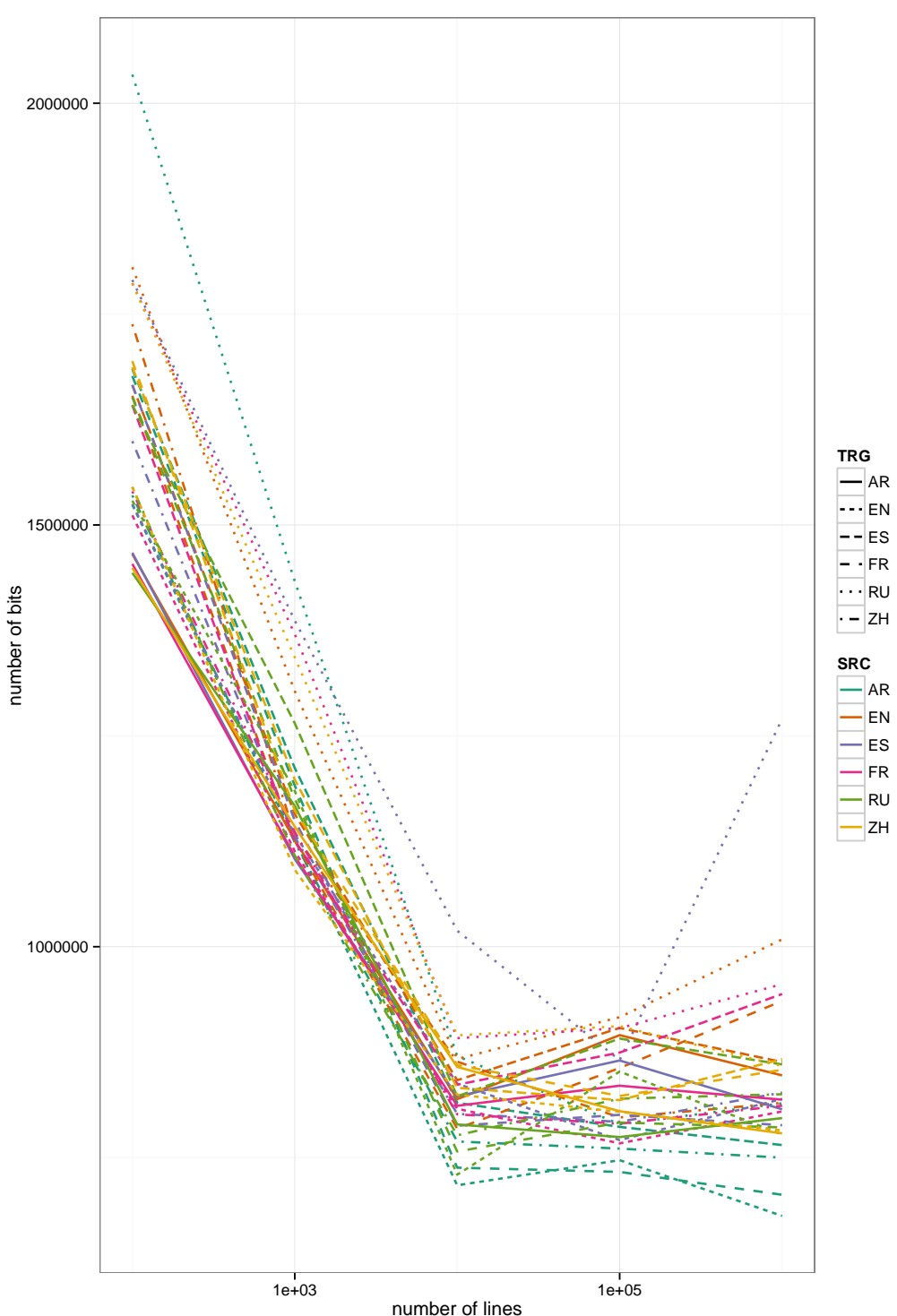

Figure 8: BYTE with $AR_{trg}$ & $RU_{trg}$ optimized with code pages 1256 & 1251 ($ARRU_{trg}$)

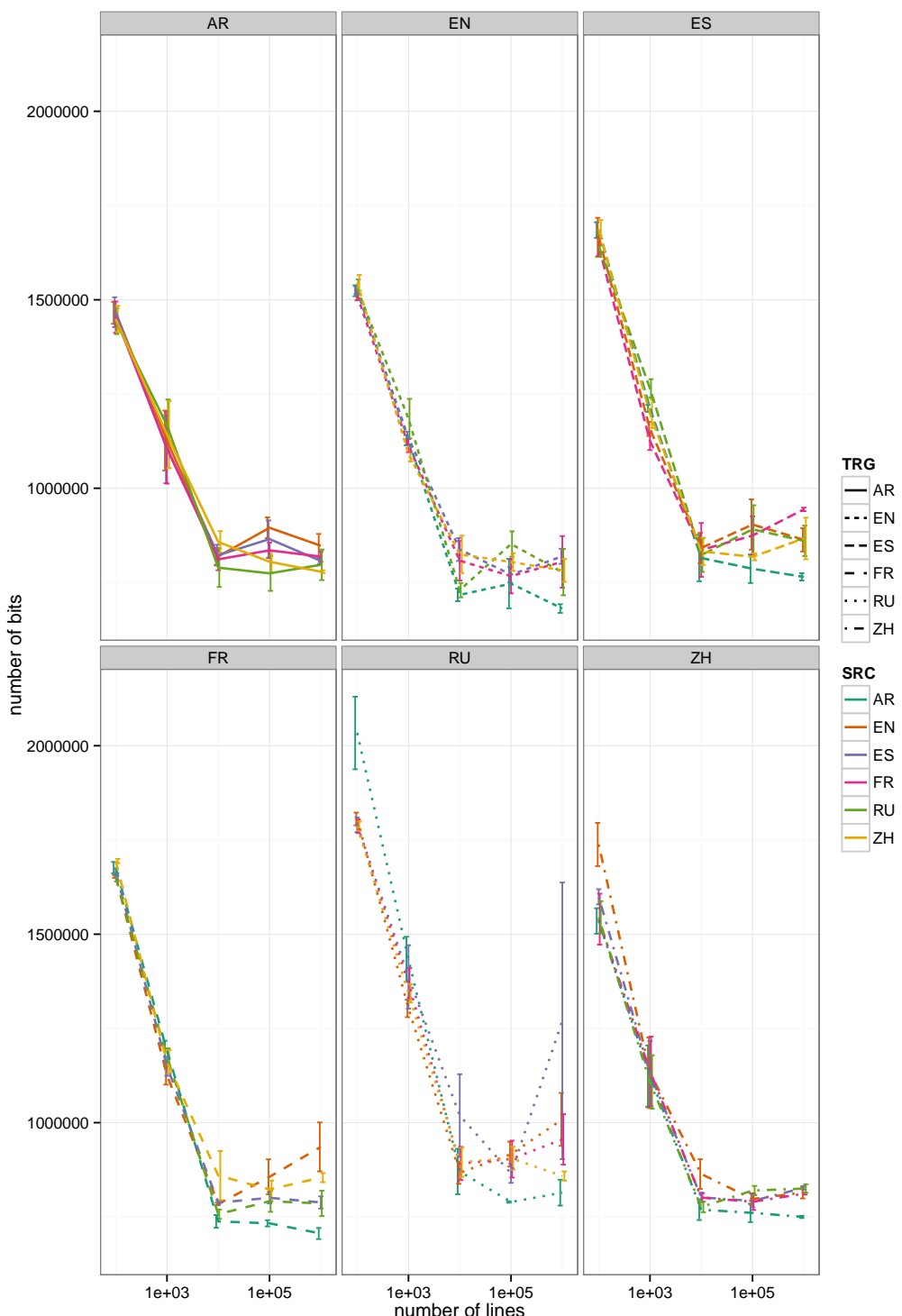

Figure 8: BYTE with $AR_{trg}$ & $RU_{trg}$ optimized with code pages 1256 & 1251 (target language as facet)

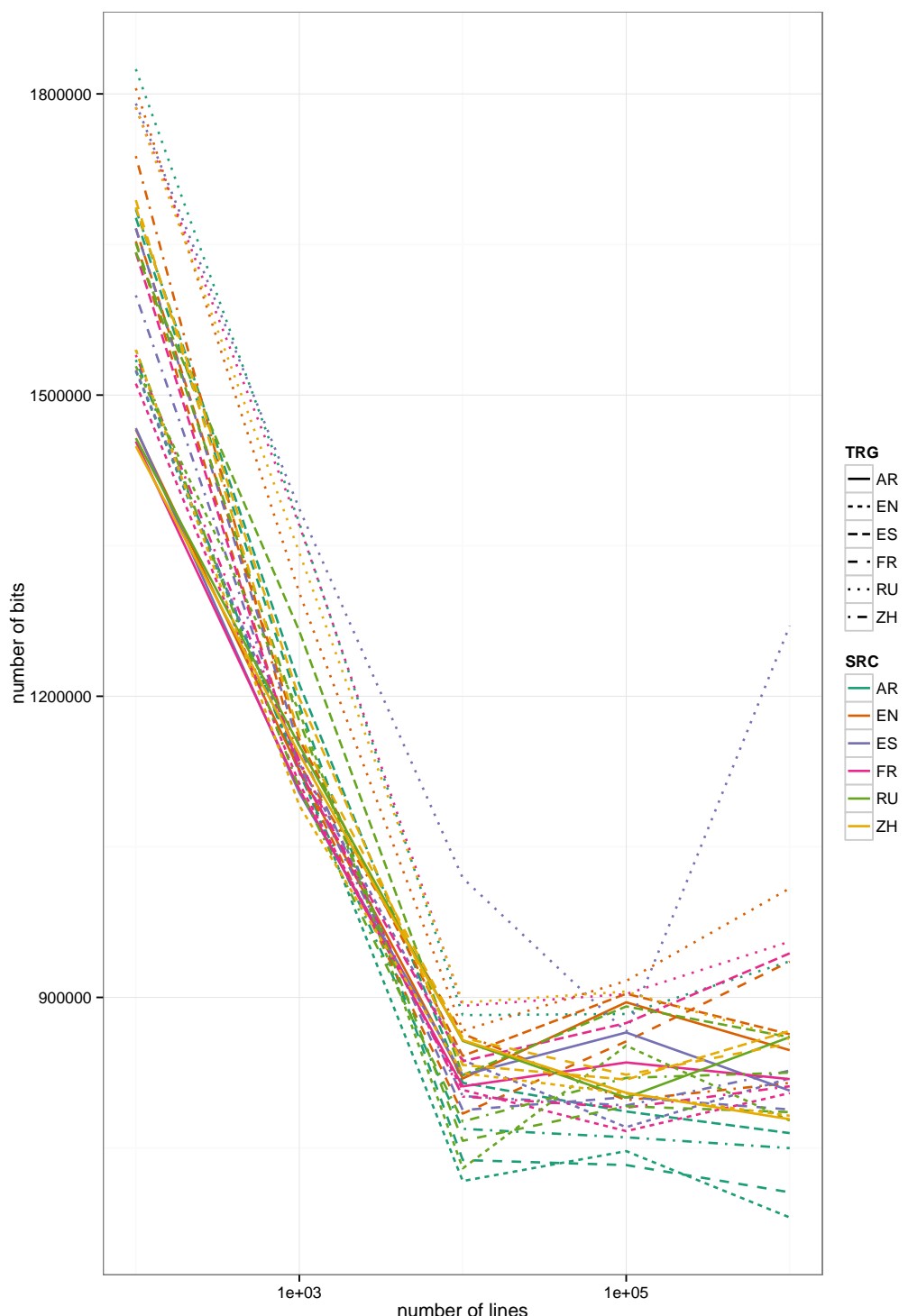

Figure 9: BYTE with directions AR-RU & RU-AR optimized on both source and target sides ($ARRU_{src,trg}$)

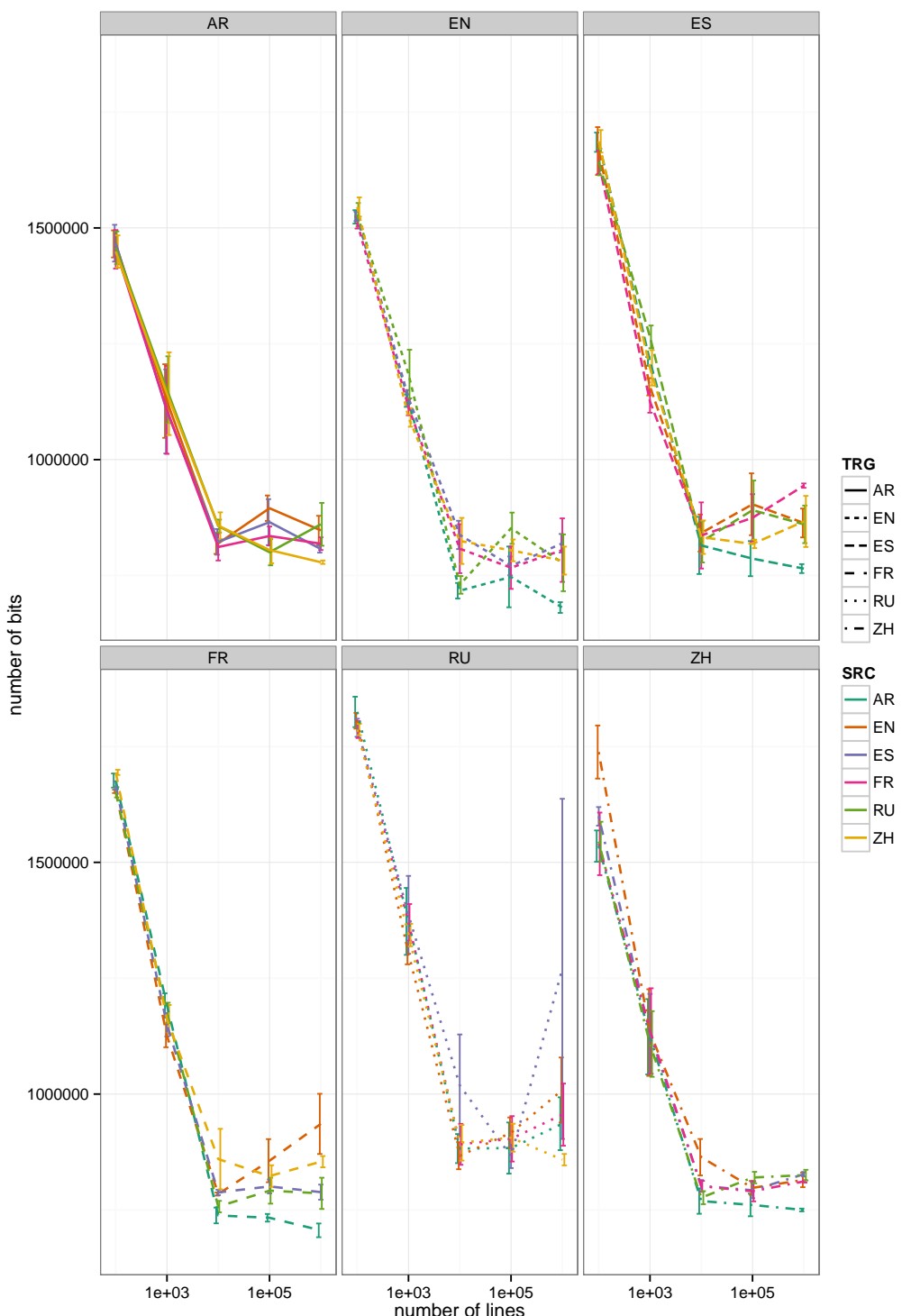

Figure 9: BYTE with directions AR-RU & RU-AR optimized on both source and target sides (target language as facet)

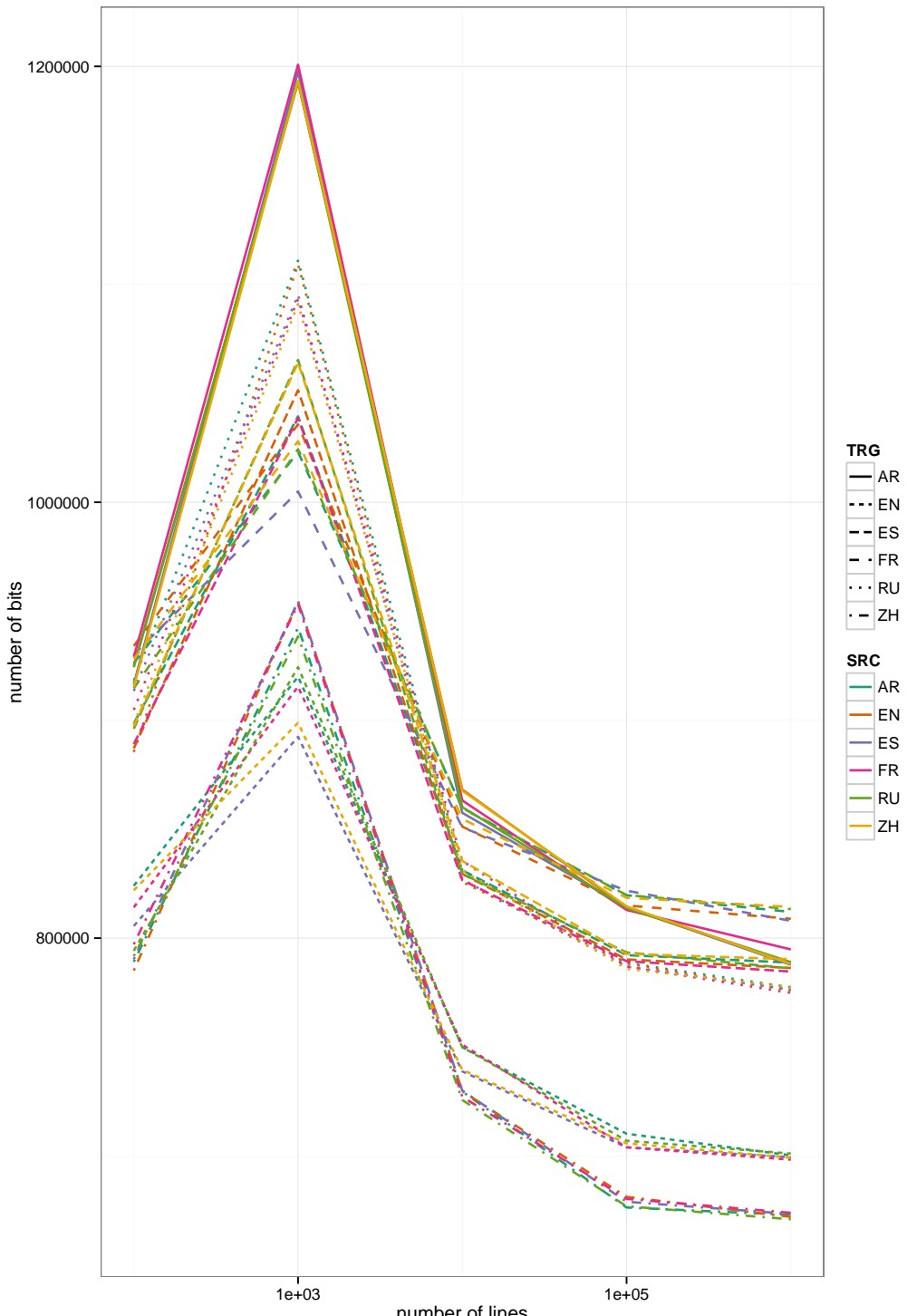

Figure 10: WORD models

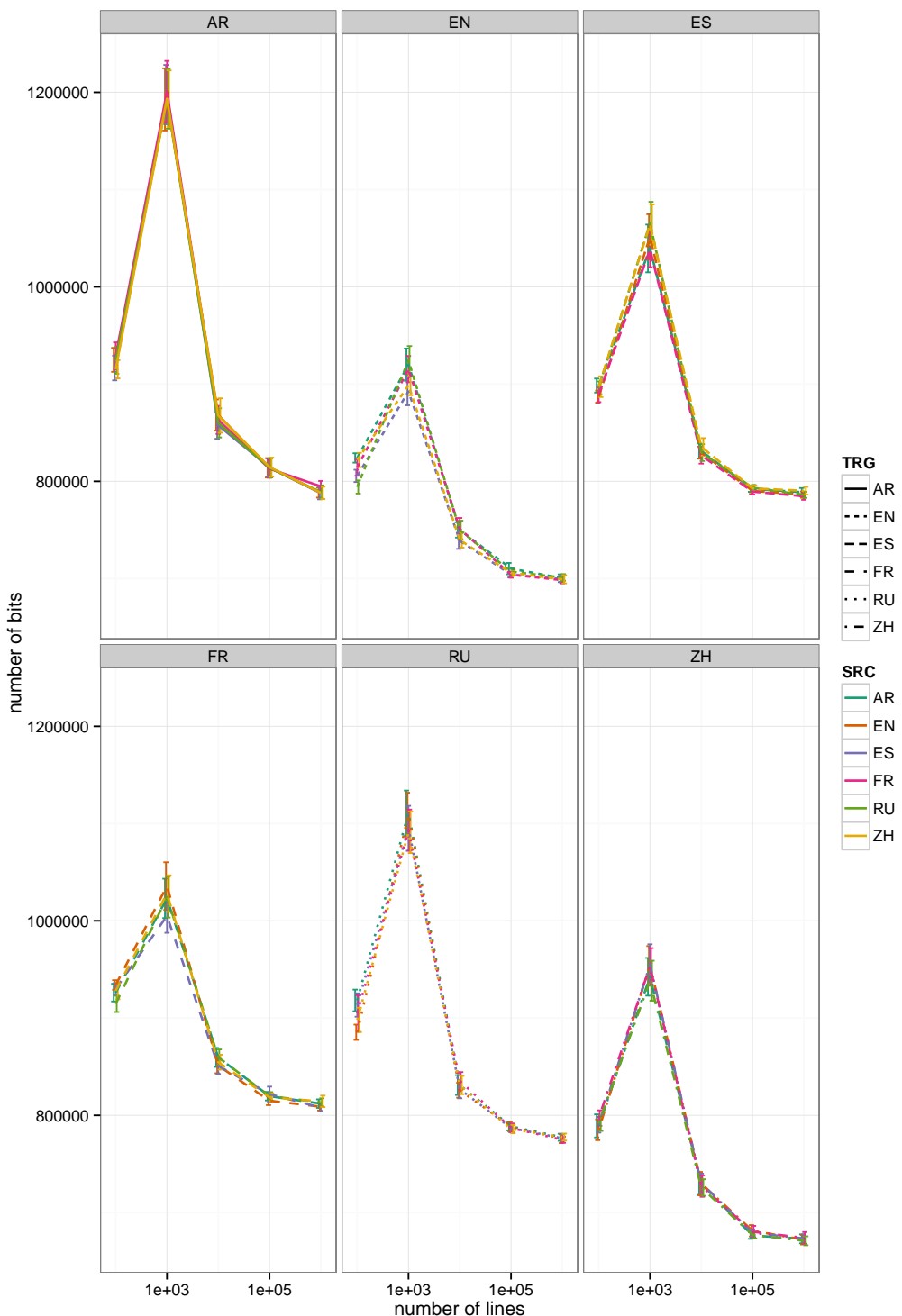

Figure 10: WORD models (target language as facet)

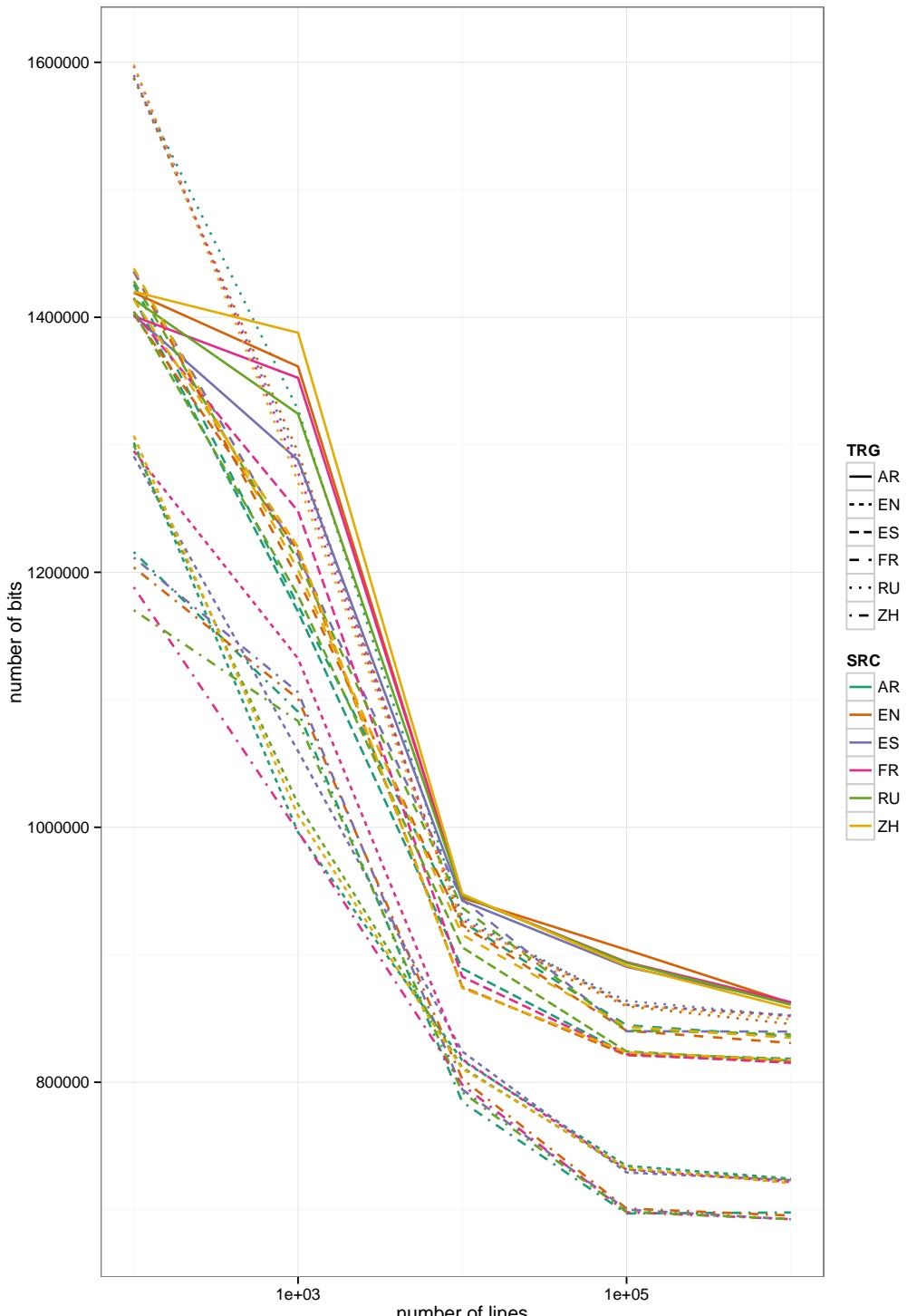

Figure 11: BPE models

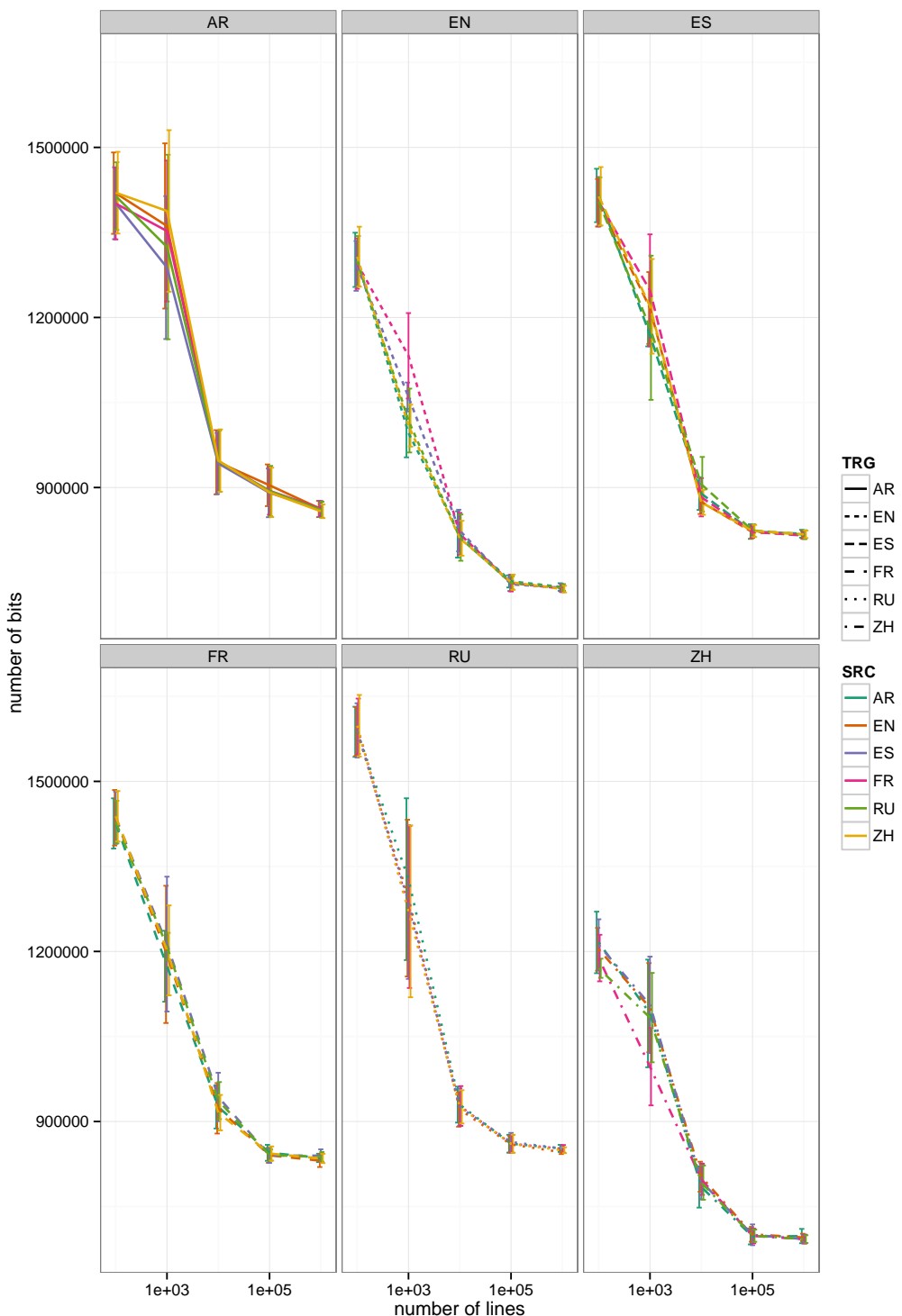

Figure 11: BPE models (target language as facet)

Fairness in Representation for Multilingual NLP: Insights from Controlled Experiments on Conditional Language Modeling

Version 1.1

ICLR 2022 camera-ready copy (20220510)

