# OpenReview forum: "Fairness in Representation for Multilingual NLP: Insights from Controlled Experiments on Conditional Language Modeling"
_ICLR.cc/2022/Conference — ICLR 2022 Spotlight_

### Official Review · Reviewer_UwwP · 2021-11-01

**Correctness:** 3
**Technical Novelty And Significance:** 2
**Empirical Novelty And Significance:** 2
**Recommendation:** 6
**Confidence:** 3

**Main Review:**

Existence of "inherent" difference in difficulty of NLP for different
languages is interesting for theoretical reasons as well as the
applied NLP. The paper also presents a large number of experiments in
a controlled way for this purpose.

Although I agree that the topic is important, and the study covers a
fair amount of experiments with sound methodology, I do not think we
learn a lot from the present paper: (1) the effectiveness of sub-word
units for morphologically rich languages is expected, and (2) the
domain of application (conditional LMs) is rather narrow. Given the
issue is studies using (non-conditional) LMs earlier (as the paper
cites), the benefits of narrowing it further down to conditional LMs
is not clear to me (after reading the paper). The issue with word
segmentation in Chinese is also studied before, and I do not see any
additional information the present paper provides.
Besides the main above point, which is the basis of my main evaluation
of the paper, there are a few minor issues I list below.

- The paper is written in a mainly error-free English, but it is
  rather difficult to read due to the organization and the flow of the
  concepts/text. Although I do not have many concrete suggestions, the
  this prevented me from understanding some simpler issues at first
  reading. Examples include:
    - There are some unclear statements like "To which extent is
      morphological complexity necessary in computing?" (p2): this
      probably means "handling/modeling morphological complexity" but
      literally, "need for complexity" is not something anyone would
      claim to be necessary.
    - There are divergences from the main topic of discussion, for
      example "Our conventional way of referring to “language” (as a
      socio-cultural product or with traditional word-based
      approaches, or even for most multilingual tasks and
      competitions) is too coarse-grained ..." (p5) is just to general
      to substantiate here while defining the training set size in the
      previous context. The same goes for the discussion of ZH as
      "high-resource" language on page 6: it is not clear to me why
      this is relevant here, and what is the relation to the overall
      aim of the paper.

- Overall, the paper feels too dense. In my opinion, paper could
  benefit from presenting less, explaining the main matters more
  clearly. For example, I do not fully see the utility of the
  byte-based models, especially that the paper goes into resolving
  practical issues due to code-set differences among languages. The
  additional data may be of interest for a small subset of the
  audience, but for the most, this is not relevant for the main
  question tackled. Leaving some of these data/exposition to appendix
  may allow allow clearer exposition of the main points.

- The figures, even the enlarged ones in the appendix, are rather
  difficult to read: too many lines/patterns and too small fonts. No
  concrete suggestions, but a clearer display of the results would
  help the reader a lot.

- "Summary of findings and insights" is too detailed at this point. A
  more concise, general version would be much better. Many of the
  references/notation here is not possible digest before reading the
  rest of paper.

- Although the data used is a standard data set, it would be useful
  for most readers to have more detailed description of the data
  (e.g., the type of documents, size of the overall data that the
  parts used are sampled, ...). It would also be good to indicate what
  a "line" corresponds to (is it a sentence, a paragraph, or some
  other unit?).

- Footnote marks should be place after punctuation.



**Summary Of The Paper:**

This paper explores the performance of conditional language models for
different languages with the aim of establishing "fairness" in NLP
systems across languages. In particular, the paper focuses on
perceived difficulty of "morphologically rich" languages for NLP.  The
paper presents a series of experiments with 6 languages in the UN
parallel corpus, using different units (words, sub-words, characters,
bytes) for training the conditional language models. The paper
concludes with the general observation that the use of characters or
bytes as units instead of words removes (or reduces) the performance
disparity between different languages.


**Summary Of The Review:**

Although I agree that the topic is important, and the study covers a
fair amount of experiments with sound methodology, I do not think we
learn a lot from the present paper: (1) the effectiveness of sub-word
units for morphologically rich languages is expected, and (2) the
domain of application (conditional LMs) is rather narrow. Given the
issue is studies using (non-conditional) LMs earlier (as the paper
cites), the benefits of narrowing it further down to conditional LMs
is not clear to me (after reading the paper). The issue with word
segmentation in Chinese is also studied before, and I do not see any
additional information the present paper provides.

---

> ### Author Response · Authors · 2021-11-16
> **first response to Reviewer UwwP (1/n)**
>
> Dear Reviewer UwwP:
>
> Thank you for your review.
>
> We appreciate your recognition for our tackling an important topic with sound empirical work.
>
> There seems to be some misalignment between what you saw from our paper, at least as reflected through your initial review, and our findings and intent behind this work. We hope we could use our rebuttal here to convince you to re-evaluate the paper from a different perspective.
>
> First of all, as the first two keywords of our paper show, we perform a fairness evaluation. We wanted to evaluate the validity of morphological complexity (based on languages both "morphologically rich" and "morphologically poor", not just "morphologically rich") and understand its nature from both a qualitative and quantitative point of view. This is a scientific paper, not an engineering/application paper.
>
> Our major novel findings include: 1. source language neutralization, 2. representation relativity, 3. explaining away / resolving morphological complexity and examining its relevance in the context of computing, and 4. relating finer granularities to the elimination of performance disparity verified through statistical comparisons. Furthermore, we adapted statistical comparison as a "fairness metric" and explained the implicit elegance of (dev) set evaluation with unnormalized PP. (That is, we interpreted merits of our current technologies in new light.) Other novel aspects include: CLM evaluation in relation to data size, representation granularity, and quantitative and qualitative fairness, a numerical analysis on DD for seq2seq models and relating the phenomenon to data types, and erraticity as a length-induced meta phenomenon.
>
> Re subword units: the point is that, despite their alleged effectiveness for morphologically rich languages, they are still not as good as characters and bytes in closing the disparity gap. We did not make that explicit before, but have now revised accordingly (in version(s) _after_ v0.2).
>
> Re CLMs: the paper points out both disparities in MT and LM results (§1). Our controlled experiments on CLMing helped improve our understanding of the inner workings of the Transformer encoder-decoder models. The one-language-to-one-language setting allows us to contrast source and target more clearly. As encoder-decoder models underlie much of current NLP applications (including MT), we don't understand why you would think that it's too narrow. Would you mind please explaining?
>
> Re CWS: our work is the first of its kind (in fact, there has been no kind like it before!) to see that there are no statistically significant performance disparities between ZH and other "morphologically rich" languages.
>
> Re the minor issues you listed:
>
> - Re "rather difficult to read":
>
>     - Re "To which extent is morphological complexity necessary in computing?":
>
>         For now, we changed it to "When is the concept of morphological complexity relevant to NLP?", but it could also be "When is the concept of morphological complexity relevant to computing?" --- would you have objections to the latter formulation? Our analytic solution holds.

---

> > ### Author Response · Authors · 2021-11-16
> > **first response to Reviewer UwwP (2/n)**
> >
> > - (cont'd) Re "rather difficult to read":
> >
> >     - Re how "[t]here are divergences from the main topic of discussion ... Our conventional way of referring to 'language' (as a socio-cultural product or with traditional word-based approaches, or even for most multilingual tasks and competitions) is too coarse-grained ...":
> >
> >         We believe this relates to your comment about the paper being rather difficult to read. The domain-specific [1] context from NLP [2] relevant to this remark relates to the line of discourse in the past decades on the relevance of linguistics or linguistic concepts in NLP. The work cited in §1.1 --- Bender (2009), Cotterell et al. (2018), and Mielke et al. (2019), as well as Fisch et al. (2019) and Ponti et al. (2019) all allude to this implicitly/explicitly. (These are just the ones most relevant to our paper.)
> >
> >         Historically, NLP started as an interdisciplinary field with computer science / information sciences and linguistics. However, there are times when only one form of linguistics (e.g. the structural linguistics that gave rise to morphology / morphological practices) became too dominant (see e.g. §1.1 and §1.2 in [i]). As technology developed over the past decades and manual engineering became less relevant, the NLP community has been finding harder and harder to incorporate traditional structuralist linguistic concepts into more fine-grained NLP systems (though many have done so as a form of courtesy entry-level pedagogical exercises [3], for the sake of "tradition"). It has been unobvious/unknown to most (or all) community members where exactly the issue lies.
> >
> >         On this front, the significance of our paper lies in how our results confirm that morphology is bounded to the word level (as per definition [4]), how morphological complexity automatically shows so long as we "word"-tokenize our data, how we eliminated complexity, and that "word" is a problematic concept, inconsistent across languages and also ambiguous within any one language. ("Word" underlies most/all of structural linguistics. Morphology, morphosyntax, and syntax are all intermingled with each other due to the fuzziness of "words", esp. in multilinguality.) Before our work, it is generally believed that morphological complexity in languages is intrinsic and real, and that the "word" view of language applies universally. We show that that is not true --- linguistic morphology is not applicable on character and byte levels. Previous work may have shown different languages as having different performance for different linguistic tasks (e.g. Durrani et al. (2019)), or pointing to how data statistics is at work in contributing to hardness (Mielke et al. (2019)), but no work has shown explicitly how different granularities have different behavioral patterns. Our work shows that there are representations to "word" that can be fairer and conceptually more robust and standardized for computing, despite how much of structural linguistics has overfitted to the "word" interpretation of language.
> >
> >     - Re "[t]he same goes for the discussion of ZH as 'high-resource' language on page 6: it is not clear to me why this is relevant here, and what is the relation to the overall aim of the paper":
> >
> >         ZH has been mis-/under-evaluated in the context of NLP and canonical structural linguistics. Some NLPers have been "looking for the right word" (so to speak) for decades. Our results confirm that we can be "word"-free now --- characters also happen to be a unit that ZH speakers tend to identify with as "words". [5]
> >
> >         In some traditions of language sciences, ZH is considered an outlier. One could achieve parity well with it usually just with phonetic representations (similar to our pinyin solution). With "word" representation, ZH is usually parsed with an interpretation that is EN-centric. [6] investigated the difficulties in parsing ZH and found error types in ZH that are rare or absent in EN. [7] performed a comprehensive automatic error analysis of error types made by ZH parsers and concluded that the distinctive properties of ZH syntax remained to be addressed. One can think of parsing ZH as only possible when it is based on a "word" structure that is _x_-centric, where _x_ can be any language --- but "we find that it is not necessary to assign a perspective that is centered on any one particular language, when we can evaluate simply by the total number of bits for a larger portion of texts/sequences." (§2 under "Fair information-theoretic evaluation metric").

---

> > > ### Author Response · Authors · 2021-11-16
> > > **first response to Reviewer UwwP (3/n, n=3)**
> > >
> > >
> > > - Re "the utility of the byte-based models":
> > >
> > >     This is actually one main result of the paper. We achieved parity (or the lack of disparity) with ZH and all the others. It shows that language complexity and the lack thereof does not exist intrinsically, i.e. as a natural or universal property of language. ("Language" here defined as language as a whole. Surely, as we show, languages wrt representation types can be harder/easier to model depending on the context, but they do not have to if equitable measures are taken.) Our findings provide empirical evidence supporting the need to be more specific, fine-grained, and technically robust when we refer to and process language.
> > >
> > > - Re figures:
> > >
> > >     Did our recommendation (from our earlier reply with subject "v0.2 uploaded") to read between the set of lines help?
> > >
> > >     E.g. the first figure in App. E (Figure 4: CHAR: character models), you see the set of lines that have a pattern different from the rest? That is with ZH_trg. That's all one needs to see from that figure.
> > >
> > >     Again, results are summed up in Table 1.
> > >
> > > - Re summary of findings and insights:
> > >
> > >     Does the summary in v0.2 now read ok?
> > >
> > > - Re dataset description and "line":
> > >
> > >     Does the description in App. A suffice?
> > >
> > > - Re footnote marks:
> > >
> > >     We use footnote marks at end of sentence in a logical manner --- if the footnote refers to just the last unit in the sentence, we place it before the period. If it applies to the sentence, after.
> > >
> > > We hope to have answered to your questions sufficiently and helped you see the merits of this paper in new light. We look forward to your feedback. Please pardon the delay in our reply as some of material discussed (e.g. historical context of this paper) is hard to talk about, but we think it is high time for it to be discussed.
> > >
> > > Thank you again for your review.
> > >
> > > Sincerely,
> > >
> > > Authors3684
> > >
> > > [1] Data science (DS) is a recognized track at ICLR. And the 3 components of DS are domain knowledge / substantive expertise, mathematics and statistics, and computer science / hacking skills (see e.g. http://drewconway.com/zia/2013/3/26/the-data-science-venn-diagram).
> > >
> > > [2] As common in the tradition of NLP or CL (Computational Linguistics), we use the term "NLP" in this work as an umbrella term to describe the practice of language processing --- which has primarily been text processing. As stated in the seminal text from [i]: "[g]enerally in Statistical NLP, people cannot actually work from observing a large amount of language use situated within its context in the world. So, instead, people simply use texts, and regard the textual context as a surrogate for situating language in a real world context." (p. 5-6).
> > >
> > > [i] Christopher Manning and Hinrich Schütze. 1999. Foundations of Statistical Natural Language Processing, Cambridge, Massachusetts, USA. MIT Press.
> > >
> > > [3] The results of this paper do, however, beg the question as to what kind of entry-level pedagogical exercises should be deemed as appropriate for NLP.
> > >
> > > [4] Since the findings of a previous version of this paper became public, the definition of morphology has been modified to being "sentence-level" (see proposal in [ii]). But "sentences" are also neither standardized nor universal units.
> > >
> > > [ii] Omer Goldman and Reut Tsarfaty. 2021. Well-Defined Morphology is Sentence-Level Morphology. https://aclanthology.org/2021.mrl-1.23.pdf
> > >
> > > [5] There are communities in the world with a definition/awareness of "word" that is like EN, there are some which have an indigenous sense that is different, there are also some which do not have such concept at all. This is why "word" is problematic --- it either doesn't translate, or if it does, it's unrobust, and if we try to use one sense, it's imperialistic/unfair.
> > >
> > > [6] Roger Levy and Christopher D. Manning. 2003. Is it harder to parse Chinese, or the Chinese treebank? https://www.aclweb.org/anthology/P03-1056.
> > >
> > > [7] Jonathan K. Kummerfeld, Daniel Tse, James R. Curran, and Dan Klein. 2013. An empirical examination of challenges in Chinese parsing. https://www.aclweb.org/anthology/P13-2018.

---

> ### Author Response · Authors · 2021-11-24
> **requesting updated feedback**
>
> Dear Reviewer UwwP:
>
> We are sorry that we have not heard back from you at all since we submitted our rebuttals.
>
> Would you mind please giving us an update on your new interpretation of our work (v0.3.2) given the revision and the information from the rebuttal phase from the past 2 weeks?
>
> What remaining doubts/questions do you have? We'd be happy to clarify and/or discuss.
>
> What would it take for us to get a good recommendation from you at this point?
>
> Thank you very much in advance for your information.
>
> Sincerely,
>
> Authors3684

---

### Official Review · Reviewer_UWb7 · 2021-11-02

**Correctness:** 3
**Technical Novelty And Significance:** 3
**Empirical Novelty And Significance:** 3
**Recommendation:** 6
**Confidence:** 5

**Main Review:**

I want to thank the authors for the interesting paper. Overall, the
paper contains a substantial amount of experiment results with some
interesting findings, but the paper is not easy to read and not easy to
follow. I am also not sure if this paper reveals substantially new
things to the readers.

The strength of the paper is as follows:

-   This paper proposes to use conditional language modeling (machine
    translation) to evaluate how hard it is for transformer models to
    model a language. This is a clever setting that allows one to fix a
    source language and compare the difficulty of modeling different
    target languages.

-   The paper uses statistical tests to verify if there exists
    performance disparity between languages of different morphological
    complexity.

-   This paper shows that transformer conditional language models are
    equally capable of modeling the six languages evaluated in the paper
    when the suitable representation level is chosen. This indicates
    that transformer models are **fair** and are not biased toward any
    the language they tested, and morphological complexity is not indicative
    of how hard it is for transformer-based conditional language models
    to learn.

The weakness of this paper and questions about the paper is listed as
follows:

### Not enough novelty compared to prior works

An important prior work of this paper is Mielke et al. (2019). I find
this paper highly resembles the work of Mielke’s from some aspects. I
will summarize the work of Mielke et al. (2019) as follows first:

1.  Research Question: Do LSTM-based language models serve all
    languages? What are the characteristics of languages that are easier
    for LSTM-based language models to model?

2.  Experimental Setup: Use parallel corpora containing 21 languages.
    Train a mono-lingual character/BPE LSTM language model on each
    language, and evaluate the development set perplexity for each
    language.

3.  Results and Findings: Language difficulties are not significantly
    predicted by linguistic factors (including morphological
    complexity), no matter the representation is character or BPE. For
    LSTM language model with character-based representation, the
    difficulty for LSTM to model correlates with the raw length in
    characters. For LSTM language model with BPE-based representation,
    the difficulty for LSTM to model correlates with the vocabulary
    size.

The main difference between this work and Mielke et al. (2019) are as
follows:
1.  The models used in this work are transformer-based conditional
    language models, while Mielke et al. (2019) used RNN-based
    (unconditioned) language models.

2.  The indicator of performance in this paper is unnormalized
    perplexity score, and Mielke et al. (2019) used the normalized
    version.

3.  This paper uses statistical tests to compare if there exists
    performance *disparity* between language pairs, which was not
    presented in Mielke et al. (2019) (since Mielke et al. (2019) did
    not use conditional language modeling).

The main findings of Mielke et al. (2019) that I listed above are also
observed in this work. And this work also has the following new findings
that are not presented Mielke et al. (2019):

1.  They find that for transformer conditional language models, the
    difficulty of modeling a certain target language is independent of
    the source language.

While the experiment settings are slightly different, from the aspects
of the model architecture and evaluation metrics, the core of the experiment
and the main takeaway does not make me learn substantially new things. I
do recognize that it is important to empirically verify the results of
Mielke et al. (2019) on transformer-based conditional language models,
but I have reservations about whether the contribution of this work is
ample enough.

Additionally, I think this paper will definitely benefit from a more
thorough discussion of their difference with the previous works.

### Experiment settings are not clear enough to follow

-   In the "Disparity/Inequality" part in Section 2, the paper states
    "Each $l_{src}$ or each $l_{tgt}$
    consists of scores from all models trained across various sizes and
    directions." I have difficulty understanding how the comparison
    tests can be done. In my opinion, this is the most novel setting in
    this work, and I would like to see more elaboration! Since for each
    source/target language, there are scores for different directions,
    dataset sizes, random seeds, and datasets, how is the comparison
    between the "distribution of the scores" done? As far as I
    understand, the "distribution of the scores" of a language will
    consist of scores obtained under different settings, and I am not
    sure if comparing the distribution as a whole is rational. Since the
    score distribution under certain factor fixed (for example, when the
    dataset size is fixed) may be statistically significantly different,
    but when all the factors change together, we cannot tell the
    influence of each factor.

-   Is there any reason why not running the $10^6$ for setting
    A3-7, B1 & C1.

-   I don’t understand how the random seeds are controlled in their
    experiments. Is the random seed only related to how the model is
    initialized? Or the random seed will also affect how dataset B and C
    are sampled? Why is there not a C(0) model (in appendix A). I think
    for different data sizes and datasets, they should have a common
    random seed, and the experiments should be run using multiple
    different common random seeds.

-   I have difficulty understanding what "transformer-preprocess d;
    transformer-prostprocess drn" in appendix B is. I have never seen
    this parameter in the context of transformer-based language models.

### Some words or statements are ambiguous

-   In Section 1.1, "favoring of certain outcomes". I do not understand
    what "certain outcomes" are in this paper after I read the whole
    paper. Do the authors mean "better at modeling certain languages"?

-   In Section 1.2, "bounded by representation level". I am not sure
    what "bounded by" means here , and I am skeptical about the phrase
    "bounded by". Since the experiments only use a set of
    hyperparameter, the claim that "representation bounds hardness" may
    be too strong. May be the hyperparameter just don’t work for certain
    representation.

-   Section 2, in **Conditional language modeling**: "without performing
    any downstream translation". What is "downstream translation"? This
    seems not to be a common phrase in the context of machine
    translation, and may need some explanation. Do the authors try to
    refer to something like beam search?

-   Section 2, in **Conditional language modeling**: "To explicitly
    focus on modeling the complexities that may or may not be intrinsic
    to the languages". I don’t understand why this paper is able to
    explicitly focus on "modeling the complexities" under their setting.
    I also don’t quite get what is intrinsic to the languages and what
    is not.

-   Section 2, in **Controlled experiments as basic research for
    scientific understandings**: "that make language data different from
    other data types". I fail to understand what "other data types" this
    paper is trying to refer to. I also don’t see the connection between
    the relationship between the quoted line above and "data-centric
    approach".

-   Section 5, "disparity is not a necessary condition". What is the
    necessary condition for?

### Some claims in the paper are not well supported by the experiments results

For example, the paper writes in page 6 that \"on the byte level,
$AR_{trg}$ and $RU_{trg}$ display non-monotonic and unstable
behaviors\". I am not sure what the \"non-monotonic behaviors\" the
paper is referring to, but I think in Figure 1(e), not only AR and RU
are non-monotonic, all results in Figure 1(e) are not monotonic. All of
them decay from $1e-02$, reach the lowest perplexity at $1e-04$, and
rise afterward.

### Miscellaneous concerns

-   I am concern about the title \"Fairness in Representation\". First,
    since the word fairness is mostly used to refer to societal biases
    in NLP community, I am not sure if the word 'fairness' may be
    misleading at the first sight. Maybe statistical bias will be a more
    better choice. Second, I am not sure what \"Fairness in
    Representation\" means; it seems to refer to the performance
    disparity between different representation levels. But since the core
    problem in the paper is the disparity between different languages of
    different morphological complexity, I guess the whole paper is more
    like \"Transformer's Fairness for Different Languages\" instead of
    \"Fairness in Representation\".

-   Some passages seem to be not highly related to the experiment
    results. For example, in page 6, I don't see what the passages \"But
    this is often \... and accepted in NLP.\" has to do with the
    experiment result in this section.

-   I am concerned about using Pinyin in Chinese. Will single Pinyin
    correspond to multiple characters in Chinese? If so, then changing
    the target from character to Pinyin does not merely change the
    vocabulary size and makes the whole comparison unclear. Whether
    Pinyin can be an alternative for character needs to be carefully
    discussed.

-   I am concerned about the paper's only using a set of hyperparameter. I
    understand that using the same set of hyperparamater is for
    \"controlled experiments as basic research for scientific
    understanding\", but sometimes the paper attributes the performance
    fluctuation to the fact that they did not search for better
    hyperparameter. This seems contradicting, and one can question that
    the results in the paper will not hold if we had searched the
    optimal parameters for different language pairs, dataset, data
    sizes.

-   If the goal is to compare morphologically rich languages (AR, RU)
    and frugal language (ZH), then why does the paper also compare the
    other three languages in United Nations Parallel Corpus? What do
    their results tell us? What is the morphological complexity of the
    remaining three languages?

-   The languages used in this paper is not too many. The conclusion of
    Mielke et al. (2019) are based on 21 languages, I am not sure
    whether the conclusion of this paper will still hold if more
    languages are included.

-   If in 2(b,d), only ZH is changed, I think it is better to only show
    the result of ZH as a target since all remaining 5 subsubfigures are
    the same as in 1(d).

### Presentation problems

The figures are not easy to read. Figure 1, 2, 3 are too small and are
unreadable on A4 paper. The problem of the figure hinders my
understanding to some extent. For example, in the second last line in
page 7, it is said that \"AR/RU \> ES/FR \> EN/ZH\", I cannot arrive at
the conclusion since Figure 1(c) is too small to read. While the enlarged
versions of figures are provided in the appendices, I still find them not
so useful for some cases; for example, the standard error bars in Figure
4 EN are overlapped with the lines.

**Summary Of The Paper:**

This paper asks: Are languages which have been traditionally considered
morphologically rich (Arabic and Russian) and poor (Chinese) equally
hard to learn by transformer-based conditional language model (CLM).

To quantify how hard it is for CLM to model a language $l_1$, this paper
proposes to train a transformer-based encoder-decoder model to translate
$l_1$ to another language $l_2$, and uses the \"unnormalized perplexity
score\" (the number of bits needed to encode the same development set
for the target language) as an indicator for the \"hardness\" for CLM to
learn the $l_2$. For two languages $l_1$ and $l_2$, there exists
**disparity** if their perplexity distributions are significantly
different. This paper includes extensive experiments across 30 language
pairs using three different representation level (including character,
byte, and word) for different dataset sizes.

They have the following findings:

1.  They find that the performance disparity between morphologically
    rich and poor languages cannot be justified due to morphological
    complexity. Representation level matters more.

2.  They find that the hardness for CLM to model a language is related
    to the level of representation used to encode the language, the
    vocabulary size, and/or the sequence length.

3.  They find that it is equally hard to learn a certain target language
    (that are tested in their experiment) using transformer-based
    conditional language models, independent of the source language.

**Summary Of The Review:**

This paper studies the fairness of transformer models to model languages
of different morphological complexity. This work extends the previous
observation from Mielke et al. (2019) and show that similar results hold
for transformer-based conditional language models: morphological
complexity is not indicative of the difficulty for transformer to conditional
language model, and representation level matters more. This indicates
that transformers are **fair** towards the languages tested in this
work, in terms of conditional language modeling. They propose to use
conditional language models to make it easier to compare the difficulty
of modeling a specific target language, and the usage of statistical
tests for testing performance disparity among languages is a novel act.
However, experiment settings and findings/results resemble the work of
Mielke et al. (2019) to some degree. The part on statistical testings is
somewhat baffling and some of the statements are ambiguous. The figures
for the main experiments are not very easy to read. The title is also
somewhat not quite indicative of the content (while it does seem
relevant), mainly because of the term \"fairness in representation\".

## Update on Nov 22
After a thorough and in-depth discussion with the authors, I upvote the score from 3 (reject, not good enough) to 5 (marginally below the acceptance threshold), meaning that I am still inclined to reject.
I change my score mainly because the authors have addressed many of my initial concerns, including the experiment settings, some odd wordings or claims, and the presentation problems.
But I still do not find the paper is good enough to be accepted.
I am still not convinced by the authors' claim of their novelty; the novelties listed by the authors seem to be more of "something that was not being done in prior works", but I am not sure how those "something" can much advance our understanding on this topic.
Overall, I think this paper presents some nice and well-controlled experiments, but the interpretation of the results tends to be too high-level and might be overstating; I often have difficulty linking the quantitative results and qualitative interpretations.
I am not sure what I have learned from this paper.


## Edit again on Nov 22
I will upvote my score to 6.
After going over the paper several times, I think there indeed are certain empirical contributions of the paper.
I think varying the data size is an important experiment that has yet to be presented in any prior works and is valuable to be empirically verified, I do agree with the authors on this.

---

> ### Author Response · Authors · 2021-11-17
> **first response to Reviewer UWb7 (1/n)**
>
> Dear Reviewer UWb7:
>
> Thank you very much for your informative review, and your finding our paper interesting and experimental approach with CLMing clever.
>
> We thought to approach it last because all our prior responses to other reviewers would help us answer your questions and help you understand more about our work. They are also intended for you. We hope you have had time to look through them.
>
> There seem to be many misunderstandings. We will try to rectify what we can here.
>
> Clarifications on your Summary of the Paper:
>
> 1. We are saying that "morphological complexity" as we know it thus far is only relevant on the "word" level. We did not say anything about representation granularity matter more or less. Different granularities have different data statistics and different behavioral patterns as learned by our Transformer CLMs. (Fig. 1).
>
> 3. We did not claim "equally hard" but that there are no statistically significant differences among the source languages and among the target languages, both sets evaluated through pairwise comparisons. It is possible that there are no differences in CLMing the 6 languages. However, as we stated p.8: "[w]hile the lack of significant differences between pairs of source languages would signify neutralization of source language instances, it does not mean that source languages have no effect on target.".
>
> Re your Main Review:
>
> - Re **novelty of our work** (this is not an exhaustive list), this is also what we wrote to Reviewer UwwP:
>
>     "Our major novel findings include: 1. source language neutralization, 2. representation relativity, 3. explaining away / resolving morphological complexity and examining its relevance in the context of computing, and 4. relating finer granularities to the elimination of performance disparity verified through statistical comparisons. Furthermore, we adapted statistical comparison as a "fairness metric" and explained the implicit elegance of (dev) set evaluation with unnormalized PP. (That is, we interpreted merits of our current technologies in new light.) Other novel aspects include: CLM evaluation in relation to data size, representation granularity, and quantitative and qualitative fairness, a numerical analysis on DD for seq2seq models and relating the phenomenon to data types, and erraticity as a length-induced meta phenomenon."
>
>     How much of the above do you not think is novel?
>
>     This paper is supposed to be a scientific paper, not an engineering/application paper. We are not here to tell you what trick you can use to train a model better. If our work helped *improve your understanding* on _anything_, then it has served its goal. And if you find our argument valid, then please accept. If not, please let us know which formulation can be improved or which argument you are not convinced by.
>
> - Re your comment "[t]his paper shows that transformer conditional language models are _equally capable_ of modeling the six languages evaluated in the paper when the suitable representation level is chosen. This indicates that transformer models are _fair_ and are not biased toward any the language they tested, and morphological complexity is not indicative of how hard it is for transformer-based conditional language models to learn.":
>
>     We show that there are _no statistically significant differences_, not "equality" (that's just what statistical comparisons can do) when the suitable representation is chosen and/or when the statistics reflect so. We did not make any statement about the "fairness" of the model. We also did not make any statement about whether it is baised toward any of the language we tested, except we did find ZH script pattern represented in Wubi _may_ be one that is problematic. But further experimentation would be necessary to substantiate this claim for certain. Re "morphological complexity": it is only on the word level, i.e. if we want to model it, though the status, esp. of ZH (but actually for all languages) is murky because of "words".
>
> - Re differences to Mielke et al. (2019):
>
>     As stated in §5 under "Additional related work", "[w]e go beyond their work in monolingual LMs to study CLMs and evaluate also in relation to data size, representational granularity, and quantitative and qualitative fairness. To the best of our knowledge, there has been no prior work on demonstrating the neutralization of source language instances through statistical comparisons, a numerical analysis on DD for sequence-to-sequence models, the meta phenomenon of a sample-wise non-monotonicity (erraticity) being related to length."
>
>     Mielke et al. did not discuss the relevance of morphological complexity wrt representation granularity. They did not try to make equitable measures such that there'd be no performance disparity. They also tested only with one data size.
>
> (to be continued)

---

> > ### Author Response · Authors · 2021-11-18
> > **first response to Reviewer UWb7 (2/n)**
> >
> > - (cont'd) Re differences to Mielke et al. (2019):
> >
> >     - You wrote "3. This paper uses statistical tests to compare if there exists performance disparity between language pairs, which was not presented in Mielke et al. (2019) (since Mielke et al. (2019) did not use conditional language modeling).":
> >
> >         One does not need to use CLM for statistical comparisons. Theoretically, Mielke et al (2019) could also have compared their results. But they only performed 1 run.
> >
> >     - Re "1. They find that for transformer conditional language models, the difficulty of modeling a certain target language is independent of the source language.":
> >
> >         This is incorrect. Again, please see 2nd paragraph in §5 beginning with: "While the lack of significant differences between pairs of source languages would signify neutralization of source language instances, it does not mean that source languages have no effect on target.". We compared source with source, target with target.
> >
> > Again, re novelty of our work: if you don't think our findings are novel, **would you please provide us with specific prior literature on the following? We would be grateful for the information, as it is the case that _to the best of our knowledge_, all these findings are new**:
> >
> > 1. source language neutralization in Transformer CLMing with statistical comparisons,
> >
> > 2. representation relativity across magnitudes of data size with visualization,
> >
> > 3. explaining away / resolving morphological complexity and examining its relevance in the context of computing,
> >
> > 4. relating finer granularities to the elimination of performance disparity, verified through statistical comparisons,
> >
> > 5. adaptation of statistical comparisons as a "fairness metric" for a relational assessment and evaluation of languages,
> >
> > 6. explaining the implicit elegance of (dev) set evaluation with unnormalized PP. (That is, we interpreted merits of our current technologies in new light.)
> >
> > 7. CLM evaluation across data sizes wrt representation granularity in a one-setting-for-all configuration,
> >
> > 8. discussion of qualitative issues for languages / language data for fairness in NLP,
> >
> > 9. a numerical analysis on DD for seq2seq models and relating the phenomenon to data types (bytes, chars, words),
> >
> > 10. erraticity as a length-induced meta phenomenon,
> >
> > (list of novel elements is non-exhaustive)
> >
> > ==========
> >
> > Re experimental settings:
> >
> > - Re statistical comparisons:
> >
> >     We use the replicability analysis toolkit from [1] for our comparisons. It was originally designed "to help researchers and engineers draw statistically sound conclusions about the difference in performance between two algorithms, based on multiple comparisons between these algorithms." [2] Instead of comparing algorithms, we used the toolkit to compare data (more specifically, we compared languages, i.e. language data). Methods are standard in statistics. For more info, please refer to §2, [1], and [2].
> >
> >     -----
> >     To get the samples:
> >
> >     For each representation, we used 3 runs in 5 sizes for each l_src and each l_trg. For example, for the character (primary) representation, for AR as source (AR_src):
> >
> >     Out of the 30 possible directions, there are 5 models involving AR_src trained for each run and data size (i.e. the directions: AR-EN, AR-ES, AR-FR, AR-RU, AR-ZH). For each direction, there are 15 models trained (3 runs x 5 sizes). We take all 75 char models (15 models x 5 directions) involving AR_src as our sample for AR_src (sample_ar_src). That's a sample of size 75.
> >
> >     Likewise, for EN_src (EN-AR, EN-ES, EN-FR, EN-RU, EN-ZH), we also have 75 data points (sample_en_src).
> >
> >     Likewise for all 6 l_src and all 6 l_trg.
> >
> >     For the comparisons, we compare pairwise, e.g. sample_ar_src with sample_en_src.
> >
> >     There are 15 pairs to compare among the source languages. And 15 pairs to compare among the target languages.
> >
> >     (We plan to elaborate on this in an extended version of the paper. The algorithm and rationale, however, have already been described in [1]. But for this paper, we will add in the above in the appendix. Thank you for raising it to our awareness that it could be more explicit.)
> >
> >     While it is also possible to compare each data size separately (or other combinations/slices of our data), _for this paper_, we compared "languages" because we wanted to find out about languages (each in its "totality", so to speak, i.e. all data points for each representation).
> >
> >     [1] Dror et al. 2017. Replicability analysis for natural language processing: Testing significance with multiple datasets. http://aclweb.org/anthology/Q17-1033.
> >
> >     [2] https://github.com/rtmdrr/replicability-analysis-NLP
> >
> >     -----
> >
> > - Reasons why we didn't run 10^6 for runs A3-7, B1 & C1? Limitations in computing resources. (These runs do not play a role in the statistical comparisons.)

---

> > > ### Author Response · Authors · 2021-11-18
> > > **first response to Reviewer UWb7 (3/n)**
> > >
> > > - Re seeds: Yes, for initialization. No, they do not affect how datasets B and C are sampled. There _is_ a C0 run. Re "I think for different data sizes and datasets, they should have a common random seed, and the experiments should be run using multiple different common random seeds.": we don't think it would matter significantly. Sure, it _might_ effect trivial differences in absolute scores. But the bar is higher if one is measuring statistically significant differences. Also, if the conclusion via correlation tests is that results mirror data statistics and they do appear so, there are good reasons to consider our results as substantive.
> > >
> > > - Re "transformer-preprocess d; transformer-prostprocess drn": This is just how Sockeye allows one to configure the cells "(d=dropout, r=residual connection, n=layer normalization)", see [3]. (Again, it does not matter. We could have used any setting, as long as all hyperparameters are the same. It is a set of controlled experiments. Our goal is _not_ to train the best system.)
> > >
> > >     [3] https://github.com/awslabs/sockeye/blob/1.18.85/sockeye/arguments.py#L674
> > >
> > > ==========
> > >
> > > Re wording:
> > >
> > > - §1.1, "favoring of certain outcomes" was meant to just clarify what "bias" means, as a short generic definition.
> > >
> > > - §1.2, "bounded by representation level": confined to the representation level?
> > >
> > >     "Bound" is a common term in computational complexity theory. It is true, however, that no one had thought of morphological complexity etc. having a lower or upper bound before wrt representation granularity. (Well, isn't our work novel then?) **May we ask why you are skeptical about the phrase "bounded by"?**
> > >
> > >     Using a set of hyperparameters does not have anything to do with the claim "[h]ardness in modeling is relative to and bounded by its representation level". And the claim is independent of absolute scores. Representation relativity just means that different representational granularity is different, data can have different behavioral patterns (_can_, not _must_). It can be useful to know, so one does not just get a result in one granularity and expect that the result would necessarily transfer to another granularity just because it is on a coarser symbolic level "the same thing".
> > >
> > > - §2, "downstream translation": we just meant "translation", i.e. decoding. It's been rectified in v0.2.
> > >
> > > (to be continued)
> > > Please allow us to take a bit of time to reflect on your questions. Some of them do help us think about things from different perspectives. Please also feel free to "interrupt" and reply to any of our questions to you at any time. Thanks.

---

> > > > ### Comment · Reviewer_UWb7 · 2021-11-18
> > > > **Re: response 3/n**
> > > >
> > > > I would like to thank the authors' detailed reply.
> > > > The responses are quite a lot, so I will start from the last one.
> > > > * I should be more specific. I meant to ask why is there not a C0(13). I understand the authors' points on "random seed might not matter significantly to the result", but I would like to know more about why the authors choose the random seed this way, instead of a more systematic way. I can accept answers like "we just choose the random seeds randomly", I just want to know why.
> > > > * Thanks to the authors for explaining the "drn". I appreciate the explanation. However, I would like to point out an important thing: the reason for clearer and more exact experiment setups are not for training the best systems. They are for reproducibility. So they do matter. Adding more descriptions of those hyperparameters can help readers understand more about the experiment setting and thus enhancing the reproducibility.
> > > > ## Re Re wordings
> > > > * Ok.
> > > > * It *is* because "bounded by" is a term that is widely used by computational complexity theory that makes me skeptical about this term. I can see from the results that hardness is *associated* with the representation level, but I do not see how they are bounded by representation level. When using the term bounded by, I will expect the paper to provide experiments that "no matter how one varies the parameter of the network, the hardness for transformers to CLM certain language will not exceed some upper or lower bound when using certain representation." What is the upper bound and lower bound the authors stated in the response? Confined by still indicates that the hardness will be restricted in a certain range, but I am not sure what the range is.
> > > > * OK.

---

> > > > > ### Author Response · Authors · 2021-11-18
> > > > > **Re Re: response 3/n**
> > > > >
> > > > > - Re "why not a C0 with seed 13":
> > > > >
> > > > >     A, B, C are 3 datasets. As you know, A is in order (except for lines that were filtered out in parallel due to length). B and C are from 2 different sets of shuffled lines. So A, B, and C are different datasets. The intent of having A0 and B0 with the same seed (13) is to see the effect of "shuffleness", the difference between A and B. C0 and A1 have the same seed 9948. So that is to contrast between A and C, B0 (13) and C1 (13) to contrast between B and C. A0 (13) and A1 (9948) is to contrast seed info for the same data A, B0 (13) and B1 (9948) for B, and C0 (9948) and C1 (13) for C, ... etc.. The purpose of a "C0 (13)" has been subsumed under these combinations. (Sure, theoretically, one can test on infinite combinations of seeds, but we had to make a call as to what is reasonable, also because these are costly experiments.)
> > > > >
> > > > > - Re "drn":
> > > > >
> > > > >     Yes, we understand the importance of reproducibility. Our comment re "not training the best system" was made to forestall your asking why we didn't pick another setting. ICLR is a ML conference, we expect readers to look into the info provided (in this case, the repo for Sockeye) for implementation details. Compared to many existing papers, we are already very exhaustive in reporting setup specifications. (Sidenote: one of the most important criterion for reproducibility that we find most under-supported in our current research landscape is tokenization specifications. So we are glad to have been able to raise awareness in this matter.)
> > > > >
> > > > > - Re "bounded by":
> > > > >
> > > > >     We agree, we did not specify the bounds in this paper. The original intent was to signal to readers that there _is_ such a possbility of refining our ideas, based on more concrete terms and definitions along this line. But that is for future work. So for this paper, for the first occurrence of the word "bounded" in §1.2, we will update 2. as "Hardness in modeling is relative to its representation level (representation relativity).". Is that ok? Are you ok with the second occurrence of the word "bounded" in §5: "this goes to show that linguistic morphology, along with its complexity, as is practiced today and that which has occurred in the NLP discourse thus far, has only been relevant on and is bounded by the “word” level."?

---

> > > > > > ### Comment · Reviewer_UWb7 · 2021-11-18
> > > > > > **Re Re Re: response 3/n**
> > > > > >
> > > > > > I think removing the term bounded by in Section 1 will be more appropriate. And it is also better to remove the term in Section 5 since the paper does not really touch anything about the bounds. This might seem to be an overclaim.

---

> > > > > > > ### Author Response · Authors · 2021-11-18
> > > > > > > **Re Re Re Re: response 3/n**
> > > > > > >
> > > > > > > Re "bound":
> > > > > > >
> > > > > > > To clarify, there are also other sciences that use the word "bound", and we find it can be important to explicitly state a conceptual "bound" for morphology to prevent future hacking around with definitions. As you said, we didn't specify a lower and upper bound, so we didn't claim anything or overclaim. One does not have to specify an upper/lower bound in order to use the word "bound". Would you not agree?
> > > > > > >
> > > > > > > That said, we've revised the sentence in §5, the one previously with the word "bounded by", to:
> > > > > > >
> > > > > > > Considering there are character-internal meaningful units in languages with logographic script such as ZH (cf. Zhang & Komachi (2018)) that are rarely captured, studied, or referred to as “morphemes”, this goes to show that linguistic morphology, along with its complexity, as it is practiced today and that which has occurred in the NLP discourse thus far, has only been relevant on the “word” level, conceptually constrained by unstandardizable units such as “words” (and “sentences”).

---

> > > > ### Author Response · Authors · 2021-11-19
> > > > **response to original review to Reviewer UWb7 (4/n, n=4)**
> > > >
> > > > We will try to finish up with addressing the rest of your initial review here.
> > > >
> > > > Re wording:
> > > >
> > > > - §2, "intrinsic complexities": we study the more fundamental process of CLMing without performing any translation, in order to explicitly focus on modeling the complexities that may or may not be intrinsic to the languages. The idea here is to remove as many confounds as possible, based on our current modeling paradigm for sequences. So we chose an intrinsic evaluation, without translation.
> > > >
> > > >     Re "what is intrinsic to the languages and what is not": as per our conclusion, there is nothing intrinsically/inherently complex to languages in the context of computing (aside from its statistical properties related to length and vocabulary).
> > > >
> > > >     Context: the traditional assumption is that morphological complexity is intrinsic/inherent in languages (doesn't matter what representation it is). One would refer to and identify languages as languages, not languages wrt granularity.
> > > >
> > > > - §2, "other data types": e.g. numeric data. So we performed our experiments with symbolic variants. That is, we wanted to see whether/how results would differ --- just a way to stay as "faithful" to symbolic language data representation on the one hand, and to attain interpretability on the other. Before this set of experiments, it was not clear that pinyin would have the behavioral pattern shown (although one would assume/know it's "easier" to model ZH using pinyin).
> > > >
> > > > - §5, "necessary condition": the sentence now reads:
> > > > "Since it is possible for our character and byte models to effect no performance disparity for the same languages on the same data, a complexity intrinsic to language does not exist."
> > > > (Before: "Considering how it is possible for our character and byte models to effect no performance disparity for the same languages on the same data, this indicates that disparity is not a necessary condition.")
> > > >
> > > > - p.6, Re non-monotonic: "non-monotonic and" now replaced with "highly"
> > > >
> > > > Re misc.:
> > > >
> > > > - As you said, "fairness is mostly used to refer to societal biases". We used it for languages. [Novelty!] In the past, there have been elements of exoticism in some tradition in language sciences (cf. [1], [2], inter alia). So this paper speaks on behalf of these "minority" languages --- at least in the context of computing, there are opportunities to be fairer. Another reading for "fairness in representation" can be understood via the disproportionate interest in character encoding vs grammar.
> > > >
> > > >     [1] Geoffrey K. Pullum. 1991. The Great Eskimo Vocabulary Hoax and Other Irreverent Essays on the Study of Language. University of Chicago Press.
> > > >
> > > >     [2] John H. McWhorter. 2014. The Language Hoax. Oxford University Press.
> > > >
> > > > - Re "But this is often ... and accepted in NLP.": this is the qualitative fairness discussion. To process ZH well with pre-neural methods, one often needs statistical methods (vs "grammatical" ones), unless you segment "words" like EN and restrict its (many possible) readings to one that is aligned with EN.
> > > >
> > > > - Re use of pinyin: this was just to test if ZH in pinyin would behave differently than in other representation. Each ZH character corresponds to multiple ASCII characters, correct.
> > > >
> > > >     Context: We have encountered reviewers (in another context) who wondered whether pinyin should behave completely differently from ZH in characters. This is not an unreasonable expectation. About 20 participants (most are experienced multilingual NLPers) at a multilingual workshop back in 2019 were (informally) asked to match Figures 1abc to char, byte, and word, and none was able to do so with certainty (a couple of them did manage to guess, though wondering what the "hump" (DD) was about). So, we are confident that our results are (still) novel.
> > > >
> > > > - Re one set of hyperparameters: it is right, because it is a relational assessment.
> > > >
> > > > - Re "why does the paper also compare the other three": because it is relational. One would like to see the "space between the lines". There is no such thing as morphological complexity (unless we will it and implement it).
> > > >
> > > > - We restricted results to 300 characters max per line. But the point is that it traces back to data statistics.

---

> > > > > ### Author Response · Authors · 2021-11-19
> > > > > **Re Re: response 2/n**
> > > > >
> > > > > - Re "Comment title: Re: response 2/n (Add some questions about experiment settings)
> > > > >
> > > > >     Comment: I do not quite get what the authors mean by "While it is also possible to compare each data size separately (or other combinations/slices of our data), for this paper, we compared "languages" because we wanted to find out about languages (each in its "totality", so to speak, i.e. all data points for each representation)."
> > > > > If the paper wants to present the result that considers "all data points", then why not just compare the result of the largest data size (10^6) that contains all data points?
> > > > >
> > > > >     > One can see the comparison across sizes as a longitudinal study of some organisms. Language in some tradition of language science treats languages as organisms. *Edit 20211119:* last sentence should read: "In some practice of language sciences, language is studied as an organism.". Addendum: It is useful for the formulation of scaling laws for engineering --- even though our size range is not typical of industry-scale, looking at data when they are small can sometimes tell one a lot about how they'd behave when they are big. Re "why not just compare ... 10^6": because this gives us a bigger picture of how things develop and can also help increase robustness (as we can see with erraticity, robustness can be spotted across size).
> > > > >
> > > > >     I am just wondering if performing the statistical comparison under different data sizes will give us new intuitions on how data sizes might affect the difficulty for CLMing.
> > > > >
> > > > >     I will be more convinced if "under different data sizes, A language and B language have no disparity, then A language and B language have no disparity", instead of "when comparing all the data sizes altogether, A language and B language have no performance disparity".":
> > > > >
> > > > >     > We can look at the differences between languages at each size from the figures currently in the paper for that answer.
> > > > >
> > > > > We hope to have answered all your questions. We would like Reviewer to consider whether it is possible that the copious questions could be an indication that this paper is indeed very novel.
> > > > >
> > > > > Thank you very much for your support.
> > > > >
> > > > > Sincerely,
> > > > >
> > > > > Authors3684

---

> > > > > ### Comment · Reviewer_UWb7 · 2021-11-22
> > > > > **Re 4/n**
> > > > >
> > > > > I still do not think using pinyin to compare is reasonable.
> > > > > Considering the paper uses "unnormalized perplexity score" to represent "how mean bits are needed to encode the corpus using the target language conditioning on source language", then encoding Chinese using pinyin will lose some information of the source corpus since the mapping from pinyin to Chinese characters is one-to-many. Translating a language to Chinese using Pinyin or chrarcter is totally different. ~~ASCII~~, Wubi, and Chinese characters are equivalent since one can convert either one of the three encodings to another one, but when converting pinyin to Chinese characters, one needs to use some language models to find out what is the most suitable character under its context.

---

> > > > > > ### Author Response · Authors · 2021-11-22
> > > > > > **Re Re 4/n**
> > > > > >
> > > > > > Thank you for your question.
> > > > > >
> > > > > > One can say that using pinyin loses the script information. We do not disagree. (And that is also not how most ZH speakers/users identify with their language.) We can make a note re the qualitative difference.
> > > > > >
> > > > > > However, it is not a matter of being "reasonable" or not. The point of the experiment is to see if/how the behavior of such data representation would differ from representations of other languages. Note that this paper aims to take readers to go from the general idea of "language" (i.e. language at large, as in how one conventionally speaks of it, in a very high-level, colloquial way) to a more technical, fine-grained notion of "language" (i.e. wrt specific representation granularity). If there were no experiments on pinyin, one would just think ZH as a whole is some type of an outlier. But in fact, one needs to think more fine-grained and to distinguish the difference between phonetics and the "logographic pattern".
> > > > > >
> > > > > > (Is there a typo in this part of your sentence "ASCII, Wubi, and Chinese characters are equivalent since one can convert either one of the three encodings to another one"? That is, did you mean to include "ASCII"? If so, we don't understand what you mean.)
> > > > > >
> > > > > > Would you object to making our discussions public in general? We think the information from our discussions can be valuable to those interested in curriculum design for NLP/DS. (It's your prerogative.)

---

> > > > > > > ### Comment · Reviewer_UWb7 · 2021-11-22
> > > > > > > **Re Re Re 4/n**
> > > > > > >
> > > > > > > Thanks for the quick reply.
> > > > > > > The ASCII is indeed a typo, I will fix that.
> > > > > > >
> > > > > > > I now can understand why the authors choose pinyin.
> > > > > > > But I still think using pinyin does not only change the "representation level".
> > > > > > > It will be great if the authors add some discussions on that.
> > > > > > >
> > > > > > > I also highly recommend the authors to add more discussions on the concept of language (conventional or technical) and the concept of word (conventional or computational) in the introduction of the paper.
> > > > > > > Since these discussions are scattered in many places in the paper, I think it is better to give some overall ideas on them, allowing the readers to understand why these discussions are relevant to the core problem the paper aims to address.
> > > > > > >
> > > > > > > I will make our discussions public.
> > > > > > >
> > > > > > > And I will **again** upvote the score.

---

> > > > > > > ### Author Response · Authors · 2021-11-22
> > > > > > > **Re: Re Re RE 4/n**
> > > > > > >
> > > > > > > Reviewer:
> > > > > > >
> > > > > > > "Thanks for the quick reply. The ASCII is indeed a typo, I will fix that.
> > > > > > >
> > > > > > > I now can understand why the authors choose pinyin. But I still think using pinyin does not only change the "representation level". It will be great if the authors add some discussions on that.
> > > > > > >
> > > > > > > I also highly recommend the authors to add more discussions on the concept of language (conventional or technical) and the concept of word (conventional or computational) in the introduction of the paper. Since these discussions are scattered in many places in the paper, I think it is better to give some overall ideas on them, allowing the readers to understand why these discussions are relevant to the core problem the paper aims to address.
> > > > > > >
> > > > > > > I will make our discussions public.
> > > > > > >
> > > > > > > And I will again upvote the score."
> > > > > > >
> > > > > > > Authors:
> > > > > > >
> > > > > > > Using pinyin does not change the "representation level" --- correct, pinyin is processed on the character level. On the character level, the reason to experiment with pinyin and Wubi was to remedy the high |V| and short length (as per correlation studies). There seems to be some misunderstanding here and we are not sure what you are trying to get at with "representation level". As we mentioned earlier today, we'll add a remark along the line of --- pinyin as a solution is qualitative dissatisfactory because the script information is lost. We hope that addresses your concern.
> > > > > > >
> > > > > > > Re on the concept of language: yes, we do intend to submit a more extended version of this paper to a journal later. There, we will furnish a more elaborated definition of "language". The sense of "language" we used is standard in NLP literature (i.e. l_machine). **Would you please specify** where you think our discussions are "scattered in many places in the paper"?
> > > > > > >
> > > > > > > Thank you.

---

> > > > > > > > ### Comment · Reviewer_UWb7 · 2021-11-22
> > > > > > > > **Re: Re: Re Re RE 4/n**
> > > > > > > >
> > > > > > > > I am willing to **specify** them.
> > > > > > > > Scattered (The followings are based on the initial submission from the authors. Better organized a overall discussions on "language", "word", "morphology" so as to make the readers more prepared when encountering the following discussions):
> > > > > > > > * Section 3 first paragraph: "Our conventional way of referring to “language” (as a
> > > > > > > > socio-cultural product or with traditional word-based approaches, or even for most multilingual tasks
> > > > > > > > and competitions) is too coarse-grained (see also Fisch et al. (2019) and Ponti et al. (2020))."
> > > > > > > > * Section 3 last paragraph: "Since the notion of word in ZH is highly contested and ambiguous
> > > > > > > > — i) it is often aimed to align with that in other languages so to accommodate academic theories and
> > > > > > > > manual feature engineering6
> > > > > > > > , ii) there is great variation among different conventions, and iii) native
> > > > > > > > ZH speakers identify characters as words — there are reasons to rethink this procedure now that fairer
> > > > > > > > and language-independent processing in finer granularity is possible. Li et al. (2019b) questioned
> > > > > > > > the necessity of CWS in Deep Learning (DL)-based ZH NLP and presented evidence in favor of
> > > > > > > > character-based processing, including results from downstream NLP tasks. In Linguistics, Duanmu
> > > > > > > > (2017) presented a summary on the contested nature of wordhood in (Mandarin) ZH in relation to EN.
> > > > > > > > A more native account of ZH, however, despite a couple of dialects/varieties of it being considered a
> > > > > > > > high-resource language, has not yet been fully recognized and accepted in NLP."
> > > > > > > > * Section 4 **Byte level**: " a complexity that is intrinsic and necessary
> > > > > > > > in language7 does not exist in computing, however diverse they may be, as our 6 are, from the
> > > > > > > > conventional linguistic typological, phylogenetic, historical, or geographical perspectives."
> > > > > > > > * Section 5 **performance disparity**: "A conceptual solution lies in the definition of “words” and morphology."

---

> > > > > > > > > ### Author Response · Authors · 2021-11-22
> > > > > > > > > **Re: Re: Re: Re Re RE 4/n**
> > > > > > > > >
> > > > > > > > > Thank you for your input.
> > > > > > > > >
> > > > > > > > > Re §3: All these sections have references to back up what is being discussed. We'd appreciate it if Reviewer would *please consult the cited work* should you feel there might have been any disconnect in discourse.
> > > > > > > > >
> > > > > > > > > Re §4: We don't understand where the issue here is with the statement on byte level. All those aspects are indeed not relevant for our computation. Our intent is to make that clear.
> > > > > > > > >
> > > > > > > > > Re §5: Please specify where your concern lies here.
> > > > > > > > >
> > > > > > > > > Thank you.

---

> ### Author Response · Authors · 2021-11-22
> **Re "Re: response 2/n: (Continued) Re Re differences to Mielke et al. (2019)":**
>
> Reviewer:
>
> "I would like to know more about why the authors think my interpretation is incorrect? Since the unnormalized perplexity scores for different sources are quite similar when the target language is the same, as shown in the figures in the paper, then the interpretation that "the difficulty of modeling a certain target language is independent of the source language" is not wrong. I did not conclude that source language has no effect on the target, I conclude that the hardness (in terms of perplexity score) is independent of the source language when the target is fixed. I will appreciate it if the authors can elaborate on why you think my interpretation of this is wrong. I think this also replies to the 2. in the 1/n response. I would also point out that "no statistically significant difference" sounds "equally hard" to the readers. And since the authors use the term "equally hard" in the main research question in Section1.2, I also abide by this term in the summary."
>
> Authors:
>
> Re "incorrect": because there is a difference between claiming that "two things are equal" and "there is no statistically significant difference between two things". The latter (our formulation) is a more faithful statement based on what we calculated, evaluated, and reported in this paper. We don't want to appear to have over-claimed. Likewise, with point 2 in response 1/n: "no statistically significant differences" is not the same as "equally (hard)".
>
> Originally, we had thought using the term as in the abstract and introduction so to give readers a general idea would be ok. That's why when we defined our method in §2, we made sure to state explicitly that "we consider there to be “disparity” or “inequality” between languages l_1 and l_2 if there are significant differences between the performance distributions of these two languages". When it comes to drawing a conclusion, we find that it'd be better to stay closer to a formulation that reflects what we did in our experiments. (Besides, previous work also used the term "equally".)
>
> That said, we think there is room to be more technical and set a higher bar. We have updated the research question to "Are there any statistically significant differences in hardness when it comes to Conditional-Language-Modeling (CLMing) languages which have been traditionally consideredmorphologically rich (AR and RU) and poor (ZH) with the 6-layer Transformer?" --- please let us know if you're ok with this or you'd prefer the shorter formulation instead.
>
> Thank you.

---

> ### Author Response · Authors · 2021-11-22
> **Re "Re: response 2/n":**
>
> Reviewer:
>
> "Re Re Novelty
> I need to say that the novelties the authors listed are mostly stated in the strengths in my main review. But I will still respond to the bullet points in a more detailed way. The "not novelty enough" I stated in the main review is "not novel enough comparing to Mielke et al. (2019)". That is why I listed the contribution and comparison of this paper and the prior work in my main review. And I think that I have also fairly summarized the difference between the prior work and this paper. Detailed responses are as follows:
>
> 1. I think this is already included in the strengths in the main review.
>
> 2. I am not sure what the experiments on different data sizes tell me. Can the authors provide insights on what the readers might learn when seeing these visualizations? (Aside from DD and erraticity, refer to the last two bullet points for details)
>
> 3. Are the authors trying to indicate that morphological complexity is not relevant for computation? If yes, then I have addressed this in the last strength in my main review.
>
> 4. I think this is also addressed in the strengths of the main review, while I still am skeptical about the experiment on Pingin.
>
> 5. What does the "adaptation" mean in the authors' comment? And why is this a "fairness metric"? In fact, I have great difficulty understanding the word fairness in this paper, since this term is not explicitly defined in the paper. I would appreciate it if the authors can elaborate on the concept of "fairness" they try to discuss in the paper.
>
> 6. I personally do not find this as a novelty. To make sure I am not misunderstanding the authors, are you trying to state that the "new light" is "unnormalized perplexity score means the number of bits to encode the target corpora (given the source)"?
>
> 7. I think the CLM part is addressed in my main review. And as I stated in the 2nd bullet point, I am not sure what dataset sizes tell me.
>
> 8. I will add this when I update the score after the discussion has ended.
>
> 9. I do not find DD and erraticity as the novelties of this paper, as the two observations are not highly related to the core research question of the paper. They are interesting observations, but I am not sure whether it is a good idea to include them in the main text. They are "orthogonal to our main research questions", as stated in Section1.2.
>
> 10. Same as above."
>
> Authors:
>
> We would like to bring Reviewer's attention to the reviewer guidelines for ICLR re novelty:
>
> As per ICLR Reviewer Guidelines [1]:
>
> "2. Be precise and concrete. For example, **include references to back up any claims, especially claims about novelty and prior work**.
>
> 5. Don’t reject a paper just because you don’t find it “interesting”. This should not be a criterion at all for accepting/rejecting a paper. The research community is so big that somebody will find some value in the paper (maybe even a few years down the road), even if you don’t see it right now."
>
> Hence, we would appreciate it if you would please provide **specific prior literature** on at least the 10 aspects +1 (on applying fairness for language) we enumerated.
>
> Additionally, to further the points you addressed:
>
> Re 2.: From the visualizations, one can see that there are more representations possible than "words". They tell us about the behavior patterns and the disparity tendencies of different granularities in the size range studied with the 6-layer Tranformer.
>
> Re 6.: it is not the act of dev set evalution that is novel, but the interpretation of seeing it as a way to be accommodating to the segmentation problem with multilingual data. E.g. there are languages with no "sentence"-ending markers. Evaluating with a larger span of text can be fairer.
>
> Re 9 & 10: We are concerned and not comfortable with how you seem to insist on comparing our work with one reference, claiming that it's not novel enough, then proceed with ignoring other novelties. They have the potential to set off new research paths. Most/all papers nowadays on DD, they mostly bypass the examination of data types. (If you found any counter-examples, please do let us know. Otherwise, our work is very novel in this regard.)
>
> 20211122: Addendum Re 5.: adaptation --> application.
>
> [1] https://iclr.cc/Conferences/2022/ReviewerGuide
>
> Thank you.

---

> > ### Comment · Reviewer_UWb7 · 2021-11-22
> > **Re Re "Re: response 2/n":**
> >
> > I want to emphasize that the reason I compare this paper to a specific prior work is because I find the previous works and this paper are highly relevant, and I do not think the authors really address too much on what the related works have done in this paper.
> > I feel sorry to make the author feel not comfortable about how I compare their work with the previous work.
> > But I believe my evaluation is fair, and I even listed a thorough comparison of this paper and the prior work in my main review.
> >
> > I also do not quite understand why the authors mention the review guideline, as I see myself far from violating the listed guidelines.
> > Specifically, regarding guideline 5, **my original review did not reject the paper because it is not interesting**.
> > I believe none of my initial reviews include any subjective comment on whether I find this work interesting or not.
> > And regarding guideline 2, each of my claim in the initial review is supported by the content of the paper, and I also clearly stated why I do not find this work being novel enough.
> > The claim of "not novel" is supported by the related work I summarized.
> > None of those concerns or weaknesses have to do with whether the paper is interesting or not.
> >
> > I also do not think there is a problem with only comparing with a specific prior work since I find it to be highly relevant.
> > I have to say that "novelty" can be quite subjective, and for me, novelty is not "we have done something that has not been done", it should be rather "we have done something that has not been done **and that lead us to gain insights on the topic**".
> > While this paper does present many novel and interesting experiments, I do not think the insights are ample enough.
> > I understand that the research problem the authors aim to tackle is important, and the paper does provide some insights/interpretations, but I find the insights provided mostly too high-level and overstating.
> >
> > I need to say I am frustrated to see the authors accusing me of "ignoring their novelties".
> > I believe I have already stated clearly my concerns on those claims of novelties in the previous thread and address each "novelty" claimed by the authors, I am not sure what am I ignoring.
> > The authors have insisted on me to provide **specific prior literature** if their "novelties" have been presented in prior work again.
> > In my previous response, I have ***already*** acknowledge the novelties of some of the listed bullet points, so I do not quite understand why should I provide **specific prior literature** on those novelties.
> > I have already included them in the strengths of the paper in the initial review, and I have also updated my overall recommendation today, shortly before the authors' previous reply.
> > As for the other "novelties" listed by the authors that I do not find them as "novelties", it is not because they have been reported in prior works.
> > Rather, I think they are not novelties because some might not be highly relevant to the core research questions of the paper, and others might not be insightful enough to be considered as novelties.
> > Instead of "ignoring", I am just "disagreeing" with the authors claiming that those are novelties of the paper.
> > Just because I disagree with the viewpoints of the authors on their novelties does not mean I am "ignoring".
> >
> > Re Re 9 & 10: I do not think the authors' comment on this is relevant to what I said in the review. The reason that I don't find DD and erraticity as "novelties" is because I do not know how to interpret the result, and I do not think the two observations are highly related to the research question the paper aims to answer.
> >
> >
> > Re 6: I do not find any relevant discussions in the paper on "but the interpretation of seeing it as a way to be accommodating to the segmentation problem with multilingual data. E.g. there are languages with no "sentence"-ending markers. Evaluating with a larger span of text can be fairer."
> > The paper only talks about "evaluating data that has not been or cannot be perfectly segmented or aligned line by line".
> > It is hard to expect the readers to understand the interpretation of the authors' while much is not elaborated in the paper.
> > Maybe it is better to state clearly about the interpretation for the readers to consider it as a novelty.
> > And which specific language does not have sentence-ending markers among the six languages evaluated in the paper?
> > Which specific language cannot be perfectly segmented?
> > Maybe giving more details can make the readers better understand why this is an important act.

---

> > > ### Author Response · Authors · 2021-11-22
> > > **Re Re Re "Re: response 2/n":**
> > >
> > > Re 6: e.g. ZH. Please refer to work cited.
> > >
> > > There are varieties of ZH [1] and other languages such as Thai with no overt sentence-ending markers [2].
> > >
> > > [1] This refers to our discussion in §3: "A more native account of ZH, however, despite a couple of dialects/varieties of it being considered a
> > > high-resource language...".
> > >
> > > [2] We'd assume reviewers, esp. those rating with high confidence about their assessment, to be knowledgeable about these basics in Multilinguality / Multilingual NLP.

---

### Official Review · Reviewer_86m6 · 2021-11-02

**Correctness:** 3
**Technical Novelty And Significance:** 2
**Empirical Novelty And Significance:** 3
**Recommendation:** 8
**Confidence:** 4

**Main Review:**

This paper contains a series of sound, controlled experiments to study the role of input representations in language-modeling. I think the experiments are well designed, and with a clear focus to answer the research questions laid out in 1.2. Overall, I do think that this paper will benefit the multilingual NLP community in thinking carefully about the choice of input representations, rather than advocating for a one-choice fits all solution. However, I do have a few concerns that prevent me from giving a higher score.

1. The claim that this paper "renders a decade-long debate on morphology irrelevant" is made multiple times in this paper, and in my opinion, is too strong a claim to be made on the basis of experiments on a single task (conditional LM), on a handful of languages (however typologically diverse they may be), and using a single architecture (6-layer Transformer). I would recommend scoping this claim down to something that can be clearly supported from the results of this paper.

2. Second, how does the practical recommendation in Section 3 ("A practical takeaway from this set of experiments: in order to obtain more robust training results, use bytes for ZH and characters for AR and RU") square with the multilingual nature of recent models which typically have a shared input representation for many languages? I think more generally this point is one of several that seems slightly disconnected from downstream applications (the claim about morphological complexity being irrelevant would also fall in this bucket.).

3. There are a few citations from prior work in NLP that are worth contextualizing here. Specifically, the work of Gillick at al. on byte level processing for sequence tagging [1], that of Wang et al. on byte-level Neural MT, and more recently, the work by Clark et al. on building a tokenization-free encoder seem particularly relevant [3]. In fact, the whole body of literature on character-level neural models are particularly relevant and the omission of these is critical, especially given the stance of the paper in advocating moving beyond "word" level models (e.g. Footnote 8)

[1] - https://arxiv.org/pdf/1512.00103
[2] - https://arxiv.org/abs/1909.03341
[3] - https://arxiv.org/abs/2103.06874

**Summary Of The Paper:**

At its core, this paper focuses on testing whether languages with different morphological complexity equally challenging to language-model in a bilingual / MT-like setting using a 6 layer Transformer architecture.  Specifically, it tests the assumption whether the previous observations about languages with higher morphological complexity are difficult to language-model, using a controlled experiment with a n-way parallel corpus across 6 languages of interest. The key finding of this paper is a simple one : that the choice of input representation is highly important, given everything else is kept constant, and that differences in language-model perplexities can be normalized and made to disappear by choosing the same representation (word/character/byte)

**Summary Of The Review:**

This paper contains sound and controlled experiments that test the role of input representation in bringing disparity to language modeling results, which I believe will be useful for the larger community. At the same time, several claims in the paper are too strong or not grounded in practical use-cases, and some relevant literature is not referred to, leading me to be marginally inclined to accept this paper.

### Edit - 22 November, 2021
I spent some time reading the discussion between the authors and the other reviewers, as well as revisiting v0.2 of the paper. In light of these discussions, I think I am fairly comfortable accepting this paper as my main concerns have been addressed by the review and the comments of the authors. Hence, I revised my overall recommendation to 8, while adjusting a few other scoring parameters too.

---

> ### Author Response · Authors · 2021-11-13
> **first response to Reviewer 86m6**
>
> Dear Reviewer 86m6:
>
> We thank you for your review. We value your input.
>
> Re 1:
>
> We wrote "renders a decades-long debate on morphological *complexity* ... irrelevant" --- "unless it is being intentionally modeled in a word-based, meaning-driven context". We find only one occurrence of this statement in both v0.1 and v0.2 of the paper.
>
> We made our formulation more explicit and specific in version 0.2 (in §1.2 and last paragraph of §5). Would you please let us know if you are satisfied with that?
>
> Re 2 & 3:
>
> The primary intent of this paper is "to bridge research in NNs/DL, language sciences, and language engineering through a data-centric perspective" (§6). There is not (yet!) a "more fine-grained statistical science of language or language data". And our (longer-term) vision is to establish this perspective/concentration within data science, in a way that it connects with but is not necessarily focused on downstream task results (because there can be much to do already with both *data* and *science*). Hence, we had a relatively nominal mentioning of the related work in application (with Li et al. (2019a) for ZH and Lee et al. (2017)) --- Li et al. (2019a) handled ZH data and Lee et al. (2017) RU.
>
> Hence, by "practical recommendation", we just meant this can be a way of thinking about data representation. It does not aim to "square with the multilingual nature of recent models which typically have a shared input representation for many languages". That would be a different set of experiments outside the scope of this present paper.
>
> Existing papers in multilingual NLP are mostly algorithm-centric and focus on downstream task results, much like the 3 papers (Gillick et al. (2016), Wang et al. (2019), and Clark et al. (2021)) you mentioned. The focus of this paper is to investigate something more intrinsic, as a form of use-inspired basic research, so to build a firmer foundation for processing multilingual data that would be more robust for statistical analyses. That is, we are evaluating data wrt algorithm, trying to answer a question about the data.
>
> Imagine if this study hadn't been performed, morphological complexity (sometimes generalized to "language complexity") had not been decomposed into characters and bytes, and disparity hadn't been compared, analyzed, and explained, an NLP practitioner could potentially think just because they are processing, e.g., RU with the Transformer, they ought to have lower performance, when it could be the fact that they are just dealing with longer sequences and the disparity can be remedied. So we want to debunk this myth over hardness.
>
> We're not sure if you still, given our responses thus far to other authors, have concerns about "the claim about morphological complexity being irrelevant". Would you please elaborate on your remark "I think more generally this point is one of several that seems slightly disconnected from downstream applications (the claim about morphological complexity being irrelevant would also fall in this bucket.)"? Is the remark still current?
>
> This is not (or not just) an application paper about tokenization or input representation. (It _could be_ for someone who is only focused on the engineering aspect of things, but it is more than that.) It is also a paper that analyzes the nature of multilingual data and evaluates it qualitatively and quantitatively wrt the Transformer configuration, so to provide answers and insights for the language sciences (e.g. cognitive modeling, comparative linguistics) from a computational perspective (one of our keywords is "science in the era of AI/DL (AIxScience)"). That's also why it is important.
>
> We do value your input for the suggested literature and will incorporate them in the background chapter of the extended version of the paper.
>
> We thank you in advance for your reply. We hope our responses and edits help convince you of the merits of this paper.
> Please also let us know if there is anything else that we can revise.
>
> Sincerely,
>
> Authors3684

---

> > ### Comment · Reviewer_86m6 · 2021-11-22
> > **Thanks for the updates.**
> >
> > I spent some time reading the discussion between the authors and the other reviewers, as well as revisiting v0.2 of the paper. In light of these discussions, I think I am fairly comfortable accepting this paper as my main concerns have been addressed by the review and the comments of the authors.

---

### Official Review · Reviewer_kpoN · 2021-11-02

**Correctness:** 3
**Technical Novelty And Significance:** 4
**Empirical Novelty And Significance:** 4
**Recommendation:** 8
**Confidence:** 3

**Main Review:**

Pros:
— Very interesting and engaging work, with
--- Experiments are very controlled, from data, number of lines, language mappings, and segmentations.
— Experiments seem to be reproducible just from the description in the Experimental Setup. Very clear.
— Thoughtful connections from human language processing and linguistics to try to explain the Char/Byte/Word performance disparities.
--- Really interesting follow up experiments on ZH!

Cons:
— The “A0, B0, C0, A1 & A2, A3-7”, etc. nomenclature for experiments is unhelpful to readers. Can the experiment names be changed to be something more intuitive, so that readers don’t need to memorize “oh, A0 is one of the controlled experiments for data size”?
-- Figures in this paper are largely unreadable and do not help as visual aids. I have to zoom in to 310% (at minimum!!!), and even then my eyes struggle to distinguish the colours and line styles from each other. Also, the subfigures within subfigures are not readable. This is a big problem, because the figures are used to support claims that the authors are trying to make, and I think many readers would be unable to draw the same conclusions from these figures themselves.


Formatting, grammar, etc.:
1. Abstract: “Performance disparity is not a necessary condition” is worded strangely to me.
2. Introduction: Formatting at bottom of 1.1 is weird. Use of bullet points makes no sense here, they just split up the sentence.
3. S2, “Experimental setup“: There’s a double parenthesis )) with the (Jurafsky & Martin, 2009) citation
4. S5, your tables do not follow the same typesetting conventions as your Figures (i.e., the description is on top)


**Summary Of The Paper:**

In this paper, authors aim to untangle the disparities in performance between morphologically rich languages (e.g. RU) and morphologically poor languages (e.g. ZH). In order to untangle such disparities, the authors propose a highly controlled experiment setup, where they control for a number of potentially confounding factors (e.g. language mappings, data size, text encoding, etc.). After the controlled experiments, the main take-away of the paper is that vocab size seems to be the biggest indicator of how well or poorly a CLM will perform — since there can be several different granularities for the concept of a “word” for morphologically rich languages, and this can explain why modelling morphologcially rich languages can be more difficult.  They perform additional experiments with different representations of ZH, and were able to mimic the struggles of morphologically rich languages with ZH, and use this as further evidence to support the claim that the debate within NLP about the best segmentation is irrelevant.

Overall, this work is exciting, has has a lot of potential to help inform future language modelling for morphologically rich languages.  The very controlled nature of the experiment in the paper is what makes it most convincing, and this is the paper’s largest contribution.

**Summary Of The Review:**

Reasons for score:
Frankly, if the figures and tables in this paper were significantly improved, this would be an 8 or higher. The main motivation of the paper is exciting, the methods are well controlled, and the findings are interesting. The core of the problem (as I discuss in "Cons") is that the claims made in the paper rely on the figures/tables for illustration as evidence, and the figures/tables are entirely unhelpful. In otherwords, the tables and figures detract from your arguments because they are unreadable. Once this can be fixed, I think its great work, and it would be interesting and directly useful to many people in NLP.

EDIT - Wednesday, 17 November, 2021: In light of the authors rebuttal to my review, and their changes to v2 of the document, I am bumping my score up from 5 (marginally below threshold) to 8 (good paper, accept). My biggest problem with the paper before, were the figures, but I think that the authors addressed my comments to the best of their abilities by adding a lot more clarity to the textual descriptions of the images, in the paper and in the captions. The figures are still not great, but with their new textual descriptions, you can look at the general shapes of things, and visually understand the point, and before v2 I did not have this same level of visual understanding.

---

> ### Author Response · Authors · 2021-11-13
> **first response to Reviewer kpoN**
>
> Dear Reviewer kpoN:
>
> Thank you very much for your motivating feedback. We are glad that you find this work exciting and we thank you for your appreciation for science and systematic experimentation.
>
> Re the nomenclature of runs ("A0, B0, C0, A1 & A2, A3-7", etc.): we decided to get rid of the mentioning of these in the main text altogether since they are not being referred to in the main text again. The information is primarily for documentation purposes. We retained it in App. A and extended it a bit. Thank you for pointing out that it might cause confusion.
>
> Re size of figures: please note comment above with v0.2 upload. (TLDR: One can just look at the spaces between sets of lines from target languages, if one wants to look at the figures, that is. Otherwise, Table 1 sums it all up. Enlarged figures are also available in App. E.)
>
> Re:
>
> 1. Abstract: "Performance disparity is not a necessary condition" --> "On the character and byte levels, we are able to eliminate statistically significant performance disparity, hence demonstrating that a language cannot be intrinsically hard."
>
> 2. §1.1:
> before: "Since: i) different input representations have been tested with different architectures with divergent results in different metrics; ii) we noticed a discrepancy in the results from Mielke et al. (2019) for ZH --— it came out as the least difficult for the character model, but it is the 6th most difficult language for the BPE model; and iii) each of the previous studies only tested with one data size, without statistical comparisons of score distributions between languages; we decided to investigate the matter more systematically once again." --> after: "We noticed, however, a discrepancy in the results from Mielke et al. (2019) for ZH — it came out as the least difficult for the character model, but it is the 6th most difficult language for the BPE model. As different input representations have been tested with different architectures with divergent results in different metrics in previous studies, each of them only testing with one data size, we decided to investigate the matter more systematically once again with statistical comparisons of score distributions between languages."
>
> 3. §2: The "double closing parentheses" are right, the whole unit goes like: ... (open vocabulary on dev (Jurafsky & Martin, 2009)).
>
> 4. §5: typesetting conventions are different between figures and tables for ICLR. We followed the ICLR style guide:  https://github.com/ICLR/Master-Template/raw/master/archive/iclr2022.zip from https://iclr.cc/Conferences/2022/CallForPapers
>
> We uploaded v0.2. Would you mind letting us know if it looks better to you? And is there anything that we could do to improve even though we couldn't do much re figures and tables?
>
> Thank you for your support.
>
> Sincerely,
>
> Authors3684

---

> > ### Comment · Reviewer_kpoN · 2021-11-17
> > **response to authors rebuttal**
> >
> > Hello authors - Thanks for your very thorough reply to my review, and your attention to detail.
> >
> > Small formatting and grammar fixes look great. (Also, sorry if it was unclear in my first review, I did not mean that these formatting/grammar things were a “Con” for your paper.) Thanks for addressing these, and my apologies for the confusion with the ICLR style. Looks good, on that front then.
> >
> >   For the figures, after looking at v2 of the paper, I'm much more satisfied with the text descriptions now, even if the figures cannot easily be adjusted. Thank you for making this more clear.

---

### Official Review · Reviewer_y4Vs · 2021-11-02

**Correctness:** 3
**Technical Novelty And Significance:** 3
**Empirical Novelty And Significance:** 3
**Recommendation:** 8
**Confidence:** 4

**Main Review:**

Strengths:
- Very nice writeup and structure, the paper is for the most part a very enjoyable read.
- Bold in tackling important questions from a fresh angle.

Weaknesses (may be obsolete, see discussion):
- The contributions are overstated in light of the experiment setup.
- Only six resource-rich languages are used and only one respective dataset, the UN parallel corpus.

**Summary Of The Paper:**

The paper asks an interesting question, one that appeared eternally relevant for NLP: Are languages harder or easier to contextually language-model by Transformer in light of their varying morphological complexity? The claimed main finding is provocative, indicating that the debate morphological variation is not relevant in the context of the given computational machinery.

**Summary Of The Review:**

I recommend that the paper be rejected so that it may further develop its empirical setup & verify (or reject) the findings on a larger dataset with more substantial linguistic depth (more resource variation) and breadth (more languages). In its current form, the paper's larger-than-life claims are simply not sufficiently supported with the choice of one dataset and six resource-rich languages, even if the hypothesis and findings are indeed interesting.

Edit after first round of author discussion, 2021-11-16:

I am concerned that there is a disconnect between the stated contributions of the paper (which are expressed as rather broad and general in its narrative) on one side, and the setup of the empirical investigation on the other side. While I may view the insight as to the (ease of) availability of broad-coverage linguistic resources sympathetically, the fact still remains that the findings are closely tied with the UN parallel corpus and its six languages. This in itself would not disqualify the paper at face value. However, text e.g. under "Summary of findings" 1-5 and later in "Conclusion" comes across as masking this fact and painting a much broader picture.

I would be much more inclined to accept this (undoubtedly insightful) paper should it insert a clear Limitations section which would reflect on the preliminary nature of these findings given the dataset constraints.

As for the language resources, I would very much love to hear which of those e.g. in OPUS (https://opus.nlpl.eu/index.php) were not fitted to the experiment while only the UN corpus passed the filter. This would be an insightful addition to the limitations section, as the author response now seems to indicate massive limitations in all the other available datasets. Do share with the community!

Edit after 2021-11-18:

Upvoting the overall score after insight re: multi-parallel datasets available (see discussion).

---

> ### Author Response · Authors · 2021-11-10
> **first response to your review (1/2)**
>
> Dear Reviewer y4Vs:
>
> Thank you very much for your feedback. And thank you for acknowledging as well that this is an important and perennial question that has plagued NLP for as long as the field has existed. We are glad that you enjoyed our write-up and would be glad if you'd allow us to clarify some potential misunderstandings so to win your support.
>
> First of all, you wrote "[t]he contributions are overstated in light of the experiment setup", would you mind clarifying which contribution you are referring to specifically?
> Is it possible that you thought that our paper here claims that there is no variability in language, or at least among the languages tested here?
> To clarify, we are not saying that languages are all the same in the context of computing or in CLMing with the 6-layer Transformer. We are saying that the variability can be expressed in terms of differences in vocabulary and length in characters and/or bytes. We showed empirically that these differences can be eliminated within the setup of CLMing in a 6-layer Transformer (the empirical solution restricts to this specific setup, up to 300 characters max per line along with all other hyperparameters listed in the appendices).
>
> The analytic and conceptual explanations are the more compelling reasons why we were able to categorically claim what we did wrt morphological complexity with only these six languages and one dataset. From our experimental results, we were able to trace back to data statistics in length and vocabulary with our correlational study which leads to our analytic solution. The disparity in word-based representations is not necessary because we don't have to word-tokenize, i.e. disregard and add in whitespaces. But the mere practice of doing so always creates a "hierarchy" and shifts the distribution. That's why the claim holds. Conceptually speaking, the definition of "words" is unstable and indeterminate. So there can be no other segmentation scheme involving words or subwords that could be optimal enough to ever "solve" the vocabulary issue. In fact, for e.g. CWS, units are more aligned with the structure of EN words and grammar. Also, as there are sub-character units that are "word"-like or meaning-bearing, saying that CWS should stop at the "EN-centric" word level seems insufficient. But doing it otherwise could break the morphological hierarchy in that ZH can no longer be considered morphologically frugal.
>
> Re "larger dataset": this is a controlled scientific experiment specifically aimed to evaluate the necessity of morphological complexity in languages traditionally considered rich and poor. We need good parallel raw texts to perform evaluation. But the UN dataset is, to the best of our knowledge, the *only reliable dataset* with more than 1k (or 10k/100k/1mio) lines for each language (which enables our comparison of score distributions across several magnitudes of data sizes). The Bible has already been used by Mielke et al. (2019). Europarl does not have ZH. The UDHR dataset (by the Unicode Consortium) only has a few hundred lines for each variety (this was also mentioned in Mielke et al. (2019)). Other datasets are either not fully parallel or have been word-tokenized (i.e. length information has been compromised). We agree that in a real world situation, many other factors will come into play, e.g. data availability/scarcity. But that is outside the scope of our present study.
> Re language choice: according to [1], there is a bimodal distribution between logographic and phonetic languages. So testing ZH as a logographic language as well as a language traditionally considered morphologically poor seems apt. Profiles of data statistics from different datasets are consistent.
>
> We do hope, however, that there would be more parallel data available to enable more evaluation in the future. This would also help better ground our default expectation of performance involving language data. We hope our paper could be viewed as having made a positive step forward in promoting more fairness and objectivity in evaluating multilingual data. A practical message of our work can be:
>
> If/When the intent is _not_ to explicitly model linguistic morphology in computing, one can simply describe languages and their statistical profiles with respect to their representational granularity in characters or bytes, or refer to sequences as longer/shorter or having a higher/lower vocabulary size when comparing them with each other, rather than richer/poorer based on the concept "word" that can be ambiguous, contested, and inaccessible to many (esp. speakers who are unfamiliar with EN). This matters much in multilinguality and a "word"-free interpretation would also help make the analyses of language data more objective and fair to a wider audience. Diversity in NLP matters and we hope that you'd agree and support our paper. (One can think of this initiative as "proposing a more politically correct way of characterizing languages".)

---

> > ### Author Response · Authors · 2021-11-10
> > **first response to your review (2/2)**
> >
> > If you are satisfied with our reply, we'd appreciate it if you'd please adjust your score to reflect your support. If anything needs further clarification, or if you'd like to see any particular part of our explanation stated more explicitly in the paper (pls specify in place of what other information, as we are already at page 9, the page limit), please do not hesitate to let us know as soon as possible. We'd be happy to comply.
> >
> > Thank you very much.
> >
> > [1] https://openreview.net/pdf?id=yySHIamiadH

---

> > ### Comment · Reviewer_y4Vs · 2021-11-16
> > **Maintaining my previous assessment re: experiment limitations**
> >
> > Dear authors!
> >
> > Thank you for the extensive follow-up here under my review and elsewhere in the paper thread; it is informative and much appreciated.
> >
> > I am still concerned that there is a disconnect between the stated contributions of the paper (which are expressed as rather broad and general in its narrative) on one side, and the setup of your empirical investigation on the other side. While I may view your insight as to the (ease of) availability of broad-coverage linguistic resources sympathetically, the fact still remains that your findings are closely tied with the UN parallel corpus and its six languages. This in itself would not disqualify the paper at face value. However, text e.g. under "Summary of findings" 1-5 and later in "Conclusion" comes across as masking this fact and painting a much broader picture.
> >
> > I would be much more inclined to accept this (undoubtedly insightful) paper should it insert a clear Limitations section which would reflect on the preliminary nature of these findings given the dataset constraints.
> >
> > As for the language resources, I would very much love to hear which of those e.g. in OPUS (https://opus.nlpl.eu/index.php) were not fitted to your experiment while only the UN corpus passed your filter. This would be an insightful addition to your limitations section, as your response now seems to indicate massive limitations in all the other available datasets. Do share with the community!
> >
> > Meanwhile, awaiting your response, I shall keep my original assessment for the most part; upvoting the originality & overall assessment scores by one rank. Thank you!

---

> > > ### Author Response · Authors · 2021-11-16
> > > **Request for specification re claim and answer re OPUS inventory**
> > >
> > > Dear Reviewer y4Vs:
> > >
> > > Thanks very much for your being inclined to accept this paper.
> > >
> > > We will add "and current multitext availability" to our last paragraph under "Performance disparity" in §5:
> > >
> > > "Target language pairs with significant differences are summarized in Table 2.  We show that morphological complexity can be empirically eliminated in this one-setting-for-all configuration with a 6-layer network, no hyperparameter tuning, and a maximum line length of 300 characters (and its corresponding equivalence in other representations) as constrained by our hardware and compute time listed in App. A *and current multitext availability*. A more analytical ..."
> > >
> > > Question: would you please specify which claim/statement/wording you're concerned about?
> > >
> > > The only edge case we can think of is if a language does not have an electronic dataset, then it's not included. Otherwise, everything can be represented by bytes, and we limited our empirical claim to 300 characters (and its equivalence in other representations, e.g. bytes) max per line. We assume for the entirety of this paper (as customary in NLP) that we are talking about textual data. Please let us know if you think we missed anything. But the main point re morphological complexity is that, it is a word-level specific phenomenon. So as long as we do not word-tokenize, it cannot be there. (Better yet, if we don't think of languages that way, it also won't be there!) Surely, there can be sequences that are longer than what we examined, but that's a length issue, not morphological complexity.
> > >
> > > Our primary intent is not so much to ease the lack of broad-coverage linguistic resources, but more to disabuse the community of the idea of morphological complexity as a concept that one needs to harbor when processing language. (One can think of it as just a more objective and politically correct way of speaking and thinking about the world.) (Appropriate) data collection is important for two reasons: 1) even if we do not need more data to train systems for application/engineering purposes, there is a cultural aspect to documentation that is good for most humankind (if it is the case that the communities would like to have their data documented and/or preserved); 2) multitexts can enhance ML evaluation as well as support systematic basic research. As disciplines are converging (e.g. language sciences and technology) and ML, we are trying to find ways that would bring about a win-win situation for all parties concerned (footnote 8).
> > >
> > > Re OPUS:
> > >
> > > There are many bitexts, but for multitexts (multiway parallel data) with the same amount of data for each of our 6 languages (i.e. truly parallel), there is just one on OPUS: the UN v20090831 with 74.1k sentences each. Below is a list of multitexts based on what's available for "zh (Chinese)" and "ru (Russian)", and we filled in also the number of sentences available between zh and the other languages we used:
> > >
> > > ZH (Chinese) - RU	AR	FR	ES	EN
> > >
> > > UNPC v1.0				14.8M	14.6M	14.6M	14.6M	14.2M
> > >
> > > WikiMatrix v1			1.3M	0.6M	0.6M	0.6M	2.6M
> > >
> > > MultiUN v1				8.2M	8.4M	8.4M	8.4M	8.0M
> > >
> > > CCMatrix v1				14.0M	6.6M	6.6M	6.6M	71.4M
> > >
> > > wikimedia v20210402		0.6k	33		33		33		0.2M
> > >
> > > Tanzil v1				0.1M	24.9k	24.9k	24.9k	0.2M
> > >
> > > XLEnt v1.1				1.1M	12.1k	12.1k	12.1k	6.0M
> > >
> > > UN v20090831			74.1k	74.1k	74.1k	74.1k	74.1k
> > >
> > > bible-uedin v1			61.8k	61.7k	61.7k	61.7k	0.1M
> > >
> > > News-Commentary v16		47.7k	66.0k	66k		66k		69.0k
> > >
> > > QED v2.0a				14.0k	20.7k	20.7k	20.7k	20.8k
> > >
> > > TED2020 v1				16.2k	16.0k	16.0k	16k		16.2k
> > >
> > > PHP v1					39.0k	-		-		-		45.0k
> > >
> > > infopankki v1			8.7k	8.9k	8.9k	8.9k	8.9k
> > >
> > > tico-19 v2020-10-28		3.1k	3.1k	3.1k	3.1k	3.1k
> > >
> > > EUbookshop v2			0		0		0		0		0
> > >
> > > TLDR: they are not all truly parallel.
> > >
> > > We will post this table either in our appendix or in the Ethics Statement section. We don't think we can fit in a Limitations section before the References this year?
> > >
> > > Please let us know if this would suffice.
> > >
> > > Thank you again for your support.

---

> > > > ### Comment · Reviewer_y4Vs · 2021-11-18
> > > > **Nice follow-up on parallel resources**
> > > >
> > > > Thank you for the insightful response! It would indeed suffice that you address the multi vs. bitext topic in the appendix building on this discussion. Regarding wording, just as long as it's always clear throughout the paper that these insights are from the UN corpus only, and there are no implications otherwise without empirical evidence, I'd appreciate it. I shall be upvoting the overall recommendation after this discussion.

---

> > > > > ### Author Response · Authors · 2021-11-19
> > > > > **re Nice follow-up on parallel resources**
> > > > >
> > > > > Thanks for your feedback.
> > > > >
> > > > > Re "there are no implications otherwise without empirical evidence": we can move "In our bilingual (one-to-one) CLMing setup with the Transformer, we find:" out of the enumeration §1.2, i.e. immediately following the header "Summary of findings and insights".
> > > > >
> > > > > We did perform correlational study and verify it with our alternate representations to come to the conclusion that hardness has to do with vocab and length. This way, we are "explaining away" morphological complexity and decomposing it. What do you mean by "no implications otherwise"?
> > > > >
> > > > > (And to confirm, you don't require a table with data info from OPUS as from our reply to you on 16 Nov 2021?)

---

> > > > > > ### Comment · Reviewer_y4Vs · 2021-11-23
> > > > > > **Info from OPUS: Yes; Clarification on "implications otherwise"**
> > > > > >
> > > > > > Thanks for following up!
> > > > > > 1. I would like you to include the OPUS information for completeness.
> > > > > > 2. By "implications otherwise" I meant that the limitation of the study (one dataset, six languages) should always be kept clear.
> > > > > >
> > > > > > Thank you again.

---

> > > > > > > ### Author Response · Authors · 2021-11-23
> > > > > > > **OPUS, dataset, languages**
> > > > > > >
> > > > > > > Thanks for your reply.
> > > > > > >
> > > > > > > Re 1: sure, no problem. We will add the OPUS information.
> > > > > > >
> > > > > > > Re 2: where specifically in the paper would you like this to be made more explicit? It is specified as we set up the paper and experiments --- in the abstract, Sections 1 & 2.
> > > > > > >
> > > > > > > To note:
> > > > > > > The analytic and the conceptual solutions are independent of datasets/languages tested. They are categorical, universal truths, all else being equal (i.e. unless something very fundamental in computing changes). That is, in the context of computing, there is no morphological complexity that can be held as relevant unless we model it explicitly. It is not an intrinsic property of language at large --- we agree on this, correct?
> > > > > > >
> > > > > > > "Language at large" means language in all aspects.
> > > > > > >
> > > > > > > Thanks.

---

> > > > > > > > ### Comment · Reviewer_y4Vs · 2021-11-24
> > > > > > > > **Confirming alignment**
> > > > > > > >
> > > > > > > > Thank you for the response. We are aligned. I do not require further edits, after another re-read. I look forward to (observing the process of) adoption of this paper in our research community, even if I'd be happier if multi-parallel resources permitted more breadth in its empirical setup.

---

> > > > > > > > > ### Author Response · Authors · 2021-11-24
> > > > > > > > > **Thank you**
> > > > > > > > >
> > > > > > > > > Thank you! Yes, we envision a plethora of possibilities with multiway parallel resources, more than what has been realized thus far. We'd appreciate it if you'd raise your score(s) and/or help advocate our work in the next evaluation phase(s) such that we could have the opportunity to discuss these new and exciting potential research directions further with the community.
> > > > > > > > >
> > > > > > > > > We will add an appendix re the current multiway parallel data situation (at least as of today, 24Nov2021), including but not limited to the OPUS information.
> > > > > > > > >
> > > > > > > > > Again, thank you very much.

---

> > > > > > > > > > ### Author Response · Authors · 2021-11-29
> > > > > > > > > > **"appendix" is turning into another paper...**
> > > > > > > > > >
> > > > > > > > > > Dear Reviewer y4Vs:
> > > > > > > > > >
> > > > > > > > > > This "supplementary material" on the status of parallel resources is about to look like another conference paper (at least a short one). IF you do not object, we would like turn it into a resource review article and submit it to another venue in the meanwhile. We will cite it in the camera-ready version of this paper. Would that be alright with you?
> > > > > > > > > >
> > > > > > > > > > Thank you in advance for your feedback.
> > > > > > > > > >
> > > > > > > > > > Sincerely,
> > > > > > > > > >
> > > > > > > > > > Authors3684

---

> > > > > > > > > > > ### Comment · Reviewer_y4Vs · 2021-11-29
> > > > > > > > > > > **Agreed, just as long as the other paper is visible**
> > > > > > > > > > >
> > > > > > > > > > > Sounds good to me. Good luck with the other submission! Hopefully the submission preprint may be visible meanwhile, but I'm fine either way---good luck!

---

> > > > > > > > > > > > ### Author Response · Authors · 2021-11-29
> > > > > > > > > > > > **Thanks!**
> > > > > > > > > > > >
> > > > > > > > > > > > Yes, we will cite a non-anonymous copy of that paper in the camera-ready. Hope we get to prepare a camera-ready for ICLR2022! Thank you!

---

### Author Response · Authors · 2021-11-10
**Thank you for all your reviews of our initial submission (v0.1)**

First of all, we would like to take the opportunity to thank all our reviewers for their reviews thus far.

We will provide our responses to each of your concerns raised in this first set of reviews, along with paper revisions (where applicable), in the coming days. We are confident about the correctness, technical and empirical novelty and significance of our findings, as well as the qualitative insights communicated through this work. We thank you in advance for all your upcoming feedback and suggestions and look forward to a fruitful discussion period.

As all of you do see, this is a very important paper with a high potential for impact. It brings in a new perspective, opens up exciting research directions, and makes a strong contribution to addressing an important problem. [1] These are criteria for outstanding paper awards as mentioned in the Opening Remarks at ICLR 2021 [2]. We are committed to improve our formulation with your help, while remaining true to scientific integrity and furthering research in healthier directions. We hope we can work together to get there.

We also welcome comments and suggestions from the general audience.

Thank you.

[1] We will bring these aspects to your attention in our upcoming comments.
[2] https://iclr.cc/virtual/2021/remarks/4385 (3:28-4:15)

---

### Author Response · Authors · 2021-11-13
**v0.2 uploaded**

Dear Reviewers:

This version aims to address most concerns raised re formulation.

As authors are not given any additional space this year to address reviewers' feedback, we ask for your understanding if you do not see your suggested edit in this version. We took in most suggestions, but could not accommodate all. We do hope to be able to submit the extended version of this work to a journal after the conference. This here is just a more concise version. That said, if you should feel somewhat strongly about an edit, please do not hesitate to let us know.

Re table and graphs:

Most of you wrote that the graphs are too small. We do want to point out that, for this paper, since we are studying the differences between languages, not how well the languages perform, Table 1 sufficiently summarizes the empirical results on disparity. It lists the number of language pairs with significant differences. The bottom line with p-value 0.001 shows that there are 0 language pairs with significant differences under Pinyin and ARRU\_t for both source and target. That's all that is needed to support the empirical contribution re the elimination of disparity in this paper. We do not need to pay attention to the absolute scores.

The figures, on the other hand, are primarily intended to show representation relativity, i.e. that different representations behave differently. For that, one does not have to look into the details. Just the basic patterns.

To get a visual confirmation of the disparity pattern, one could, however, look at the spaces between the lines. The outliers or languages that are significantly different are usually further from the others. The figures are also available enlarged in App. E. Compare the spaces between sets of lines with the same target language.

Another (off-the-paper) ethics statement:

As the majority of NLP or "computing with language data" is heavily entwined with some form of a "word"-based, meaning-driven operation which has been ingrained for generations in the tradition of various forms of language science and engineering and symbolic/human-centric AI, "we are keenly aware that both our goal and our means can be criticized" (to cite [1]). There are reasons why we think clarification is needed.

There is a gap between our science, which connects with our "typical" worldview, and our engineering. We recommend, therefore, that a "word"-free perspective of language brought forth in this paper be treated as an avant-garde research initiative. As a community, we are under-informed as to how the world (of NLP, or in general) would look like without "words". We do not know which of our past assumptions and methods should continue to hold in finer granularities. So far, almost all our expectations about language are from a "word"-based perspective. Although some NLP researchers did work with character methods as early as a few decades ago, most of us never looked at language in terms of characters and bytes as having a different "behaviorial pattern". Up until now, we would use finer-grained units to model "word"-based expectations of language.

Special projects should be dedicated to evaluate how and which (neural or non-/pre-neural) methods work across different granularities (not only to test which would outdo which in what setting and with what equipment, but also to see if our prior expectations hold). Knowledge and methods need to be translated into finer granularities, more quality data need to be collected for more rigorous evaluation, experiments need to be run and results interpreted. At the same time, we need to have a transparent, fair, and comprehensive curriculum in place. "Word"/grammar-based methods may be cheaper and faster, but if this is the only perspective we teach our students, the only identity they'd be given to relate to technology at large, they may have trouble reconciling with future methods and their careers might also be easily jeopardized by an upgrade in hardware. In an age where technology advances at an unprecedented rate, we need to make sure that we equip our students with sustainable skill sets, beyond what is needed for cheap and fast solutions. We should also instill in them an appreciation for comprehensive understanding and transparency in science, as well as compassion and respect for other professionals as we help each other grow.

We have yet to address questions from Reviewers kpoN, 86m6, UWb7, and UwwP individually.

Thank you.

[1] Collobert et al. 2011. Natural Language Processing (Almost) from Scratch. https://jmlr.org/papers/volume12/collobert11a/collobert11a

---

### Author Response · Authors · 2021-11-17
**Patterns, "Language" --- clarification on these might help**

Dear Reviewers:

We have reflected on why there might be an unease/dissatisfaction for some.

**Patterns**

There may perhaps be a qualitative dissatisfaction in not being able to cover the ZH script pattern --- though it is not just "script" per se, because Abjad, Cyrillic... etc. are also known as "scripts". One can think of it as a difference in *patterns* based on the phonetic vs logographic distinction (as mentioned in [1], though they only speak of it as a "textual" phenomenon in machine processing, we think their claim can actually be broadened. The hard part is to find an audience which is informed about human languages and text processing, but does not, say, "believe in", or is ready to work without the confines of, "word"/"sentences" or morphology/syntax). [2]

Let's say, there are two classes of languages --- the "phonetic class" and the "logographic class".

For the "phonetic class", some of the recurring subsequences are sometimes understood/studied as morphs/morphemes. But morphology only recognizes units that are meaningful. A subsequence that recurs that is not a meaningful unit is not considered a morph(eme). Also, a subsequence that crosses "word" boundaries is also usually not considered a morph(eme). Hence, morph(emes) constitute only a subset of all possible recurring subsequences or of all possible patterns in the "phonetic class" of languages.

Some humans see some recurring subsequences and spot them as meaningful units (also reinforced by the tradition of grammar and "words"). But machines recognize more possible recurring subsequences. Hence, "morphological complexity" is irrelevant in the context of computing unless we model it explicitly.

For the "logographic class", our being able to eliminate disparity with ZH script with byte representations depends on how byte representations are designed. But this depends on our implementation. (Dis-)parity is up to us.

Our claims and statements are correct, e.g. the one in the abstract:

"Evidence from data statistics, along with the fact that word segmentation is qualitatively indeterminate, renders a decades-long debate on morphological complexity (unless it is being intentionally modeled in a word-based, meaning-driven context) irrelevant in the context of computing."

(Side remark: historically, people refer to these aspects as "speech" and "script". But that's not exactly it. These are also two qualitatively different perspectives through which many humans look at the world and relate to language at large. However, most of us are only familiar with languages from one of these classes and hence there is something lost in translation in perspectives.) #one can disregard this side remark for now if it is too much to understand

**"Language"**

Another reason why one might find things unclear, we suspect, might have to do with the term "language".

To be able to talk about language at large (human/machine, intuition/use... etc. --- all aspects), one needs to know when "language" is being referred to as "language in the context of machine computing" (l_machine) and "language as a communicative system outside of this machine computing context" (say, l_human, for the sake of brevity). At present, it is customary in the NLP literature to use the term "language" to refer to l_machine --- that is also how we have used and intended to use the term "language" all along in _this paper_.

Another criterion that one needs to consider (though this is always implicitly assumed in academic/scientific literature) is that our conclusion and findings are "based on our current knowledge of the world and of languages and based on data availability". Data availability and our knowledge of the world inform each other in a circular manner. We do not cover languages that do not yet exist and we do not go into fine details about what is considered a "language" in general. (It is also as fuzzy as the concept "word", but _in computing_ there are ways for us to constrain this based on standards we establish.)

We will include most of this in the journal version of the paper. For this conference submission, we will note that our conclusion is based on current data availability (which implies our knowledge about the world). We will check our formulation in the paper again, but the above should help clarify our intended reading. Please let us know if this helps clarify things, and if you agree/disagree or have other concerns.

Thank you.

Authors3684

[1] Anonymous. A statistical typology of (textual) language in finer granularity. https://openreview.net/pdf?id=yySHIamiadH

[2] Discussing this might take more than a journal article! (And there is currently no inter-/transdisciplinary data science publication which goes into this granularity/depth with data, part of the reason why we submitted to ICLR with OpenReview because we suspect we might be understood by those who are not confined by the way we currently see things within the NLP circles proper.)

---

> ### Author Response · Authors · 2021-11-17
> **Additional note on coverage**
>
> Also, our "based on data availability" clause implies that we do not cover, and are not required to cover for this study, scripts that have not been encoded.
>
> (More info re script encoding on this front: https://linguistics.berkeley.edu/sei/index.html )

---

> ### Author Response · Authors · 2021-11-17
> **a _healthy_transdisciplinary space**
>
> Clarification on Footnote 2 here:
>
> "... And there is currently no inter-/transdisciplinary data science publication which goes into this granularity/depth with data, part of the reason why we submitted to ICLR with OpenReview because we suspect we might be understood by those who are not confined by the way we currently see things within the NLP circles proper.)":
>
> We should clarify that it is _not_ the case that the entirety of NLP community has operated on "word"-based, meaning-driven linguistic concepts. But as we mentioned before, disciplines are converging, the intent behind our comment here is to advocate for a healthy space in which community members have a shared goal of doing good science, free from latent "interests" or "rivalry" between domains/groups, between "tradition" and "new school".
>
> One might think data science is to suffice in covering all kinds of data types or sciences. But data can be represented in so many ways in computing, do we have an interpretative and evaluative science for "representation learning" beyond pure numerical methods? Data can be used for applications, but there are also scientific aspects of data that need reconciling with and deserve understanding and developing. These can inform/enrich domain-specific knowledge. Where can cross-/trans-disciplinary new ideas be fostered? Where can one communicate scientific findings when the goal is to evaluate traditional assumptions or methods in relation to neural ones, to translate between systems of coarser and finer granularity, or to provide scientific interpretations and knowledge effected by new technologies?
>
> We hope our work would also help prompt a community discussion on these issues.

---

> ### Author Response · Authors · 2021-11-28
> **Re the "side remark" above**
>
> We thought to clarify so there is more clarity re the "side remark" mentioned in passing earlier, repeated here:
>
> > (Side remark: historically, people refer to these aspects as "speech" and "script". But that's not exactly it. These are also two qualitatively different perspectives through which many humans look at the world and relate to language at large. However, most of us are only familiar with languages from one of these classes and hence there is something lost in translation in perspectives.) #one can disregard this side remark for now if it is too much to understand
>
> Humans native to a language with a phonetic-alphabetic script tend to see "logographs" as just a script, as "orthography" [i]. But to a (monolingual) human native to a language with a logographic script, logographs are the only signs that are mapped to their language. Logographs are not _just_ the "written" signs. To those humans, they represent their language.
>
> There is a culture gap in our perspectives, in our sciences, but there is an opportunity for ML/NNs to close that gap. This would be a significant advance in our science as a whole, to not just say we embrace **diversity and inclusivity, but also to enact it**.
>
> [i] There are scholarly debates as to whether Saussure, who posited the nature of sign to be arbitrary, knew ZH.

---

> > ### Author Response · Authors · 2021-11-28
> > **Re "logographic class"**
> >
> > Now that we clarified the side remark, a note on this "logographic class":
> >
> > The statistical property behind this "logographic class" is that it is short in sequence length and high in |V|. (The term "logographic class" can be thought of as a shorthand here, when in doubt, one should refer to the data statistics.)
> >
> > Do note that our use of the term "logographic class" is to contrast cases such as Vietnamese and ZH [a], for which there are minimal pairs in the UDHR dataset for both the phonetic alphabetic and the logographic representations but with different statistical profiles. That said, any symbolic descriptions have their shortcomings. For instance, one can see in Fig. 1 in [1] that KO is also somewhat shorter in length and higher in |V| but its script (Hangul), at least in the UDHR corpus, is not one with much "logography" involved.
> >
> > There is a relation between script, culture (or individual situations in longitudinal view), and language usage (but without involving "word"-based representations). There are opportunities to study these more "advanced" connections [b]. We didn't have a chance to discuss these opportunities in detail, as it is only supplemental to our submission. But our work will prepare the community to broaden the horizon in the way we see "language" or language data, it will lead to new paths in many more interesting directions.
> >
> > (We sense that there are readers out there who might lack a background in languages/linguistics (and history thereof). Hence the addendum here.)
> >
> > [a] As described in [1], but that work provides only supplemental evidence to some of our arguments, not directly related to our present review, hence we are not to defend it at length.
> >
> > [b] instead of, say, to have our next-gen students obsess over grammar and grammaticality (again)
> >
> > Again, [1] refers to: Anonymous. A statistical typology of (textual) language in finer granularity. https://openreview.net/pdf?id=yySHIamiadH

---

### Author Response · Authors · 2021-11-23
**v0.3 uploaded**

Dear Reviewers:

v0.3 has been uploaded.

We'd be happy if you wouldn't mind taking a look. Whatever we can revise before the deadline for revisions at 2359, 22Nov2021, AOE, we will try to do so. If not, please rest assured that we will see to it later.

EDIT/Addendum 20211123: Please note that it has not been possible for us to revise the abstract or the keywords on OpenReview (the information you see at the top of this page). There are minor edits to our abstract compared to the original submission. Please read the one from v0.3, repeated here:

> We perform systematically and fairly controlled experiments with the 6-layer Transformer to investigate  the hardness in conditional-language-modeling languages which have been traditionally considered morphologically rich (AR and RU) and poor (ZH). We evaluate through statistical comparisons across 30 possible language directions from the 6 languages of the United Nations Parallel Corpus across 5 data sizes on 3 representation levels --- character, byte, and word. Results show that performance is relative to the representation granularity of each of the languages, not to the language as a whole. On the character and byte levels, we are able to eliminate statistically significant performance disparity, hence demonstrating that a language cannot be intrinsically hard. The disparity that mirrors the morphological complexity hierarchy is shown to be a byproduct of word segmentation. Evidence from data statistics, along with the fact that word segmentation is qualitatively indeterminate, renders a decades-long debate on morphological complexity (unless it is being intentionally modeled in a word-based, meaning-driven context) irrelevant in the context of computing. The intent of our work is to help effect more objectivity and adequacy in evaluation as well as fairness and inclusivity in experimental setup in the area of language and computing so to uphold diversity in Machine Learning and Artificial Intelligence research. Multilinguality is real and relevant in computing not due to canonical, structural linguistic concepts such as morphology or "words" in our minds, but rather standards related to internationalization and localization, such as character encoding --- something which has thus far been sorely overlooked in our discourse and curricula.

Thank you very much.

Sincerely,

Authors3684

---

> ### Author Response · Authors · 2021-11-23
> **v0.3.2**
>
> v0.3.2 is the latest

---

### Comment · Reviewer_UwwP · 2021-11-27
**Changes based on the discussion**

I haven't been able to join the discussion actively (due to health reasons), but after reading all the discussion so far, I am happy to bump my score to a 6: I still am not convinced that the paper/experiments show us something substantially new, but many other issues are addressed carefully by the authors, and the other reviewers seem to find the findings significant.

---

> ### Author Response · Authors · 2021-11-27
> **Re: Thank you for your score upgrade**
>
> Dear Reviewer UwwP:
>
> Thank you for your score upgrade and comment. (We hope you are feeling better.)
>
> Re "I still am not convinced that the paper/experiments show us something substantially new": just so we can understand your perspective --- what findings in the past in **science** (for Multilingual NLP / Multilinguality / evaluation / representation learning for language data / AIxScience) do you consider to have shown you something **substantially new**?
>
> Our findings and insights have prompted morphologists to re-define morphology (morphology based on "words" has been around for _at least_ a few decades). Is it possible that you are by chance not aware of the fact that we do not (yet) have a proper science of language beyond/without "words" (and "sentences")?
>
> Would you mind please citing some previous work that bridged "an understanding between language sciences and engineering (the latter being the dominant focus in NLP), and between traditional symbolic sciences and ML", including but not limited to how the Transformer helped us _understand_ something about _language_ in the context of computing, and how a "word"-free view is worth promoting?
>
> This is a hybrid paper --- first of its kind in its transdisciplinarity. Sure, one can say, hybrids are not anything new because they are parts of something more traditional/canonical. But did the "traditional canon" make any of the contributions we did and document them for posterity, for development (i.e. did they open up new research directions with these findings)?
>
> - When "ML proper" studied DD, did anyone look into data types?
>
> - When "language engineering / NLP proper" was busy looking for the best scores for MT/LMing, did any one asked why some languages lagged behind and, more importantly, investigate the reasons behind and the **discrepancies** in results? Did they implicitly assume that all languages are fundamentally different or similar (i.e. differences not significant/justified)? Did they try to make sense of it to the extent that they would point out the systemic bias "implicit in our theoretical/scientific assumptions that results in the varying performance of different languages in computing" (§1) was in our "words"?
>
> - When "language sciences proper" studied language, did they (at least, most of them) ever do so without "words"? Did anyone indicate that they are aware that their goal is to reconcile categories of languages studied with a "linguistic structure" that is implicitly based on EN and similar languages?
>
> This disconnect between these different "fields" or different foci of studies has caused strife in the past. Part of it is also what is contributing to the mistrust and "hatred" towards ML and AI. People do not understand. Considering the most persistent and vocal bottleneck to technological advance is human, can we do something that would _at least_ allow people to understand ML/NNs better?
>
> The final stage of discussion period is not over until 29Nov2021. We'd be happy to discuss more until then through this weekend. We want people to value data science. This "paper" took a lot of work (more than 10k hours of manual labor) --- from experimental setup (excluding actual compute time), to figuring out the results (quantitative and qualitative disparties in ZH, DD, erraticity...), to looking up literature from various disciplines to come to a comprehensive account, to formulating our write-up in a way that could reach as many potential ICLR reviewers for our track as possible (at least that was our hope). We are certain that this helped many improve their understanding of the state of our art and sciences. We would like you to help us foster a scientific culture that does not take this work for granted.
>
> Last but not least, this work brings in new perspectives, opens up new research directions, and makes a strong contribution to addressing an important problem. (Just to reiterate a point we made from our very first post to all reviewers from 10Nov2021.) We'd appreciate it if you could let us know how we can improve so you would be **more impassioned in appreciating and supporting our effort**.
>
> We thank you for and look forward again to your feedback.
>
> Sincerely,
>
> Authors3684

---

> > ### Author Response · Authors · 2021-11-29
> > **update requested**
> >
> > Dear Reviewer UwwP:
> >
> > We hope you are doing well.
> >
> > Would you mind please giving us an approximate time window as to when you would be responding to our reply? The discussion period with the authors is coming to an end in a matter of hours and we would not like to miss your response due to time difference. We have been standing by.
> >
> > We hope you would take a chance to reconsider your rating in light of the points raised in support of the merits and substantial novelty of our work.
> >
> > Thank you!
> >
> > Authors3684

---

### Public Comment · ~Ada_Wan1 · 2024-03-25
**Note to submission**

This work, "Fairness in Representation for Multilingual NLP: Insights from Controlled Experiments on Conditional Language Modeling", is a re-formulation of a part of "Representation and Bias in Multilingual NLP: Insights from Controlled Experiments on Conditional Language Modeling" (submitted to and rejected by ICLR 2021, https://openreview.net/forum?id=dKwmCtp6YI).

---

### Decision · Program_Chairs · 2022-01-20

**Decision:**

Accept (Spotlight)

**Comment:**

The authors address a very important question pertaining to the relevance of morphological complexity in the ability of transformer based conditional language models. Through extensive (controlled) experiments using 6 languages they answer as well as raise very interesting questions about the role of morphology/segmentation/vocab size which mat spawn more work in this area.

All the reviewers were positive about the paper and agreed that the paper made significant contributions which would be useful to the community. More importantly, the authors and reviewers engaged in meaningful and insightful discussions through the discussion phase. The authors did a thorough job of addressing all reviewer concerns and changing the draft of the paper accordingly.

I have no hesitation in recommending that this paper should be accepted.